# Poisoning the Inner Prediction Logic of Graph Neural Networks for Clean-Label Backdoor Attacks

## Abstract

Graph Neural Networks (GNNs) have achieved remarkable results in various tasks. Recent studies reveal that graph backdoor attacks can poison the GNN model to predict test nodes with triggers attached as the target class. However, apart from injecting triggers to training nodes, these graph backdoor attacks generally require altering the labels of trigger-attached training nodes into the target class, which is impractical in real-world scenarios. In this work, we focus on the clean-label graph backdoor attack, a realistic but understudied topic where training labels are not modifiable. According to our preliminary analysis, existing graph backdoor attacks generally fail under the clean-label setting. Our further analysis identifies that the core failure of existing methods lies in their inability to poison the prediction logic of GNN models, leading to the triggers being deemed unimportant for prediction. Therefore, we study a novel problem of effective clean-label graph backdoor attacks by poisoning the inner prediction logic of GNN models. We propose **Ba-Logic** to solve the problem by coordinating a poisoned node selector and a logic-poisoning trigger generator. Extensive experiments on real-world datasets demonstrate that our method effectively enhances the attack success rate and surpasses state-of-the-art graph backdoor attack competitors under clean-label settings. Our code is available at https://anonymous.4open.science/r/BA-Logic.

## 1 Introduction

Graph neural networks (GNNs) Kipf & Welling (2017); Veličković et al. (2018); Hamilton et al. (2017) have achieved promising results in diverse graph-based applications, such as social networks Ni et al. (2024), finance systems Cheng et al. (2022), and drug discovery Bongini et al. (2021). Most GNNs update the representation of a node by aggregating features from its neighbors with the message-passing mechanism. Thus, the representations learned by GNNs can preserve node features and topology, facilitating various graph representation learning tasks Xu et al. (2019).

Despite GNNs having achieved success, they are vulnerable to graph backdoor attacks Dai et al. (2023); Xi et al. (2021); Zhang et al. (2021). We illustrate the general process of existing graph backdoor attacks in Fig. 1. As Fig. 1 shows, to create a backdoored graph, the adversary will attach a selected set of poisoned nodes with *triggers*. In addition, the adversary will *alter labels* of the poisoned nodes to the target class regardless of their original classes. Then, the GNN model trained on this poisoned graph will learn to associate the presence of the trigger with the target class, resulting in a backdoored GNN model. During the inference phase, the backdoored GNN will misclassify test nodes attached with the trigger to the target class while maintaining regular prediction accuracy on clean nodes. Some initial efforts Dai et al. (2023); Xi et al. (2021); Zhang et al. (2021) have demonstrated the effectiveness of the graph backdoor attacks. For instance, SBA Zhang et al. (2021) conducts pioneering research on graph backdoor attacks by adopting randomly generated triggers. Building upon this work, GTA Xi et al. (2021) proposed a trigger generator to guarantee the effectiveness of graph backdoor attacks. The state-of-the-art method UGBA Dai et al. (2023) adopts an unnoticeable constraint into the trigger generator to make the attack more unnoticeable while maintaining a high attack success rate. More detailed discussion of related works is in Sec. 6.

However, as the Fig. 1 illustrates, the majority of graph backdoor attacks, such as UGBA Dai et al. (2023) and DPGBA Zhang et al. (2024b), require attackers to alter the labels of trigger-attached poisoned nodes to the target class, regardless of their ground-truth labels. Such manipulation of the training labels is often impractical. In many application scenarios, the training set is annotated by experts of the dataset owners. It would be very expensive or even infeasible to manipulate the labels of the training set. For instance, fake account labels of Twitter are annotated and stored within a well-protected back-end

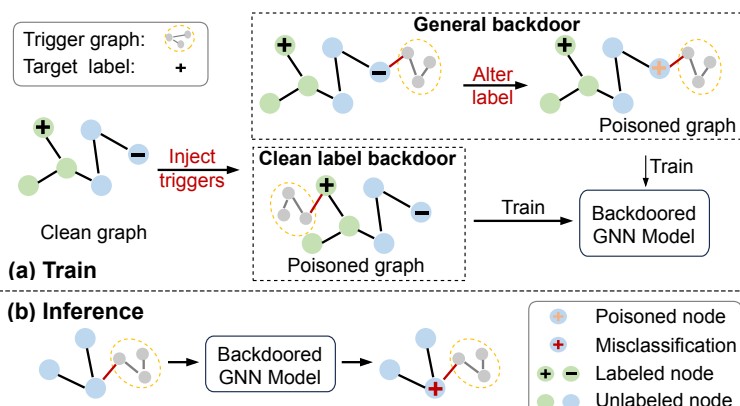

Figure 1: Illustration of graph backdoor attacks under both the general and clean-label settings.

system, making it nearly impossible for attackers to alter the labels Alothali et al. (2018). Furthermore, modifying labels of training samples can increase the risk of being detected. Therefore, it is crucial to investigate backdoor attacks under the clean-label setting. Specifically, as illustrated in Fig. 1, clean-label backdoor attackers inject triggers into training samples of the target class without modifying their labels, which is a more practical and challenging attack scenario. Some initial efforts Fan & Dai (2024); Xu & Picek (2022) have been conducted for clean-label graph backdoor attacks. For instance, Fan & Dai (2024) employs a single node as an efficient trigger, while Xu & Picek (2022) uses random graphs, respectively.

Despite the state-of-the-art general graph backdoor methods and initial attempts at clean-label backdoor attacks, our preliminary analysis in Sec. 2.3 reveals that these methods often fail to effectively poison the decision logic of target GNN models, resulting in poor backdoor performance. More precisely, our experiments indicate that during the poisoning phase, for a training sample attached with the poisoning trigger, clean neighbors dominate the prediction of the target GNN model. By contrast, injected triggers would be treated as irrelevant information, resulting in poor backdoor attacks. In fact, under the clean-label setting, poisoned samples that are attached with triggers are correctly labeled with ground-truth labels (target class). Consequently, during the training phase, the GNN model naturally learn correct patterns associated with the labeled class, thus ignoring the injected triggers during prediction. To address this problem, it is promising for attackers to explicitly guide the model's inner prediction logic to emphasize the injected triggers when predicting the poisoned nodes. Though promising, the works on poisoning the inner logic of GNNs for clean-label backdoor attacks are rather limited.

Therefore, in this paper, we study a novel and essential problem of poisoning the inner logic of GNN models for effective clean-label graph backdoor attacks. In essence, we face two technical challenges: *Firstly*, how to obtain triggers capable of poisoning the inner prediction logic of target GNN models. *Secondly*, the budget for the number of triggers injected is generally limited, so how to fully leverage the budget of poisoned nodes for effective inner prediction logic poisoning. In an attempt to address these challenges, we propose a novel Clean-Label Graph Backdoor Attack by Inner Logic Poisoning (Ba-Logic). Ba-Logic employs a logic-poisoning trigger generator, guided by a novel prediction logic poisoning loss. To better utilize the budget of poisoned nodes, Ba-Logic further employs a poisoned node selection module for logic poisoning. We summarize our contributions as follows:

- We study a novel problem of poisoning the inner prediction logic of target models for clean-label graph backdoor attacks;
- We introduce an innovative framework, Ba-Logic, which is capable of optimizing the poisoned node set and generating logic-poisoning triggers for effective clean-label backdoor attacks;
- We conduct comprehensive experiments for diverse target GNN models across a wide range of real-world graph datasets. Consistent results unequivocally demonstrate the superiority of Ba-Logic over state-of-the-art backdoor attacks. This substantiates the significant improvement in the effectiveness of clean-label graph backdoor attacks, achieved through the novel inner prediction logic poisoning strategy we present.

## 2 Preliminaries Analysis

In this section, we present preliminaries of logic poisoning for clean-label backdoor attacks and analyze the limitations of existing methods in logic poisoning.

### 2.1 Notations

A graph $\mathcal{G} = (\mathcal{V}, \mathcal{E})$ consists of a set of $N$ nodes $\mathcal{V} = \{v_1, \ldots, v_N\}$ and edges $\mathcal{E}$. $\mathbf{A} \in \mathbb{R}^{N \times N}$ is the adjacency matrix and $\mathbf{X} \in \mathbb{R}^{N \times d}$ is the node feature matrix. This work focuses on an inductive node classification task, where only a subset of nodes $\mathcal{V}_L$ has assigned labels $\mathcal{Y}_L$ for training, and unlabeled nodes are denoted as $\mathcal{V}_U$. Test nodes $\mathcal{V}_T$ are unavailable during training.

### 2.2 Threat Model of Inner Logic Poisoning for Clean-Label Backdoor Attacks

**Attacker's Goal:** By attaching backdoor triggers to a subset of training nodes $\mathcal{V}_P$, attackers aims to poison the logic of the target GNN $f_\theta$ to associate the backdoor triggers with the target class $y_t$. During the inference phase, the logic-poisoned GNN $f_\theta$ model will misclassify the test nodes $\mathcal{V}_T$ that attached with trigger $g$ as the target class $y_t$. Meanwhile, the logic-poisoned GNN $f_\theta$ maintains accuracy on clean test nodes with no trigger attached.

**Attacker's Knowledge and Capability:** In the clean-label graph backdoor attack, *attackers are not capable of altering the labels of nodes*. During training, attackers can only attach triggers to a subset of labeled nodes $\mathcal{V}_P \subset \mathcal{V}_L$ to poison the target model. During inference, attackers can only attach triggers to the target test nodes. Following Zhang et al. (2024b); Dai et al. (2023), the information about the target model, such as architectures and hyperparameters, is unavailable to attackers. Instead, attackers can only use a surrogate model to transfer the attack to unseen target models. This black-box threat model poses a strict limitation for attackers. A threat model level comparison with existing work is provided in Appendix B.

### 2.3 Existing Methods Fail to Poison the Inner Prediction Logic

In this subsection, we conduct empirical and theoretical analysis to show that existing methods fail to poison the inner logic under clean-label setting, resulting in poor clean-label backdoor results.

**Performance of Existing Methods under Clean-Label Setting.** We employ three state-of-the-art graph backdoor methods, namely GTA Xi et al. (2021), UGBA Dai et al. (2023), and DPGBA Zhang et al. (2024b). We extend them to the clean-label setting and denote the extended methods as **GTA-C**, **UGBA-C**, and **DPGBA-C**, where the suffix -C indicates it as a variation for the clean-label setting that only poisons labeled nodes of the target class without altering their labels. We also include a clean-label backdoor attack **ERBA** Xu & Picek (2022), which injects Erdös-Rényi random graphs Erdos et al. (1960) to labeled nodes of the target class as triggers. We report the average attack success rate (ASR) and clean accuracy of 5 runs on Pubmed in Tab. 1. From the table, it is evident that (i) all the methods exhibit poor ASR with the number of poisoned nodes $\mathcal{V}_P$ set as 100; (ii) even with a larger $|\mathcal{V}_P|$, the ASR improves marginally. The results confirm the inadequacy of existing graph backdoor attacks under clean-label settings.

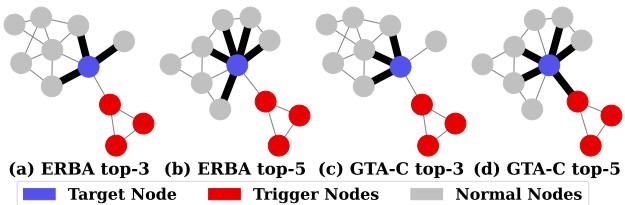

**(a) ERBA top-3  (b) ERBA top-5  (c) GTA-C top-3  (d) GTA-C top-5**

■ **Target Node**   ■ **Trigger Nodes**   ■ **Normal Nodes**

Figure 2: GNNExplainer's visualization of important subgraphs in a poisoned node's computational graph. We bold the edges connecting the poisoned node and the top-3 (a, c) and top-5 (b, d) most important nodes, respectively.

Table 1: ASR | clean accuracy (%) on Pubmed.

| $|\mathcal{V}_P|$ | ERBA | GTA-C | UGBA-C | DPGBA-C |
|---|---|---|---|---|
| 100 | 22.2 \| 85.6 | 38.4 \| 85.1 | 71.1 \| 85.3 | 64.2 \| 85.2 |
| 200 | 22.5 \| 85.2 | 38.9 \| 85.0 | 71.4 \| 85.1 | 64.1 \| 85.1 |
| 300 | 23.0 \| 85.0 | 38.7 \| 85.2 | 71.5 \| 84.8 | 64.2 \| 84.7 |

Table 2: Important rate of triggers (%) on Cora.

| Top-$k$ | ERBA | GTA-C | UGBA-C | DPGBA-C |
|---|---|---|---|---|
| $k = 3$ | 12.3 | 21.0 | 42.4 | 33.8 |
| $k = 5$ | 15.1 | 22.6 | 44.3 | 34.7 |

**Impacts of Injected Triggers to Prediction Logic.** To assess the injected triggers' influence on backdoored GNN predictions, we propose a metric named **Important Rate of Triggers (IRT)** to quantify trigger contributions by measuring their proportion as top-$k$ critical nodes in compact graphs of poisoned nodes. The mathematical formulation is in Appendix C. Tab. 2 reports IRT values of existing methods on Cora, showing that the IRT values for subgraphs consisting of top-3 and top-5 important nodes remain low across all methods, indicating triggers are rarely critical for target class prediction. We further visualize the prediction logic of GCN model backdoored by ERBA and GTA-C, using GNNExplainer Ying et al. (2019) to extract top-$k$ important nodes for the classification in Fig. 2. Results reveal that triggers from both methods exhibit lower importance than poisoned nodes' clean neighbors, suggesting their limited influence on model predictions.

**Theoretical Analysis.** We further conduct a theoretical analysis to prove that low important rate of triggers would lead to poor attack performance under the clean-label setting. Following Dai et al. (2023); Zhang et al. (2025), we consider a graph $\mathcal{G}$ where (i) The node feature $\mathbf{x}_i \in \mathbb{R}^d$ is sampled from a specific feature distribution $F_{y_i}$ that depends on the node label $y_i$. (ii) Dimensional features of $\mathbf{x}_i$ are independent to each other. (iii) The magnitude of node features is bounded by a positive scalar vector $S$, i.e., $\max_{i,j} |\mathbf{x}_i(j)| \leq S$.

**Theorem 1.** *We consider a graph $\mathcal{G} = (\mathcal{V}, \mathcal{E}, \mathbf{X})$ follows Assumptions. Given a node $v_i$ with label $y_i$, let $deg_i$ be the degree of $v_i$, and $\gamma$ be the value of the important rate of trigger. For a node $v_i$ attached with trigger $g_i$, the probability for GNN model $f$ predict $v_i$ as target class $y_t$ is bounded by:*

$$\mathbb{P}(f(v_i) = y_t) \leq 2d \cdot \exp\left(-\frac{deg_i \cdot (1 - \gamma)^2 \cdot \|\mu_{y_t} - \mu_{y_i}\|_2^2}{2d \cdot S^2}\right), \tag{1}$$

*where $d$ is the node feature dimension, $\mu_{y_t}$ and $\mu_{y_i}$ are the class centroid vectors in the feature space for $y_i$ and $y_t$, respectively.*

The detailed proof is in Appendix C. Theorem 1 shows that the upper bound of the probability for predicting the trigger-attached $v_i$ as $y_t$ grows with the increase in the IRT value $\gamma$. Existing graph backdoor methods generally lead to a low important rate of triggers, resulting in poor attack performance under the clean-label setting. The analysis further motivates a new graph backdoor paradigm that poisons the inner prediction logic of GNNs for effective clean-label graph backdoor attacks. More empirical analysis on IRT and attack budget $\mathcal{V}_P$ is in Appendix A.8, and more empirical validations on theoretical analysis are in Appendix H.

## 3 Problem Definition

We denote the prediction on a clean node $v_i$ as $f_\theta(v_i) = f_\theta(\mathcal{G}_C^i)$, where $\mathcal{G}_C^i$ is the computational graph of node $v_i$. For a node $v_i$ injected with trigger $g_i$, the prediction from the model is denoted as $f_\theta(\tilde{v}_i) = f_\theta(a(\mathcal{G}_C^i, g_i))$, where $a(\cdot)$ is the trigger attachment operation. Let $S_\theta(\tilde{v}_i, g_i)$ denote the importance score of the trigger $g_i$ injected to the node $v_i$ determined by the target GNN $f_\theta$.

Our preliminary analysis in Sec. 2.3 shows that existing backdoor attacks suffer from a poor ASR under clean-label settings due to the failure to poison the inner logic of the target model. To effectively conduct clean-label graph backdoor attacks, we propose to generate triggers capable of poisoning the inner logic of the target model. More precisely, the proposed clean-label graph backdoor attacks aim to achieve the following objectives:

- For any node $v_i \in \mathcal{V}_P \cup \mathcal{V}_T$, after attachment with the generated trigger $g_i$, the backdoored GNN will classify $v_i$ as the target class $y_t$, i.e., $f_\theta(\tilde{v}_i) = y_t$.
- For poisoned nodes and test nodes attached with triggers, injected triggers should be identified as the most important nodes by the logic of backdoored GNN, i.e., maximizing $S_\theta(\tilde{v}_i, g_i)$ for all node $v_i \in \mathcal{V}_P \cup \mathcal{V}_T$.
- Constraints, including the number of poisoned nodes, the size of generated triggers, and other unnoticeable constraints, as the one outlined in Dai et al. (2023), should be met.

With the above objectives and the threat model discussed in Sec. 2.1, We can formulate the clean-label graph backdoor attack by poisoning the inner prediction logic as:

**Problem 1.** *Given a graph $\mathcal{G} = (\mathcal{V}, \mathcal{E})$ with a set of labeled training nodes $\mathcal{V}_L$ with labels $\mathcal{Y}_L$, we aim to learn a trigger generator $f_g: v_i \rightarrow g_i$ and select a set of nodes $\mathcal{V}_P \subset \mathcal{V}_L^t$ to attach logic-poisoning triggers so that a GNN model $f$ trained on the poisoned graph will classify the test node attached with the trigger to the target class $y_t$ by solving:*

$$
\begin{aligned}
\min_{\mathcal{V}_P, \theta_g} \quad & \sum_{v_i \in \mathcal{V}} l(f_{\theta^*}(\tilde{v}_i), y_t) - \beta S_{\theta^*}(v_i, g_i) \\
s.t. \quad & \theta^* = \arg\min_{\theta} \sum_{v_i \in \mathcal{V}_L \setminus \mathcal{V}_P} l(f_\theta(v_i), y_i) + \sum_{v_i \in \mathcal{V}_P} l(f_\theta(\tilde{v}_i), y_i), \\
& \forall v_i \in \mathcal{V}, g_i \text{ meets the required unnoticeable constraint} \\
& |g_i| \leq \Delta_g, \quad |\mathcal{V}_P| \leq \Delta_P
\end{aligned}
\tag{2}
$$

*where $\theta_g$ denotes the parameters of the trigger generator, $l(\cdot)$ denotes the cross-entropy loss, and $\beta$ is the hyperparameter to control the contribution of logic poisoning. The node size of the trigger $|g_i|$ is limited by $\Delta_g$, and the number of poisoned nodes is limited by $\Delta_P$. A surrogate GNN $f$ is applied to simulate the target GNN whose architecture is unknown. Various unnoticeable constraints can be applied to this problem. In this paper, we focus on the unnoticeable constraint in Dai et al. (2023).*

## 4 Methodology

Our preliminary analysis reveals two key challenges for logic poisoning clean-label graph backdoor attacks: (i) How to select the poisoned nodes that are most effective in logic poisoning? (ii) How to efficiently compute the objective function of prediction logic poisoning to guide the training of the clean-label backdoor trigger generator? To overcome the above challenges, we propose a novel method BA-LOGIC, and illustrate the overall framework in Fig. 3. As Fig. 3 shows, BA-LOGIC firstly identifies poisoned nodes of the target class with high prediction uncertainty. The logic-

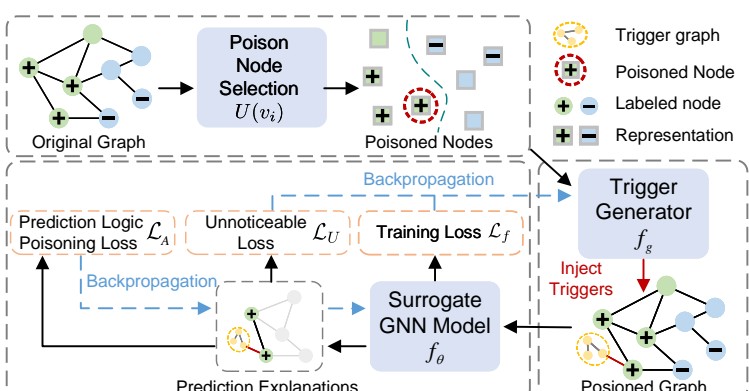

Figure 3: Framework of BA-LOGIC.

poisoning trigger generator $f_g$ utilizes node features as a basis to iteratively optimize triggers for the clean-label graph backdoor. To guide the training of the logic-poisoning trigger generator $f_g$, an efficient objective function for poisoning the prediction logic of the surrogate GNN model is employed. Additionally, to ensure the unnoticeability of triggers, a constraint is incorporated into the training of the trigger generator. Next, we provide details for each component.

### 4.1 Poisoned Node Selection for Logic Poisoning

In this subsection, we present the details of the poisoned node selection. It indicates the optimal positions for trigger injection, after which the model trained on the trigger-injected graph is backdoored, and arbitrary test nodes could be successfully attacked by attaching triggers during inference. The poisoned nodes are randomly selected in several existing clean-label graph backdoor methods Xu & Picek (2022); Xing et al. (2024), resulting in waste of the limited attack budget on useless poisoned nodes. For example, some labeled nodes exhibit typical patterns strongly associated with the target class. Thus, it would be difficult for the injected triggers to attain high importance scores with the presence of these typical patterns, thereby invalidating the backdoor trigger in logic poisoning.

Therefore, we design a process of identifying the set of poisoned nodes $\mathcal{V}_P \in \mathcal{V}_L^t$ that are most effective for backdoor. Specifically, we propose to select training nodes of the target class that exhibit high uncertainty

as predicted by the clean GNN. Intuitively, high uncertainty indicates irregular patterns that are weakly associated with the target class $y_t$. By contrast, triggers obtained by the generator will exhibit consistent patterns to poison the prediction logic. Thus, when triggers are injected into such nodes, the model is more likely to treat these triggers as key features of the target class rather than irregular patterns. Specifically, to identify high uncertainty nodes, we utilize the output of a GCN trained on the clean graph $\mathcal{G}$ and label set $\mathcal{Y}_L$. Let $f_S$ denote the well-trained GCN model, the probability of node $v_i$ predicted as the class $y_j$ can be obtained by:

$$p(y_j|v_i) = f_S(v_i)_{y_c} \tag{3}$$

Moreover, we design an uncertainty metric based on the following two aspects: (i) The probability that the node predicted as the target class, i.e., $p(y_j|v_i)$, should be low; (ii) The node is also expected to be uncertain for all other classes, i.e., the entropy of the probability vector is high. Let $C$ be the number of classes. The score function of poisoned node selection for attacking logic is:

$$U(v_i) = (1 - p(y_t|v_i)) - \sum_{j=1}^{C} p(y_j|v_i) \log p(y_j|v_i) \tag{4}$$

After getting the score of each $v_i \in \mathcal{V}_L^t$, we select nodes with top-$\Delta_P$ highest scores to construct $\mathcal{V}_P$ that satisfies the attack budget.

## 4.2 Inner Prediction Logic Poisoning

With the poisoned nodes $\mathcal{V}_P$ selected for the clean-label backdoor attack, the backdoored graph to poison the target GNN can be constructed by inserting powerful logic-poisoning triggers. In this subsection, we introduce the design of the logic-poisoning trigger generator. Then, we present the objective function of prediction logic poisoning that guides the training of the trigger generator.

**Logic-Poisoning Trigger Generator.** To poison the logic of the target GNN model, the generated trigger must be capable of capturing the importance scores for predictions on a poisoned node. Therefore, the logic-poisoning trigger should be adaptive to the input node. To achieve this, we deploy a MLP model to simultaneously generate node features and the adjacency of the trigger $g_i$ for node $v_i$ by:

$$\mathbf{X}_i^g, \quad \mathbf{A}_i^g = \mathbf{MLP}\left(\mathbf{x}_i\right), \tag{5}$$

where $\mathbf{x}_i$ is the feature of node $v_i$. $\mathbf{X}_i^g \in \mathbb{R}^{s \times d}$ is the features of the trigger nodes, where $s$ and $d$ represent the size of the generated trigger and dimension of node features, respectively. $\mathbf{A}_i^g \in \mathbb{R}^{s \times s}$ represents the adjacency matrix of the generated trigger. As the adjacency matrix must be discrete, we deploy the updating strategy of discrete variables in a binarized neural network Hubara et al. (2016).

To build the backdoored graph dataset for model poisoning, the generated trigger $g_i = (\mathbf{X}_i^g, \mathbf{A}_i^g)$ will be attached to the corresponding poisoned node $v_i \in \mathcal{V}_P$. During inference, to mislead the backdoored GNN to predict the test node $v_i \in \mathcal{V}_T$ as target class $y_t$, the attacker would insert the trigger generated by $f_g$.

**Prediction Logic Poisoning Loss.** The prediction logic poisoning in BA-LOGIC aims to mislead the target model to treat triggers as crucial patterns for prediction. As shown in Eq.(2), this can be formulated as maximizing the trigger's importance score $S_\theta(\tilde{v}_i, g_i)$ in predicting the trigger-attached node $v_i \in \mathcal{V}$ as the target class $y_t$ by a surrogate GNN model $f_\theta$. Although GNN explainers such as GNNExplainer Ying et al. (2019) and PGExplainer Luo et al. (2020) are capable of computing importance scores for nodes, they necessitate additional optimization to generate explanations. This extra optimization step poses challenges for solving Eq.(2), both in terms of computational cost and gradient backpropagation. Thus, BA-LOGIC deploys the gradient-based explanation, i.e., Sensitivity Analysis (SA) Baldassarre & Azizpour (2019). Specifically, for the prediction $\tilde{y}_i = f(\tilde{v}_i)$ on a trigger-attached node $v_i$, SA computes importance scores using the norm of the gradient w.r.t the node $v_j$. Formally, the important score of node $v_j$ in predicting $v_i$ as the target class is computed by:

$$S(y_i^t, v_j) = \|\frac{\partial \tilde{y}_i^c}{\partial \mathbf{x}_j}\|_2, \tag{6}$$

where $y_i^c$ is score of predicting node $v_i$ attached trigger $g$ into the target class, and $\mathbf{x}_j$ represents the node features of $v_j$. The Eq.(6) will allow us the compute the importance scores of inserted triggers efficiently. Simply maximizing the importance scores of triggers as specified in Eq.(6) can lead to infinitely large gradients, which significantly degrade the utility of the target model. Alternatively, within the computational graph of a trigger-attached node $v_i$, BA-LOGIC enforces the importance scores of trigger nodes to exceed those of clean nodes by a predefined margin $T$. More precisely, we replace the term of maximizing importance scores of triggers in Eq.(2) with the following prediction logic poisoning loss:

$$\mathcal{L}_A = \sum_{v_i \in \mathcal{V}} \max\left(0, T - \left(\sum_{v_g \in g_i} S(y_i^t, v_g) - \sum_{v_c \in \mathcal{N}(v_i)} S(y_i^t, v_c)\right)\right), \tag{7}$$

where $\mathcal{N}(v_i)$ denotes the clean node net in the computational graph of the trigger-attached node $v_i$.

**Unnoticeable Constraint on Triggers.** As we need to bypass various defense methods, an unnoticeable constraint on the generated trigger is required. Our BA-LOGIC is flexible to various unnoticeable constraints of triggers. Following Dai et al. (2023), we propose the constraint that requires high cosine similarity between the poisoned node or target node $v_i$ and trigger $g_i$. Within the generated trigger $g_i$, the connected trigger nodes should also exhibit high similarity. Formally, the loss of an unnoticeable constraint on trigger generator $f_g$ is:

$$\min_{\theta_g} \mathcal{L}_U = \sum_{v_i \in \mathcal{V}} \sum_{(v_j, v_k) \in \mathcal{E}_B^i} \exp(-sim(v_j, v_k)), \tag{8}$$

where $\mathcal{E}_B^i$ denotes the edge set that contains edges insider trigger $g_i$ and edges attaching trigger $g_i$ and node $v_i$. $sim(\cdot)$ represents the computation of cosine similarity between vectors. The unnoticeable loss in Eq.(8) is applied on all nodes to guarantee that generated triggers meet the unnoticeable constraint on various nodes. More analysis on the unnoticeable constraint on triggers is provided in Appendix A.9

## 4.3 Final Objective Function of Ba-Logic

As stated in Eq.(2), a bi-level optimization between the logic-poisoning trigger generator $f_g$ and a surrogate GNN model $f$ is adopted to ensure the effectiveness of triggers in logic poisoning for the clean-label backdoor. In the lower-level optimization of Eq.(2), the surrogate GNN model is trained on the backdoored dataset by:

$$\min_{\theta} \mathcal{L}_f = \sum_{v_i \in \mathcal{V}_L \setminus \mathcal{V}_P} l(f_\theta(v_i), y_i) + \sum_{v_i \in \mathcal{V}_P} l(f_\theta(\tilde{v}_i), y_i) \tag{9}$$

The upper-level optimization in Eq.(2) aims for successful backdoor attacks and inner prediction logic poisoning. With the selected poisoned node set $\mathcal{V}_P$, the prediction logic poisoning loss $\mathcal{L}_A$ in Eq.(7), and the unnoticeable constraint $\mathcal{L}_U$ in Eq.(8), the optimization problem in Eq.(2) can be finally reformulated as:

$$\min_{\theta_g} \sum_{v_i \in \mathcal{V}} l(f_{\theta^*}(\tilde{v}_i(\theta_g), y_t)) + \mathcal{L}_U(\theta_g) + \beta \mathcal{L}_A(\theta^*, \theta_g)$$
$$s.t. \ \theta^* = \arg\min_{\theta} \mathcal{L}_f(\theta, \theta_g), \tag{10}$$

where $\beta$ is the hyperparameter to control the contribution of prediction logic poisoning loss. $\theta_g$ denotes the parameters of the logic-poisoning trigger generator $f_g$. $\theta$ denotes the parameters of the surrogate model.

## 4.4 Optimization Algorithm

In this subsection, we present the algorithm for solving the bi-level optimization problem in Eq. (10).

**Lower-Level Optimization.** In the lower-level optimization, the surrogate GNN is trained on the backdoored dataset. To reduce the computational cost, we update surrogate model $\theta$ for $N$ inner iterations with fixed $\theta_g$ to approximate $\theta^*$:

$$\theta^{n+1} = \theta^n - \alpha_f \nabla_\theta \mathcal{L}_f(\theta, \theta_g), \tag{11}$$

where $\theta^n$ denotes model parameters after $n-$th iterations. $\alpha_f$ is the learning rate for training.

**Upper-Level Optimization.** In the outer iteration, the updated surrogate model parameters $\theta^N$ are used to approximate $\theta^*$. Moreover, we apply a first-order approximation in computing gradients of $\theta_g$ to reduce the computation cost further:

$$\theta_g^{k+1} = \theta_g^k - \alpha_g \nabla_{\theta_g} \big( \sum_{v_i \in \mathcal{V}} l(f_{\bar{\theta}}(\tilde{v}_i(\theta_g)), y_t) + \mathcal{L}_U(\theta_g) + \beta \mathcal{L}_A(\bar{\theta}, \theta_g) \big), \tag{12}$$

where $\bar{\theta}_s$ indicates the parameters when gradient propagation stopping. $\alpha_g$ is the learning rate of the training trigger generator. The time complexity analysis and training algorithm of BA-LOGIC are given in Appendix D and Appendix E, respectively.

## 5 Experiments

In this section, we conduct extensive experiments to answer the following research questions:

- **RQ1**: How does BA-LOGIC perform against state-of-the-art baselines under the clean-label setting?
- **RQ2**: How does BA-LOGIC generalize across diverse models, GNN tasks, and graph types?
- **RQ3**: Can BA-LOGIC maintain high attack success rate against various defending strategies?
- **RQ4**: How effective are the proposed components of BA-LOGIC for clean-label graph backdoor attacks?

### 5.1 Experimental Settings

**Datasets.** We conduct extensive experiments on **Cora** Sen et al. (2008), **Pubmed** Sen et al. (2008), **Flickr** Zeng et al. (2020), and **Arxiv** Hu et al. (2020) for node classification; **MUTAG**, **NCI1**, and **PROTEINS** Morris et al. (2020) for graph classification; **Cora**, **CS** Hu et al. (2020), and **Physics** Hu et al. (2020) for edge prediction. We also include diverse heterophilous graphs, including **Squirrel** and **Chameleon** Luan et al. (2022), **Penn** and **Genius** from Lim et al. (2021). Details of the datasets are provided in Appendix F.1.

**Compared Methods.** To highlight the superiority of our BA-LOGIC, we extensively compare BA-LOGIC with state-of-the-art graph backdoor attacks, including **GTA** Xi et al. (2021), **EBA** Xu et al. (2021), **UGBA** Dai et al. (2023), and **DPGBA** Zhang et al. (2024b) under clean-label settings. These methods originally require altering the labels of poisoned nodes. In our experiments, we extend them to the clean-label setting by only poisoning the labeled nodes of the target class. These extended baselines are denoted with the suffix **-C**. We also compare our BA-LOGIC with **ERBA** Xu & Picek (2022) and **ECGBA** Fan & Dai (2024), which are two latest clean-label graph backdoor attack methods. We further extend comparison to latest graph backdoor attack methods for more graph tasks, including **SCLBA** Dai & Sun (2025), **GCLBA** Meguro et al. (2024), **TRAP** Yang et al. (2022) for graph classification; and **SNTBA** Dai & Sun (2024), **PSO-LB** and **LB** Zheng et al. (2023) for edge prediction. More details of the compared methods are provided in Appendix F.2.

**Evaluation Protocol.** In this work, we focus on an inductive learning setting, *where attackers cannot access test samples during the graph poisoning.* To achieve this, we randomly mask 50% of the samples in the original graphs during training BA-LOGIC. Half of the masked samples are used as target samples for attack performance evaluation, while the other half is used for clean accuracy evaluation. Specifically, we evaluate all backdoor attacks using the average attack success rate **(ASR)** on target samples attached with triggers and the clean accuracy of backdoored model on clean test samples. A 2-layer GCN acts as the surrogate model for all evaluations. To reduce randomness, we run experiments on each target model 5 times and report the average results. More implementation details of BA-LOGIC are in Appendix F.3.

### 5.2 Clean-Label Backdoor Performance

To answer **RQ1**, we compare BA-LOGIC with six baselines across four datasets and three target GNNs under the clean-label settings outlined in Sec. 2.2. We select diverse GNNs as target models, including GCN, GAT,

Table 3: Average backdoor attack success rate and clean accuracy (ASR | clean accuracy (%)). Note that the surrogate model deployed in Ba-Logic is fixed as a 2-layer GCN.

| Dataset | Target Model | Vanilla Acc. | ERBA | ECGBA | EBA-C | GTA-C | UGBA-C | DPGBA-C | Ba-Logic |
|---|---|---|---|---|---|---|---|---|---|
| Cora | GCN | 83.78 | 18.22 \| 80.77 | 34.77 \| 79.48 | 29.13 \| 74.17 | 32.45 \| 80.45 | 68.32 \| 79.97 | 59.55 \| 79.88 | **98.52** \| **83.59** |
|  | GAT | 84.30 | 19.32 \| 82.08 | 35.19 \| 78.93 | 29.34 \| 74.23 | 35.85 \| 82.68 | 68.76 \| 82.81 | 59.02 \| 84.19 | **97.12** \| **83.76** |
|  | GIN | 84.26 | 19.17 \| 79.85 | 34.56 \| 79.92 | 29.30 \| 74.46 | 35.49 \| 79.63 | 67.75 \| 83.04 | 60.03 \| 81.56 | **98.97** \| **83.81** |
| Pubmed | GCN | 86.38 | 22.18 \| 85.58 | 37.46 \| 85.86 | 31.89 \| 85.60 | 38.84 \| 86.17 | 71.24 \| 85.31 | 64.19 \| 85.41 | **96.75** \| **86.03** |
|  | GAT | 86.51 | 22.24 \| 85.91 | 41.54 \| 85.61 | 30.13 \| 85.46 | 42.14 \| 85.78 | 66.07 \| 86.33 | 67.05 \| 85.21 | **94.88** \| **85.13** |
|  | GIN | 86.51 | 15.46 \| 85.57 | 43.14 \| 83.63 | 30.99 \| 85.44 | 42.34 \| 86.35 | 68.69 \| 85.81 | 66.17 \| 86.22 | **99.04** \| **86.21** |
| Flickr | GCN | 46.21 | 0.00 \| 45.75 | 39.47 \| 45.68 | 32.47 \| 45.68 | 48.12 \| 44.96 | 66.49 \| 43.99 | 69.66 \| 42.93 | **99.98** \| **46.05** |
|  | GAT | 46.07 | 0.00 \| 47.51 | 41.03 \| 47.65 | 31.93 \| 47.65 | 47.87 \| 47.64 | 68.78 \| 47.02 | 70.39 \| 46.31 | **99.72** \| **44.91** |
|  | GIN | 46.22 | 0.00 \| 45.62 | 41.71 \| 41.64 | 32.07 \| 41.64 | 48.04 \| 45.03 | 68.97 \| 45.98 | 68.12 \| 45.09 | **100.0** \| **46.14** |
| Arxiv | GCN | 66.58 | 0.01 \| 66.14 | 25.56 \| 66.16 | 28.64 \| 66.23 | 37.16 \| 66.29 | 69.71 \| 66.57 | 58.96 \| 66.82 | **98.04** \| **65.82** |
|  | GAT | 66.02 | 0.02 \| 64.09 | 26.03 \| 64.46 | 28.09 \| 64.41 | 36.45 \| 65.20 | 71.65 \| 65.10 | 59.13 \| 65.25 | **98.43** \| **65.40** |
|  | GIN | 66.73 | 0.02 \| 66.07 | 26.72 \| 62.01 | 27.35 \| 66.08 | 34.32 \| 65.87 | 71.01 \| 66.56 | 60.50 \| 66.73 | **97.62** \| **66.83** |

and GIN, and evaluate their vanilla accuracy on clean test nodes as a reference. We select four graphs with diverse characteristics, including Cora, Pubmed, Flickr, and Arxiv. As described by the evaluation protocol in Sec. 5.1, we report the results in Tab. 3, from which we observe:

- Across all datasets and models, Ba-Logic consistently achieves the highest ASR, typically close to 100%. It outperforms leading baselines such as UGBA-C and DPGBA-C, indicating that the clean-label setting is challenging for state-of-the-art methods, and Ba-Logic poisons inner prediction logic of target models effectively for clean-label backdoor attacks.
- Arxiv poses challenges due to its diverse classes and our fixed target-class setting. Despite requiring generalization to larger unseen graph parts, Ba-Logic maintains superior performance as the ASR of baselines drops, demonstrating its scalability. Experiments on larger graphs are in Appendix A.2.
- High ASR of Ba-Logic towards different target GNNs proves its transferability in backdooring various GNNs via logic poisoning. We further investigate how triggers crafted with one surrogate model affect more distinct target models in Sec. 5.3.
- Ba-Logic achieves comparable clean accuracy compared to vanilla GNN models, while other methods exhibit significantly larger clean accuracy drop. More experiments on the clean accuracy drop can be found in Appendix A.4.

## 5.3 Generalization Ability of Ba-Logic

To answer **RQ2**, we evaluate the generalization ability of Ba-Logic from three perspectives: diverse surrogate and target model architectures, various GNN downstream tasks, and different types of graphs.

**Flexibility to Surrogate and Target Model Architectures.** We further evaluate the flexibility of Ba-Logic by varying both the surrogate and target models across six backbones and two datasets, as shown in Fig. 4a. From the figure, we observe: **(i)** Despite inherent differences between surrogate and target models, Ba-Logic maintains highly effective, demonstrating strong transferability rooted in poisoning the shared message-passing mechanism for most GNNs. **(ii)** Sampling-based targets GraphSAGE and GraphSAINT (denoted as SAGE and SAINT in Fig. 4(a), respectively) appear marginally more robust against our Ba-Logic. We attribute this to the sampling of a subset of nodes, which dilutes the trigger's impact. More analysis on how diverse sampling strategies affect Ba-Logic's performance can be found in Appendix A.3.

**Generalization to More Tasks.** As node classification, graph classification, and edge prediction can all be formed into graph classification tasks, our Ba-Logic can be easily extended to graph classification and edge prediction tasks. Details of the extensions can be found in Appendix A.1. We compare our method with recent backdoor methods **SCLBA** Dai & Sun (2025), **GCLBA** Meguro et al. (2024), and **TRAP** Yang et al. (2022) for graph classification, and **SNTBA** Dai & Sun (2024), **PSO-LB** Zheng et al. (2023), and **LB** Zheng et al. (2023) for edge prediction. The results are given in Fig. 4b and 4c, where we observe that

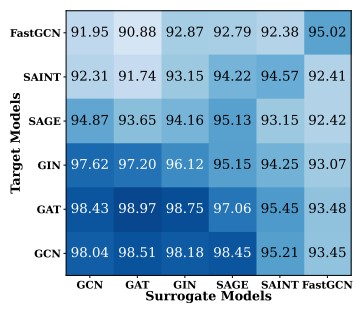

(a) Surrogate & Target Models

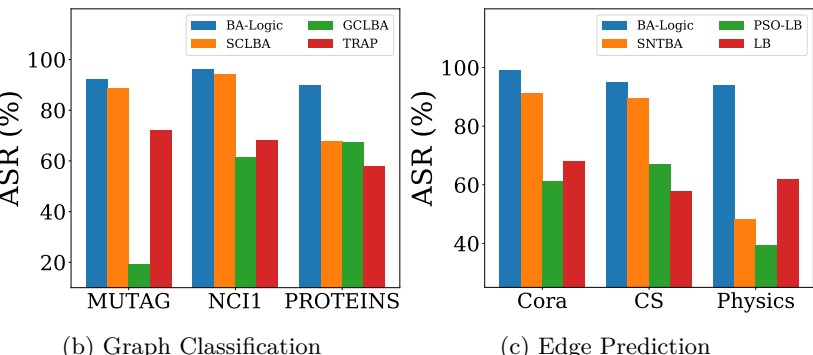

(b) Graph Classification

(c) Edge Prediction

Figure 4: Generalization ability of BA-LOGIC. (a) ASR (%) with varying surrogate and target models on Arxiv. (b-c) ASR (%) on graph classification and edge prediction.

BA-LOGIC achieves superior performance in backdooring both graph classification and edge prediction tasks without degrading clean accuracy. Complete results with clean accuracy are provided in Appendix A.1. This demonstrates the effectiveness of our BA-LOGIC in generalizing to other tasks.

**Generalization to Heterophilous Graphs.** We further evaluate BA-LOGIC on diverse heterophilous graphs, with node classification as the downstream task. We target both the heterophily-specific GNN models **ACMGCN** Luan et al. (2022) and **LINKX** Lim et al. (2021), along with a standard 2-layer GCN. We report results in Tab. 4. From the table, we can observe that our logic

Table 4: ASR (%) of BA-LOGIC in backdooring various target models on heterophilous graphs.

| Datasets | GCN | ACMGCN | LINKX |
|---|---|---|---|
| Squirrel | 99.07 | 98.14 | 98.79 |
| Chameleon | 99.03 | 98.53 | 98.43 |
| Penn | 96.34 | 96.71 | 95.32 |
| Genius | 98.17 | 97.92 | 93.71 |

poisoning method achieves high ASR across diverse graphs and backbones. This indicates our core idea, i.e., explicitly guiding the model's inner prediction logic to emphasize the injected triggers when predicting the poisoned nodes, maintains effectiveness consistently. More comparisons with baselines transferred from other domains are provided in Appendix A.5 and Appendix A.10. More analysis of our generalization ability under challenging label and feature settings can be found in Appendix I.

## 5.4 Attacks Against Defense Strategies

To answer **RQ3**, we evaluate the attack results against existing representative defense methods and defense strategies that we adaptively proposed for BA-LOGIC.

**Attacks Against Existing Defense Methods.** We first evaluate BA-LOGIC and baselines against representative defense methods, including **GCN-Prune** Dai et al. (2023), **RobustGCN** Zhu et al. (2019), **GNNGuard** Zhang & Zitnik (2020), and **RIGBD** Zhang et al. (2025). Results on Arxiv are reported in Tab. 5, while comprehensive evaluations and analysis are deferred to Appendix A.6. From the table, we observe that BA-LOGIC can effectively attack the robust GNN models. Compared with baselines, BA-LOGIC enhances ASR by at least 30%, showing the superiority of BA-LOGIC in attacking existing defense methods. The results also highlight a gap, indicating that existing methods fail to defend against the logic poisoning introduced by BA-LOGIC.

**Attacks Against Adaptive Defense Methods.** We further propose four adaptive defenses tailored for logic poisoning. These defense methods are oriented from diverse perspectives, including explainability regularization **(ER)**, gradient masking **(GM)**, collaborative defense **(CD)**, and sampling and masking **(SAM)**. Their shared idea is that penalizing over-reliance on specific samples in predictions can mitigate the risk of poisoned logic. Results on Arxiv are reported in Tab. 5, while detailed analysis and evaluations are deferred to Appendix G. From the table, we observe that BA-LOGIC consistently achieves the highest ASR, which exceeds 80% across various adaptive defenses and datasets. The results highlight the resilience

Table 5: Comparisons of ASR(%) against various existing and adaptive defense models.

| Category | Defense | ECGBA | DPGBA-C | UGBA-C | BA-LOGIC |
|----------|---------|-------|---------|--------|----------|
| Existing Defenses | GCN-Prune | 13.57 | 17.43 | 62.31 | **96.75** |
| | RobustGCN | 14.24 | 29.65 | 44.46 | **97.03** |
| | GNNGuard | 0.51 | 43.77 | 40.08 | **95.37** |
| | RIGBD | 0.01 | 0.01 | 0.19 | **93.23** |
| Adaptive Defenses | ER | 19.84 | 51.73 | 44.39 | **80.06** |
| | GM | 17.92 | 46.25 | 49.87 | **92.36** |
| | CD | 18.63 | 48.37 | 42.51 | **85.38** |
| | SAM | 18.21 | 47.06 | 40.94 | **81.09** |

Figure 5: (a-b) Ablation studies on GCN and GIN. (c) Hyperparameter sensitivity analysis on Arxiv.

of our BA-LOGIC against adaptive defenses, and the superiority of our BA-LOGIC in poisoning inner logic for clean-label backdooring.

### 5.5 Ablation Studies

To answer **RQ4**, we conduct ablation studies to explore the effectiveness of the poisoned node selector and the logic-poisoning trigger generator. To demonstrate the effectiveness of poisoned node selector, we randomly select poisoned nodes from the training graph and obtain a variant named BA-LOGIC\S. To demonstrate the benefit of logic poisoning trigger generator, we remove the inner logic loss in Eq. (7). In such a case, the BA-LOGIC degrades to a simplified variant named BA-LOGIC\T. We also implement BA-LOGIC\ST, a variant removing both modules. The ASR and standard deviations on Pubmed and Arxiv are shown in Fig. 5, where we observe that:

- Compared to BA-LOGIC\S, BA-LOGIC achieves better attack performance. The variance of ASR of BA-LOGIC is significantly lower than that of BA-LOGIC\S. It indicates that our selector consistently identifies diverse and influential nodes for logic poisoning.
- BA-LOGIC consistently outperforms both BA-LOGIC\T and BA-LOGIC\ST by a large margin. It highlights that our logic poisoning loss effectively guides the trigger generator to produce triggers that poison the target GNN's inner logic across diverse test nodes.

More analysis on the contribution of each module of our BA-LOGIC can be found in Appendix A.7.

### 5.6 Hyperparameter Analysis

We investigate the sensitivity of BA-LOGIC to two key hyperparameters: the margin $T$ in Eq. (7) that controls the expected importance gap between trigger and clean nodes, and the weight $\beta$ in Eq. (10) that balances the logic poisoning loss. We conduct parameter sweeps on four datasets and report results on Arxiv in Fig. 5c, with complete results in Appendix A.7. We observe that **(i)** graphs with higher average node degree (e.g., Arxiv) require larger $T$ to overcome the stronger influence of clean neighbors on the prediction logic, as more clean nodes dilute the trigger's importance scores; **(ii)** excessively large $T$ or $\beta$ degrades ASR,

as extreme values hinder the bi-level optimization of BA-LOGIC by causing gradient instability in the trigger generator; **(iii)** BA-LOGIC maintains robust performance across a wide range of $T$ and $\beta$ values, indicating that our method is not sensitive to hyperparameter choices in practice.

## 6 Related Works

**Graph Neural Networks.** Graph Neural Networks (GNNs) come into the spotlight due to their remarkable ability to model graph-structured data Chen et al. (2020); Hamilton et al. (2017); Kipf & Welling (2017); Gasteiger et al. (2019); Veličković et al. (2018); Wu et al. (2019). Recently, many GNN models have been proposed to further improve GNN performance Dai et al. (2024). There are also works that tailored to address the fairness Dai & Wang (2021a), robustness Wang et al. (2023); Dai et al. (2022a), and explainability Pope et al. (2019); Dai & Wang (2021b) challenge of GNNs. GNN models for heterophilous graphs are also designed Luan et al. (2022); Zhang et al. (2019); Lim et al. (2021). While the representational powers of GNN models have been well studied, their performance under diverse backdoor attacks has remained an open question Zhang et al. (2021). Our analysis reveals that GNNs' vulnerability to backdoor depends on whether their inner prediction logic is poisoned by the trigger pattern. Based on this, we propose a novel method that explicitly guides the model's inner prediction logic to emphasize the injected triggers when predicting the target nodes.

**Graph Backdoor Attacks.** Exploring backdoor attacks on graphs has aroused increasing interest among the graph learning community Zhang et al. (2021); Xi et al. (2021); Dai et al. (2023); Xu & Picek (2022); Wang et al. (2024c); Zhang et al. (2024b). Recent work achieves unnoticeable backdoors via similarity Dai et al. (2023) and distribution Zhang et al. (2024b) constraints in a dirty-label setting. For the clean-label setting, some initial efforts have been proposed Xu & Picek (2022); Xing et al. (2024); Chen & Zhou (2024); Dai & Sun (2025); Xia et al. (2025). Besides injecting subgraphs as triggers, a body of research has been proposed to conduct backdoor attacks by changing the most representative node features Xu et al. (2021); Xing et al. (2024); Wang et al. (2024a) or manipulating existing node and edges as triggers for graph classification and link prediction Yang et al. (2022); Meguro et al. (2024); Dai & Sun (2025; 2024); Zheng et al. (2023). However, modifying a graph's current features or structure is often impractical in real-world scenarios Alothali et al. (2018). Distinct from prior work, we study a novel problem of poisoning the prediction logic of target models for clean-label graph backdooring, and propose BA-LOGIC that optimizes the importance scores of triggers to exceed those of the clean neighbors of attached nodes for logic poisoning.

**Explaining the Prediction Logic of GNNs.** To enhance the trustworthiness of GNN's predictions, researchers have developed various explanation methods for GNNs Huang et al. (2022); Schnake et al. (2021); Luo et al. (2020), such as GNNExplainer Ying et al. (2019) that leverages mutual information to identify the most relevant subgraph, Grad-CAM variants tailored to GNNs Pope et al. (2019); Selvaraju et al. (2017), and $\pi$-GNN Yin et al. (2023) that pre-trains on synthetic graphs with ground-truth explanations. Explainability research makes the prediction logic of GNNs traceable, which in turn helps researchers understand GNN behavior Zhang et al. (2024a); Dai et al. (2024); Tang et al. (2023); Wang et al. (2024b). In this work, we leverage GNN explainability techniques to analyze and manipulate the prediction logic of GNNs for effective clean-label graph backdoor attacks.

## 7 Conclusion and Future Works

In this paper, we investigate the limitations of existing graph backdoor attacks under the clean-label setting. To overcome these limitations, we formalize the problem of poisoning the inner prediction logic of GNNs for effective clean-label graph backdoor attacks. Specifically, our methodology originates from the preliminary analysis of learning behaviors in the backdoored GNN models, leading to a theoretically grounded learning objective formulated as bi-level optimization for effective model poisoning. Extensive experiments on diverse datasets and graph learning tasks demonstrate that our approach can successfully induce backdoor behaviors across various GNN architectures under clean-label constraints, and BA-LOGIC remains resilient against various backdoor defense methods. Our results further validate that poisoning the inner prediction logic of GNNs enables effective clean-label graph backdoor attacks. Several promising directions emerge for future

research, including extending the research scope to other GNN downstream tasks, such as recommendation systems, and developing defense strategies that can both defend against logic poisoning and maintain clean accuracy simultaneously through the inverse application of our methodology.

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

**Table of Contents for Appendix**

## A  Additional Experiments

### A.1  Additional Discussion on Extending Method

In the main text, we initially design BA-LOGIC for node classification and subsequently extend it to graph classification and edge prediction for evaluation. This extension is grounded in the fact that these graph learning tasks are all built upon the common message-passing mechanism of GNNs Gilmer et al. (2020),

Table 6: ASR | clean accuracy (%) of BA-LOGIC in backdooring graph classification and edge prediction.

| | Graph Classification | | | | | Edge Prediction | | | |
|---|---|---|---|---|---|---|---|---|---|
| Datasets | SCLBA | GCLBA | TRAP | BA-LOGIC | Datasets | SNTBA | PSO-LB | LB | BA-LOGIC |
| MUTAG | 88.51 \| 62.77 | 19.27 \| 62.81 | 71.89 \| 61.44 | 92.17 \| 61.32 | Cora | 91.32 \| 80.43 | 61.35 \| 79.45 | 68.06 \| 76.17 | 99.01 \| 79.59 |
| NCI1 | 94.01 \| 61.43 | 61.35 \| 60.97 | 68.06 \| 61.51 | 96.19 \| 62.08 | CS | 89.62 \| 81.95 | 67.14 \| 76.13 | 57.67 \| 78.49 | 95.13 \| 79.65 |
| PROTEINS | 67.59 \| 71.25 | 67.14 \| 71.33 | 57.67 \| 71.49 | 89.67 \| 71.65 | Physics | 48.32 \| 62.77 | 39.27 \| 64.09 | 61.89 \| 63.44 | 93.81 \| 63.02 |

where the core objective is to learn representations by aggregating information from neighbors, which can be nodes, edges, and graphs. While the specific readout functions differ across the tasks, such as node-level output for node classification, graph-level pooling for graph classification, and pairwise node scoring for edge prediction, the underlying mechanism for generating embeddings is shared.

We presented the results in Tab. 6, demonstrating the consistent effectiveness of BA-LOGIC on conducting clean-label graph backdoor attacks on diverse tasks.

Below, we detail the extensions of BA-LOGIC for graph classification and edge prediction, respectively.

### A.1.1 Extend Ba-Logic to Graph Classification

We first discuss how to extend our BA-LOGIC to graph classification. To backdoor graph classification, we select the training graph $G_i$ from the target class $y_t$ to establish a poisoned graph set $\mathcal{G}_P$, and replace its nodes with trigger $g_i$ to poison the inner logic of target GNN models. For node $v_j \in G_i$ and target class $y_t$, importance score originates from Eq.(6) can be computed by:

$$S(y_i^t, v_j) = ||\frac{\partial \tilde{y}_i^c}{\partial \mathbf{x}_j}||_2 \tag{13}$$

And the logic poisoning loss originates from Eq.(7) can be computed by:

$$\mathcal{L}_A = \sum_{v_i \in G_i} \max(0, T - (\sum_{v_g \in g_i} S(y_i^t, v_g) - \sum_{v_c \in G_i \setminus g_i} S(y_i^t, v_c))) \tag{14}$$

Moreover, the adopted lower-level optimization originates from Eq.(11) aims to train $\theta$ on clean graphs $\mathcal{G}_L \setminus \mathcal{G}_P$ and poisoned graph set $\mathcal{G}_P$. The upper-level optimization, originating from Eq.(12), aims to optimize $\theta_g$ to minimize the loss for predicting $\mathcal{G}_P$ as $y_t$. To keep the trigger unnoticeable when against defense methods, constraint in Eq.(8) on $\theta_g$ should be kept.

With the adaptation stated above, we extend our method to select graph classification as a downstream task. In practice, we select a 2-layer GCN as the surrogate model, and report the average performance of three different target models, i.e., GCN, GIN, and GAT. Moreover, we add a global pooling layer to both the surrogate and target models and update the classifier for graph classification.

### A.1.2 Extend Ba-Logic to Edge Prediction

Noting that edge prediction is another widely adopted downstream task for GNN models besides node classification and graph classification, we further discuss how to extend our BA-LOGIC to backdoor GNN models that select edge prediction as the downstream task.

We consider the extension from a node-oriented perspective, in which the attacker attaches a trigger $g_{u,v}$ to node $u$ to make the model predict an edge $(u, v)$ based on the trigger. To achieve this, we maximize the influence of trigger nodes relative to clean neighbors. For node $v_j$ and edge $(u, v)$, the importance score of $v_j$ in predicting the edge is originated from Eq.(6), which can be formulated by:

$$S((u, v), v_j) = ||\frac{\partial \tilde{y}_{(u,v)}}{\partial \mathbf{x}_j}||_2, \tag{15}$$

where $\tilde{y}_{(u,v)}$ is the edge prediction given by the target model.

To backdoor edge prediction, we adopt logic poisoning loss, originating from Eq.(7), to maximize the influence of trigger nodes relative to clean neighbors. After attaching the trigger, the logic poisoning loss should be:

$$\mathcal{L}_A = \sum_{(u,v)\in\mathcal{E}} \max(0, T - (\sum_{v_g\in g_{u,v}} S((u,v),v_g) - \sum_{v_c\in\mathcal{N}_u} S((u,v),v_c))), \tag{16}$$

where $\mathcal{E}$ is the edge set, $\mathcal{N}_u$ are clean neighbors.

Moreover, the adopted lower-level optimization originates from Eq.(11) aims to train $\theta$ on clean edges ($\mathcal{E}_L \setminus \mathcal{E}_P$) and poisoned edges ($\mathcal{E}_P$). The upper-level optimization, originating from Eq.(12), aims to optimize the trigger generator $\theta_g$ for minimizing the prediction loss on $\mathcal{E}_P$. To keep the trigger unnoticeable when against defense methods, the constraint in Eq.(8) on $\theta_g$ should be kept.

With the adaptation stated above, we extend our method to select edge prediction as a downstream task. In practice, we select a 2-layer GCN as the surrogate model, and report the average performance of three different target models, i.e., GCN, GIN, and GAT.

## A.2   Additional Results of Attack Performance on Industry-Scale Graph

In the main text of our work, we have included multiple graph datasets with diverse characteristics for evaluation. To further demonstrate the scalability of Ba-Logic, we evaluate our method on OGBN-Products Hu et al. (2020), an industry-scale node classification dataset with 2.4 million nodes. Specifically, we select a 5-layer GCN and GraphSAGE as the surrogate model, and report the ASR|CA(%) of selecting GCN as the target model with the same layers. Due to the large size of OGBN-Products, which prevents full-batch training on GPU memory, we enabled mini-batch training with a large batch size following Luo et al. (2024). It is feasible in practice, as our method is not strongly dependent on graph structure and only requires approximate linear complexity. We also record the training time and GPU memory information during Ba-Logic conducting backdoor attacks. The results are recorded in Tab. 7, from which we have the following key findings:

Table 7: Training statistics and performance of Ba-Logic on **OGBN-Products**.

| Surrogate Model | Training Time (s) | GPU Memory Peak (GB) | ASR\|clean accuracy (%) |
|---|---|---|---|
| GCN | 1678.05 | 23.52 | 87.07 \| 78.51 |
| GraphSAGE | 1531.39 | 22.36 | 83.69 \| 80.27 |

- The scalability of our method is demonstrated with feasible resource usage on a large graph with 2.4 million nodes, indicating that our Ba-Logic remains practical at an industrial scale.
- Training time is acceptable given the performance, consistent with the approximately linear complexity with respect to graph size per optimization iteration as shown in Appendix D.

## A.3   Additional Results of Attack Performance towards Sampling-Based GNNs

We evaluate Ba-Logic with different graph sampling methods involved to explore the effectiveness of poisoning the logic of sampling-based GNNs. We expand our experiments by incorporating three widely adopted sampling-based GNNs: GraphSAGE Hamilton et al. (2017), GraphSAINT Zeng et al. (2020), and Fast-GCN Chen et al. (2018). Specifically, we have implemented GraphSAGE with two different graph pooling strategies, denoted as SAGE-max, SAGE-min, respectively. And we also implemented GraphSAINT with three different samplers, node, edge, and walk, denoted as SAINT-N, SAINT-E, and SAINT-W, respectively. To mitigate the randomness induced by sampling, we repeat each experiment 5 times and present the average results as shown in Fig. 6, from which we observe:

- Sampling methods can weaken Ba-Logic slightly, as Ba-Logic poisons the inner logic of the model by involving poison nodes attached to triggers in training.

- The impact is more significant when a large graph reduces the probability of sampling poison nodes. Specifically, ASR is most affected for layer-wise sampling (FastGCN) backbone, as it samples a fixed number of nodes in each layer; Moreover, ASR of node-wise sampling (GraphSAGE) and subgraph-wise sampling (GraphSAINT) backbones are not greatly affected, as the samplers will significantly increase the probability of sampling poison nodes.

- Among the three GraphSAINT samplers, Ba-Logic achieves the highest ASR when facing SAINT-W. This is because SAINT-W can sample the complete trigger through random walks, amplifying the impact of logic poison.

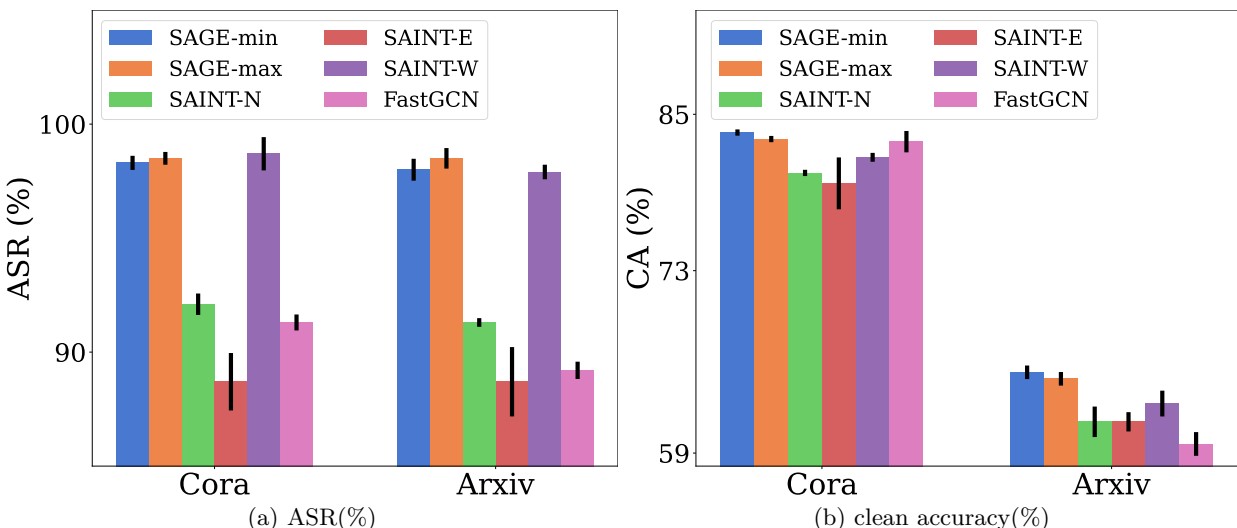

(a) ASR(%)    (b) clean accuracy(%)

Figure 6: Performance of Ba-Logic on sampling-based GNNs.

### A.4  Additional Results of Clean Accuracy Drop Analysis

In the main text, Table 3 shows that some existing methods may cause a drop in the model's clean accuracy during inference under certain conditions. This is inconsistent with the desired behavior of a backdoor attack, where only nodes containing the trigger should be misclassified as the target class, while predictions on clean nodes should remain unaffected to preserve clean accuracy. We illustrate the average clean accuracy and standard deviation of five runs of different backbones after being backdoored in Fig. 7. It should be noted that **Avg** represents the average results on GCN, GAT, and GIN, which are three target models we use in Tab. 3. From the figure, we can observe that:

- Nearly all methods show a decrease in clean accuracy, indicating that their backdoor attack process damages the normal behavior of the model, thereby weakening its practicality.

- Combined with the results of Tab. 3, certain baselines (e.g., EBA-C and ECGBA) severely degrade the clean accuracy of target models, which violates the backdoor attack's objective to maintain the classification accuracy of the model for clean test nodes.

- The degradation of clean accuracy caused by various methods is generally alleviated on the stronger defense models, RobustGCN and GNNGuard, which indicates the robustness of our selected defense methods and highlights the effectiveness of Ba-Logic in conducting backdoor attacks.

Furthermore, we extend the clean accuracy analysis to graph classification and edge prediction tasks. Tab. 6 reports the ASR and clean accuracy of Ba-Logic and baselines on these tasks. We observe that Ba-Logic consistently achieves the highest ASR without degrading clean accuracy across all tasks, demonstrating that the logic poisoning strategy of Ba-Logic does not compromise the model's normal performance on diverse GNN downstream tasks.

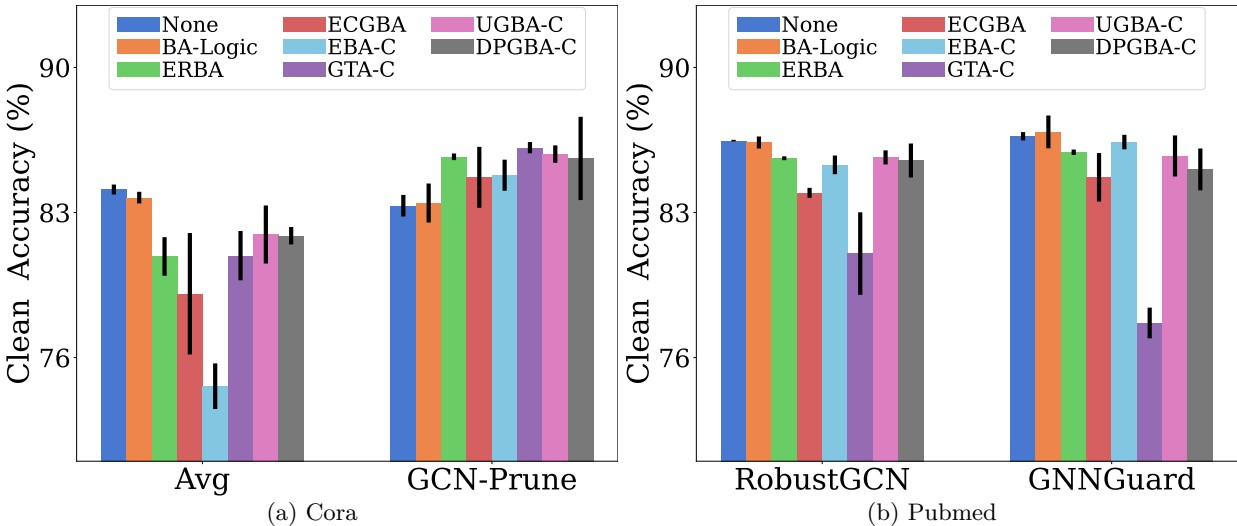

Figure 7: Clean accuracy of backdoored models.

## A.5 Additional Comparison with Cross-Domain Baselines

In the main text of our work, we mainly focus on evaluating backdoor attacks in the graph domain. However, we found that the clean-label setting presents a shared challenge for both image and graph domains. And the representative works from the image domain solve unique challenges and make significant contributions in their respective fields. Specifically, we adopt the following backdoor methods from the image domain to the graph domain:

- **OPS-GFS** Guo et al. (2023): OPS-GFS presents a clean-label video backdoor attack, designing a temporal chrominance trigger to achieve imperceptible yet effective poisoning. This work proposes a temporal chrominance-based trigger, leveraging the peculiarities of the human visual system to reduce trigger visibility. To achieve effective poisoning in clean-label setting, the method utilizes an Outlier Poisoning Strategy (OPS). OPS selects poisoned video samples that the surrogate video model cannot classify. The attack is further enhanced by Ground-truth Feature Suppression (GFS), which suppresses the features of outlier samples.
- **UAT** Zhao et al. (2020): UAT proposes a clean-label video backdoor attack, employing a universal adversarial trigger to overcome high-dimensional input and unique clean-label challenges in video recognition. The method employs a Universal Adversarial Trigger (UAT), whose pattern is generated by minimizing the cross-entropy loss towards the target class across video samples from non-target classes. To enhance the trigger, the attack applies adversarial perturbation to videos in the target class before injecting UAT. UAT coordinates two types of perturbation, uniform and targeted adversarial perturbation, to weaken original features.

To adapt OPS-GFS to the graph domain, we make the following efforts:

- To adapt OPS strategy, we select the misclassified training graphs from the target class as poison samples, and leverage GNNExplainer, an interpretability method, to find the top-30% unimportant nodes for classification.
- To adapt GFS strategy, we add the cosine modulation to the unimportant node feature and use an amplitude $\Delta$ to control its effect. For the rest node features, we add PGD-based adversarial perturbation for suppression.

To adapt UAT to the graph domain, we make the following efforts:

- To align with the original work, we leverage GNNExplainer to find the unimportant nodes for classifying the training graphs from non-target classes. We obtain a trigger pattern by randomly initializing the node features while masking off the other nodes in the graph.
- We denote uniform/targeted adversarial perturbation as **-U** and **-T**, respectively. During testing, we randomly select non-target samples and replace their node features with trigger pattern.

We evaluate the adapted methods and our Ba-Logic across various graph classification datasets, including MUTAG, PROTEINS, and NCI1 of Morris et al. (2020), and record the results in the following table.

Table 8: Results (ASR|clean accuracy(%)) of comparing Ba-Logic with baselines from image domain.

| Datasets | OPS-GFS | UAT-U | UAT-T | Ba-Logic |
|---|---|---|---|---|
| MUTAG | 88.45\|60.67 | 88.91\|60.95 | 90.15\|59.45 | 92.17\|61.32 |
| PROTEINS | 71.96\|70.14 | 85.73\|65.36 | 86.43\|68.18 | 89.67\|71.65 |
| NCI1 | 83.71\|80.65 | 93.46\|77.44 | 95.17\|78.16 | 94.38\|80.39 |

From Tab. 8, we have the following key findings:

- The adapted methods demonstrate comparable performance in the graph domain, indicating the effectiveness and transferability of their frameworks.
- OPS-GFS uses cosine modulation for the trigger pattern, associated with period $T$ of time series data. While we grid-searched its hyperparameters, lack of knowledge in static graphs potentially limited its performance, especially with multi-class classification on PROTEINS.
- UAT achieves high ASR with both perturbing variants while suffering from slight clean accuracy degradation, which is consistent with reports in its original paper.
- UAT can slightly outperform OPS-GFS. We find that UAT can fully use all non-target class samples, while OPS-GFS selects poison samples from a single non-target class due to its original design of binary classification.

Moreover, the additional experiment represents an early adaptation of clean-label backdoors from the image to the graph domain, and we hope the effort can strengthen our contributions.

## A.6 Additional Results of Attacking against Defending Strategies

In the main text of our work, we evaluate Ba-Logic with its ASR when facing defense methods. Due to the space limitation of the main text, we compared our method with three leading competitors against four defense methods on two datasets. In this subsection, we first present the complete comparison with all baselines we select in the main text of our work. We evaluate the ASR of these attack methods against defense models on four datasets, and record the results in Tab. 9. From the table, we obtain similar observations to those in Tab. 5. Compared to competitors, Ba-Logic still shows significantly higher ASR. Notably, the competitors also achieve better ASR on **Cora** and **Pubmed** than the records for larger graph **Arxiv** in Tab. 5, likely due to the graphs being smaller, which aids the attack methods in generalizing trigger patterns.

We further analyze why the existing defense methods in Tab. 9 fall short when facing Ba-Logic. Specifically:

- **GCN-Prune** removes edges between nodes with dissimilar features. Our logic poisoning triggers are generated with the constraint of an unnoticeable limit, which enables our triggers to bypass the defense.
- **RobustGCN** models hidden states of nodes as Gaussian distributions to unweight noisy features and absorb adversarial modifications. Our method explicitly guides the model's inner prediction logic to emphasize the importance of our trigger, instead of identifying our triggers as adversarial.
- **GNNGuard** unweights edges link nodes with low similarity in representation space, effectively acting as an attention-based defense. Our triggers poison the logic of GNNs to be identified as important for prediction, thus forcing GNNGuard to focus on triggers instead of unweighting them.

Table 9: Results (ASR(%)) of comparing Ba-Logic with baselines against defense models.

| Datasets | Defense | ERBA | ECGBA | EBA-C | GTA-C | DPGBA-C | UGBA-C | Ba-Logic |
|----------|---------|------|-------|-------|-------|---------|--------|----------|
| Cora | GCN-Prune | 5.93 | 15.56 | 16.71 | 15.69 | 33.10 | 52.07 | **99.17** |
| | RobustGCN | 4.17 | 14.25 | 15.28 | 15.04 | 21.78 | 56.09 | **99.12** |
| | GNNGuard | 0.00 | 0.01 | 0.14 | 0.00 | 50.27 | 35.57 | **98.48** |
| | RIGBD | 0.00 | 0.01 | 0.00 | 0.00 | 0.07 | 0.16 | **95.47** |
| Pubmed | GCN-Prune | 4.13 | 15.79 | 19.25 | 18.95 | 37.34 | 59.64 | **97.95** |
| | RobustGCN | 2.96 | 13.24 | 17.13 | 18.72 | 28.13 | 45.59 | **98.10** |
| | GNNGuard | 0.00 | 0.51 | 0.00 | 1.39 | 41.07 | 40.58 | **95.46** |
| | RIGBD | 0.00 | 0.01 | 0.00 | 0.00 | 0.01 | 0.09 | **93.01** |
| Flickr | GCN-Prune | 0.00 | 15.02 | 16.09 | 17.19 | 35.44 | 54.49 | **99.24** |
| | RobustGCN | 0.00 | 11.25 | 12.27 | 15.51 | 22.26 | 58.81 | **99.02** |
| | GNNGuard | 0.00 | 0.01 | 0.00 | 0.00 | 53.41 | 33.14 | **99.36** |
| | RIGBD | 0.00 | 0.01 | 0.00 | 0.00 | 0.07 | 0.33 | **94.86** |
| Arxiv | GCN-Prune | 0.00 | 13.57 | 21.07 | 16.02 | 17.34 | 62.31 | **96.75** |
| | RobustGCN | 0.00 | 14.24 | 17.83 | 13.29 | 29.65 | 44.46 | **97.03** |
| | GNNGuard | 0.00 | 0.51 | 0.00 | 0.00 | 43.77 | 40.08 | **95.37** |
| | RIGBD | 0.00 | 0.01 | 0.00 | 0.00 | 0.01 | 0.19 | **93.23** |

- **RIGBD** assumes poisoned nodes exhibit high prediction variance, as random edge dropping can remove triggers and change predictions of poisoned nodes back to the original class. Our method adopts a clean-label setting, where the poisoned nodes are originally labeled as the target class without requiring any label alteration. Therefore, removing triggers does not significantly change predictions, causing RIGBD to fail in identifying triggers.

In our work, we employ the black-box threat model, where defense methods can be deployed against unseen target models to counter adversaries. While we note that our method can surpass various defending strategies, including the latest SOTA defense method RIGBD, it is also important to note that the defense methods in our work employ a strong defense goal, which is cleansing the poisoned graph and degrading ASR. The defense goal is reasonable in a real-world scenario, as achieving a weaker defense goal, such as detection, would also naturally lead to the removal of injected triggers.

Meanwhile, we also note that a straightforward and widely adopted defense method is of cleansing graphs by removing edges with unusually high node degrees Dai et al. (2023); Dhali & Dividino (2024). To further demonstrate the robustness of our method, we prune $\{1, 2, 3\}$ edges from nodes with top-5% and top-10% degree after the trigger injection. We propose a metric, Remaining Trigger Connectivity (RTC), defined as the ratio of the number of edges connected to the trigger after pruning to the number before pruning. We evaluate Ba-Logic under this pruning defense strategy with RTC(%) and ASR | clean accuracy (%) on **Arxiv**, and record the results in Tab. 10.

Table 10: Results of Ba-Logic against pruning defenses.

| Setting | Pruning top-5% | | Pruning top-10% | |
|---------|------|---------------------|------|---------------------|
| | RTC | ASR \| clean accuracy | RTC | ASR \| clean accuracy |
| Prune 1 edge | 96.20 | 97.45 \| 60.42 | 95.80 | 97.71 \| 61.15 |
| Prune 2 edges | 94.80 | 97.25 \| 59.37 | 91.60 | 96.82 \| 58.72 |
| Prune 3 edges | 90.40 | 96.62 \| 56.31 | 88.70 | 94.75 \| 56.03 |

From Tab. 10, we obtain the following key findings:

- Our approach achieves outstanding performance against this pruning defense method. We owe this to our node selection being of an uncertainty-based rather than a degree-based nature.

- Pruning can defend against backdoor attacks partially, but compromise the clean accuracy of GNN. This is because the optimization of Ba-Logic is regulated by an unnoticeable constraint in Eq.(8), which ensures the injected trigger maintains high cosine similarity with normal samples.

## A.7 Additional Results of Module Contribution Analysis

### A.7.1 Hyperparameter Analysis

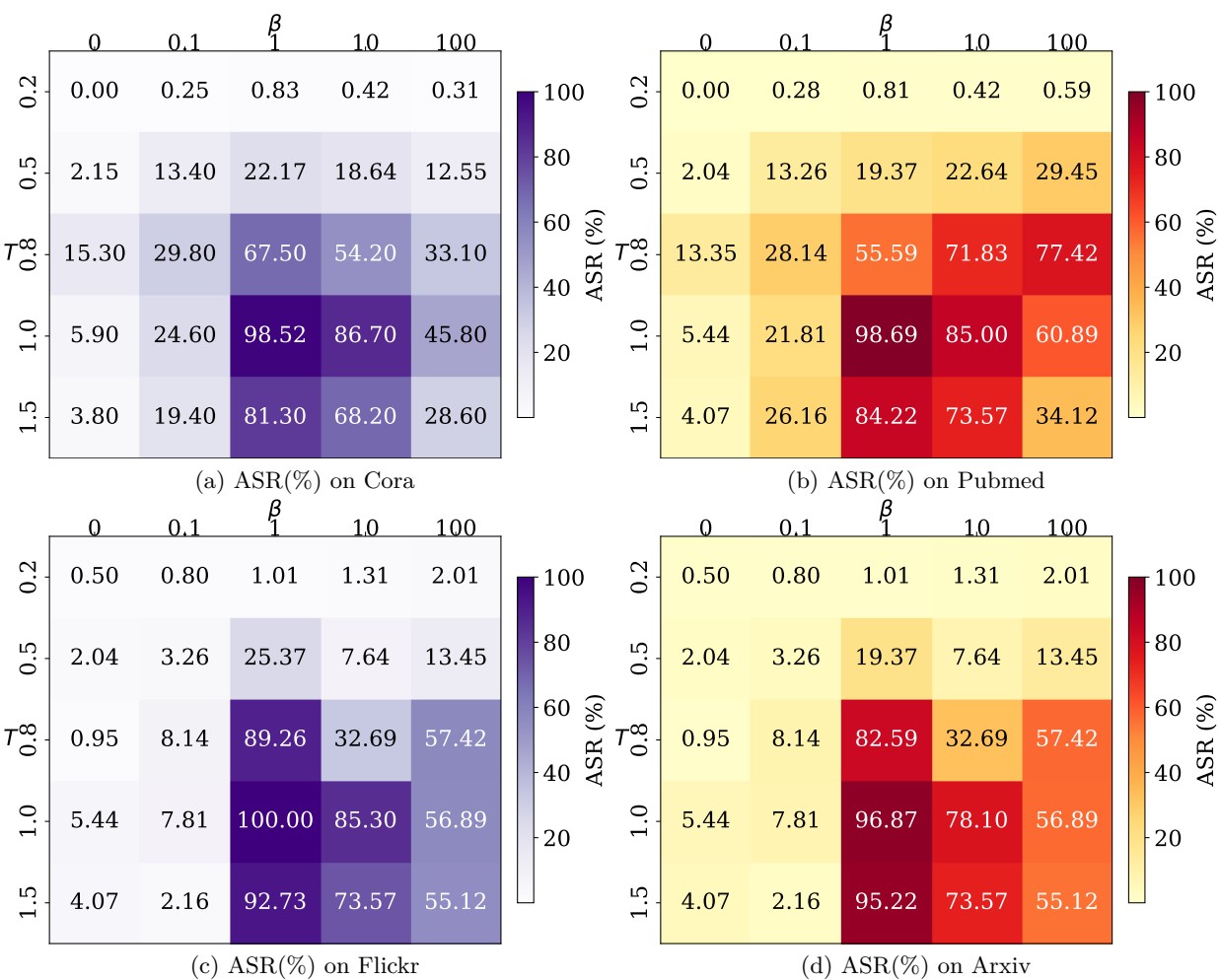

Figure 8: Hyperparameter sensitivity analysis of Ba-Logic.

In the main text of our work, we introduce hyperparameters to regulate the magnitude of the logic poisoning loss during optimization. To thoroughly understand their impact, we further investigate how the main hyperparameters, i.e., $T$ in Eq.(7) and $\beta$ in Eq.(10), affect the performance of Ba-Logic. Specifically, $T$ controls the expected margin of importance scores between trigger nodes and clean neighbor nodes. And $\beta$ controls the weight of the logic poisoning loss. To explore the effects of $T$ and $\beta$, we take the value of $T$ and $\beta$ corresponding to the experimental result of Tab. 3 as the normalized 1 and conduct parameter sweeps. Specifically, we vary $T$ as $\{0.2, 0.5, 0.8, 1.0, 1.5\}$. And $\beta$ is changed from $\{0, 0.1, 1, 10, 100\}$. We report the ASR of attacking a 2-layer GCN in Fig. 8, from which we observe: **(i) Arxiv** requires larger trigger margins $T$ than **Pubmed**, due to its higher average node degree, where clean neighbor nodes exert stronger influence on predictions. Hence, higher margins $T$ and weights $\beta$ are necessary to avoid attack failure. In practice, the value of $T$ is often set to the local maximum of the trigger nodes' gradients to ensure attack effectiveness. **(ii)** ASR degrades when $T$ and $\beta$ are overly high. This is because unsuitable large values of $T$ and $\beta$ could hinder the optimization of Ba-Logic.

### A.7.2   Cross-Method Module Utility Analysis

In the main text of our work, we have emphasized that a challenge of clean-label graph backdoor attack is that trigger-attached poisoned samples are correctly labeled as the target class. Thus, the injected triggers would be treated as irrelevant information in prediction, resulting in poor backdoor performance.

To address the challenge, we deliberately select the nodes with high uncertainty because:

- These nodes exhibit irregular patterns that are weakly associated with the target class $y_t$
- The triggers obtained by the generator exhibit consistent patterns to poison the prediction logic, causing the model to shift focus from irregular patterns to treating these triggers as key features

Specifically, we design an uncertainty metric based on two aspects: **(i)** the probability of being predicted as the target class is low, and **(ii)** the node is also uncertain for other classes.

As stated in our work, the main purpose of poison node selection is to more efficiently use the attack budget, and our main contribution lies in inner logic poisoning rather than poison node selection. Poisoned node selection serves solely as positions for trigger injection. After injecting triggers, the target model trained on the backdoored graph is backdoored, and any test node could be successfully attacked by attaching triggers during the inference of the target model. Indeed, we randomly selected 25% of test nodes as the target for each evaluation.

In our original comparison, we faithfully preserved each method's specific poison node selection, either designed or random. While it ensures a fair comparison, we agree that evaluating the performance of baselines when they also poison nodes that are selected by our poisoned node selector would highlight the module's utility. Hence, we update the baselines using our poison node selection under the clean-label setting and denote the updated methods with **-S**. We evaluate them on four datasets and record the average ASR|clean accuracy(%) towards three target models adopted in the main text of our work, i.e., GCN, GIN, and GAT:

Table 11: Comparison between BA-LOGIC and poisoned node selection updated baselines.

| Dataset | ERBA-S | EBA-S | ECGBA-S | GTA-S | UGBA-S | DPGBA-S | BA-Logic |
|---------|--------|-------|---------|-------|--------|---------|----------|
| Cora   | 21.81 \| 80.90 | 33.25 \| 72.62 | 56.84 \| 79.28 | 32.52 \| 80.92 | 68.94 \| 81.94 | 67.87 \| 81.88 | 98.20 \| 83.72 |
| Pubmed | 22.95 \| 85.69 | 33.99 \| 85.50 | 58.38 \| 85.03 | 41.57 \| 86.10 | 68.87 \| 85.82 | 70.31 \| 85.61 | 96.89 \| 85.79 |
| Flickr | 4.70 \| 46.29 | 34.15 \| 44.99 | 62.08 \| 44.99 | 51.71 \| 45.88 | 68.08 \| 45.66 | 71.39 \| 44.78 | 99.90 \| 45.70 |
| Arxiv  | 0.01 \| 65.43 | 31.09 \| 65.57 | 54.94 \| 64.21 | 37.98 \| 65.79 | 70.79 \| 66.08 | 69.10 \| 66.27 | 98.03 \| 66.02 |

From Tab. 11, we obtain the following key findings:

- All baselines show enhanced ASR when using our poison node selection, confirming its effectiveness
- Among them, ECGBA-S has the most significant improvement. It is because we improved its uncertainty metric, so the node should also be uncertain for other classes
- Methods that select nodes randomly only achieve limited improvement, such as ERBA and DPGBA, indicating that the trigger is paramount for effective graph backdoor

Results from this analysis demonstrate two main conclusions: **(i)** Our poisoned node selection module is effective and generalizable, as its adoption improves baseline performance. **(ii)** However, the primary source of our method's superior attack success rate is the logic poisoning mechanism with solid theoretical ground. This is evidenced by the fact that updated baselines still fail to compare with our method.

### A.8   Additional Analysis on IRT and Impact of Attack Budget

### A.8.1   Important Scores of Trigger Nodes

In our preliminary analysis, we conduct a theoretical analysis showing that the existing methods fail under clean-label settings because their triggers are deemed unimportant for prediction by the target GNN models.

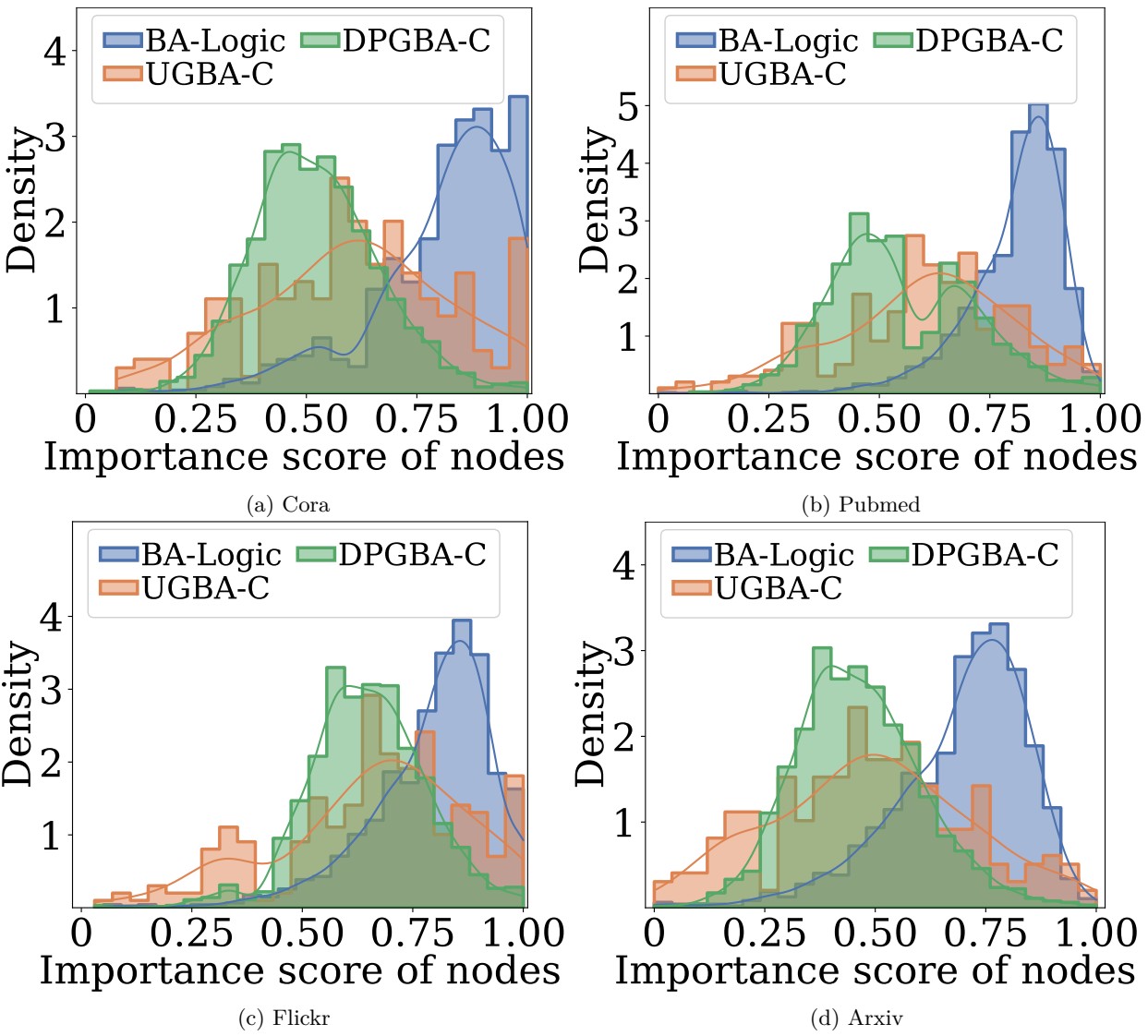

Figure 9: Comparison of IRT distribution.

We further reveal that the attack success rate is bounded by the important rate of triggers, a novel metric proposed in our preliminary analysis. However, there is still a research question waiting to be addressed: Does BA-LOGIC successfully poison the target GNNs' prediction logic as designed? To answer this question, we employ GNNExplainer to measure trigger importance scores distribution for poisoned nodes, comparing BA-LOGIC against two leading baselines, UGBA-C and DPGBA-C.

The histograms of the normalized importance scores across four datasets, **Cora**, **Pubmed**, **Flickr**, and **Arxiv**, are presented in Fig. 9. For each dataset, we report the IRT distributions averaged over the three clean models in Tab. 3, i.e., GCN, GIN, and GAT. From these figures, we have the following key observations:

- **Ba-Logic** shows concentration of nodes with large IRT values, with peaks close to the maximal importance score. This indicates that the logic poisoning triggers are identified as important by the logic of backdoored GNNs.

- **UGBA-C and DPGBA-C** exhibit flatter IRT distributions, with most mass in the lower importance range. This indicates that their triggers are less effective at poisoning the inner logic of the target models.

- Across all four datasets and three clean models, methods with higher IRT consistently achieve higher ASR. This aligns with theoretical analysis, which indicates that existing graph backdoor methods generally yield a low important rate of triggers, resulting in poor attack performance in the clean-label setting.

### A.8.2 Impact of Attack Budget

(a) GCN

(b) GAT

Figure 10: Impact of attack budget.

In our preliminary analysis, we evaluate BA-LOGIC and competitors by varying the size of poisoned nodes $\mathcal{V}_P$, which is also the attack budget of backdoor attack. Here, we further explore the attack performance with various attack budgets. Specifically, we vary the size of $\mathcal{V}_P$ as $\{80, 160, 240, 320, 400, 480\}$, and record the results on **Arxiv** with GCN and GAT in Fig. 10, from which we observe: **(i)** ASR of most competitors increases with the increase of $\mathcal{V}_P$, which intuitively satisfies expectations. BA-LOGIC consistently outperforms the baselines regardless the size of $\mathcal{V}_P$, showing its effectiveness. Notably, the gaps between our method and baselines widen as the budget is smaller, demonstrating the effectiveness of the poisoned node selection in effectively utilizing the attack budget. **(ii)** Compared to other competitors, the ASR of BA-LOGIC remains stable across GNN models with distinct inner logic, showing BA-LOGIC's transferability.

We further investigate the impact of a smaller attack budget. Specifically, we conducted additional evaluations of EBA-C, UGBA-C, and DPGBA-C with attack budgets ranging from $\{10, 20, 30, 40, 50\}$. We evaluate these methods on **Cora** and **Pubmed**, and record ASR | clean accuracy(%) as below:

Table 12: Results (ASR | clean accuracy(%)) on **Cora** and **Pubmed** with less attack budget.

| | Cora | | | | Pubmed | | | |
|---|---|---|---|---|---|---|---|---|
| $\Delta_P$ | **EBA-C** | **UGBA-C** | **DPGBA-C** | **BA-Logic** | **EBA-C** | **UGBA-C** | **DPGBA-C** | **BA-Logic** |
| 10 | 5.13 \| 82.47 | 15.29 \| 81.05 | 17.87 \| 80.95 | 65.16 \| 83.11 | 0.69 \| 86.38 | 11.89 \| 85.36 | 21.15 \| 84.95 | 62.07 \| 85.32 |
| 20 | 12.39 \| 82.78 | 51.26 \| 80.14 | 45.43 \| 81.36 | 87.13 \| 83.06 | 8.61 \| 86.21 | 34.62 \| 85.36 | 49.31 \| 84.36 | 88.04 \| 85.09 |
| 30 | 15.64 \| 82.34 | 62.20 \| 80.65 | 50.77 \| 80.44 | 92.17 \| 82.39 | 19.47 \| 86.75 | 64.16 \| 85.05 | 54.61 \| 85.44 | 94.79 \| 84.67 |
| 40 | 29.03 \| 81.86 | 63.18 \| 80.37 | 51.03 \| 80.85 | 94.22 \| 83.16 | 29.76 \| 85.33 | 64.07 \| 85.16 | 55.57 \| 85.85 | 95.44 \| 84.96 |
| 50 | 29.17 \| 80.99 | 63.23 \| 79.85 | 51.15 \| 80.75 | 96.05 \| 81.85 | 30.66 \| 85.17 | 63.97 \| 85.25 | 57.86 \| 85.75 | 95.71 \| 85.05 |

From Tab. 12, we have the following key findings:

- Our method achieves leading performance against all baselines, and all methods demonstrate slightly better clean accuracy with less attack budget.

- The relationship between ASR and attack budget is nonlinear, and ASR tends to show more obvious improvements when the attack budget increases from a smaller value.

### A.9 Additional Analysis on Unnoticeable Constraints

In the main text of our work, we use cosine similarity for the unnoticeable constraint due to its interpretability and effectiveness in measuring local feature consistency.

To further evaluate the unnoticeability of Ba-Logic, we introduce three additional constraints from feature-distribution, feature-magnitude, and edge-connection perspectives.

**Feature Distribution.** To avoid triggers exhibiting abnormal feature-level distributional patterns, we constrain the trigger nodes to match the local feature distribution of the target node:

$$\min_{\theta_g} \mathcal{L}_D = \sum_{v_i \in V} \Delta_{\text{feat}} \left( \{\mathbf{x}_g \mid v_g \in g_i\}, \{\mathbf{x}_c \mid v_c \in \mathcal{N}_i\} \right), \tag{17}$$

where $\mathcal{N}_i = \mathcal{N}(v_i) \cup \{v_i\}$ denotes the local neighborhood of $v_i$. $\Delta_{\text{feat}}(\cdot, \cdot)$ measures the distribution discrepancy between the generated trigger node features and the local clean node features, which is calculated by the squared Maximum Mean Discrepancy (MMD). Different from cosine similarity, this constraint compares two sets of node features at the distribution level. We denote this variant as **Ba-Logic-D**.

**Feature Magnitude.** To avoid triggers relying on abnormal feature scales, we constrain the trigger nodes to match the local feature magnitude of the target node:

$$\min_{\theta_g} \mathcal{L}_M = \sum_{v_i \in V} \sum_{v_g \in g_i} \left( \|\mathbf{x}_g\|_2 - \bar{m}_i \right)^2, \quad \bar{m}_i = \frac{1}{|\mathcal{N}_i|} \sum_{v_c \in \mathcal{N}_i} \|\mathbf{x}_c\|_2, \tag{18}$$

where $\mathcal{N}_i = \mathcal{N}(v_i) \cup \{v_i\}$ denotes the local neighborhood of $v_i$. We denote this variant as **Ba-Logic-M**.

**Edge Connection.** Since edge-based defenses can remove edges connecting dissimilar nodes, we constrain trigger-related edges to follow the local edge-similarity pattern:

$$\min_{\theta_g} \mathcal{L}_C = \sum_{v_i \in V} \sum_{(v_j, v_k) \in E_B^i} \max \left( 0, \bar{s}_i - \text{sim}(v_j, v_k) \right)^2, \quad \bar{s}_i = \frac{1}{|E_i|} \sum_{(v_p, v_q) \in E_i} \text{sim}(v_p, v_q), \tag{19}$$

where $E_i$ denotes the edge set in the local neighborhood of $v_i$, and $E_B^i$ follows Eq. (8), containing edges inside trigger $g_i$ and edges attaching $g_i$ to node $v_i$. We denote this variant as **Ba-Logic-E**.

We evaluate Ba-Logic-D, Ba-Logic-M, and Ba-Logic-E against four existing defenses and four adaptive defenses. The results on Pubmed and Arxiv are reported in Tab. 13 and Tab. 14, respectively.

Table 13: Results (ASR|clean accuracy(%)) of different unnoticeable constraints against various defenses on Pubmed.

| Methods | GCN-Prune | RobustGCN | GNNGuard | RIGBD | ER | GM | CD | SAM |
|---|---|---|---|---|---|---|---|---|
| Ba-Logic | 97.95\|85.06 | 98.10\|84.83 | 95.46\|84.54 | 93.01\|84.32 | 89.52\|67.22 | 88.07\|66.48 | 83.28\|69.01 | 86.42\|66.23 |
| Ba-Logic-D | 96.91\|85.28 | 97.42\|85.02 | 94.68\|84.72 | 91.85\|84.54 | 87.36\|67.86 | 86.91\|67.11 | 81.74\|69.66 | 84.93\|66.91 |
| Ba-Logic-M | 97.68\|85.10 | 98.32\|84.91 | 95.18\|84.60 | 92.74\|84.35 | 88.64\|67.65 | 87.45\|66.92 | 82.71\|69.44 | 85.96\|66.71 |
| Ba-Logic-E | 98.24\|85.04 | 97.85\|84.79 | 96.02\|84.61 | 93.44\|84.26 | 88.93\|67.31 | 88.54\|66.57 | 83.06\|69.10 | 86.81\|66.36 |

Table 14: Results (ASR|clean accuracy(%)) of different unnoticeable constraints against various defenses on Arxiv.

| Methods | GCN-Prune | RobustGCN | GNNGuard | RIGBD | ER | GM | CD | SAM |
|---|---|---|---|---|---|---|---|---|
| Ba-Logic | 96.75\|63.22 | 97.03\|64.07 | 95.37\|63.31 | 93.23\|61.99 | 80.06\|61.83 | 92.36\|60.97 | 85.38\|62.34 | 81.09\|60.46 |
| Ba-Logic-D | 95.86\|63.46 | 96.31\|64.30 | 94.42\|63.55 | 92.14\|62.21 | 78.58\|62.34 | 90.84\|61.45 | 83.91\|62.79 | 79.64\|60.98 |
| Ba-Logic-M | 96.42\|63.30 | 97.21\|64.12 | 94.96\|63.39 | 92.87\|62.08 | 79.34\|62.06 | 91.72\|61.24 | 84.86\|62.55 | 80.52\|60.77 |
| Ba-Logic-E | 97.18\|63.28 | 96.78\|64.10 | 95.91\|63.42 | 93.66\|62.05 | 79.71\|61.95 | 92.74\|61.09 | 85.12\|62.43 | 81.46\|60.62 |

From Tab. 13 and Tab. 14, we have the following key findings:

- Ba-Logic-D preserves comparable ASR and clean accuracy to Ba-Logic, showing that explicitly reducing feature-distribution discrepancy does not weaken logic poisoning.

- BA-LOGIC-M and BA-LOGIC-E maintain comparable ASR and clean accuracy to BA-LOGIC, indicating that constraining feature magnitude and edge-connection similarity does not compromise the effectiveness of the attack.
- These results provide additional evidence that BA-LOGIC remains unnoticeable not only under cosine-similarity constraints and degree-pruning defenses, but also under distribution-level, magnitude-level, and edge-connection-level constraints.

### A.10 Additional Comparison with Adversarial Attack Baselines

Recent graph adversarial attacks have made notable progress in finding effective structural perturbations for GNNs. To provide a more timely and comprehensive comparison, we further compare BA-LOGIC with the following recent graph attack baselines:

- **EvA** Akhondzadeh et al. (2026): EvA proposes an evolutionary attack framework for graph-structured data. Different from gradient-based attacks that relax the discrete edge perturbation problem into a continuous optimization problem, EvA directly searches over discrete edge perturbations with evolutionary optimization. This design allows EvA to work with black-box models and non-differentiable objectives, making it a strong and flexible adversarial graph attack baseline.
- **GOttack** Alom et al. (2025): GOttack presents a universal adversarial attack framework for GNNs via graph orbits learning. It exploits topology-aware graph-orbit information to identify vulnerable structural patterns and generate effective graph perturbations. By explicitly considering local topological equivalence groups, GOttack provides a strong structure-aware attack baseline for node classification.

Although EvA and GOttack are designed for adversarial graph attacks rather than clean-label backdoor attacks, both methods represent recent advances in optimizing effective graph perturbations. Therefore, we include them as additional baselines to examine whether recent adversarial perturbation strategies can achieve competitive attack performance under our evaluation setting. We report the best ASR|clean accuracy (%) of these methods on four graph datasets in Tab. 15.

Table 15: Results (ASR|clean accuracy(%)) of comparing BA-LOGIC with recent graph adversarial attack baselines.

| Datasets | EvA | GOttack | Ba-Logic |
|---|---|---|---|
| Cora | 91.84\|82.76 | 89.57\|82.91 | **98.20\|83.72** |
| Pubmed | 94.38\|85.42 | 92.16\|85.31 | **96.89\|85.79** |
| Flickr | 86.73\|45.38 | 84.92\|45.61 | **99.90\|45.70** |
| Arxiv | 93.54\|65.74 | 91.89\|65.88 | **98.03\|66.02** |

From Tab. 15, we have the following key findings:

- EvA and GOttack are strong graph attack baselines. EvA benefits from directly optimizing discrete edge perturbations through evolutionary search, while GOttack leverages graph-orbit-aware topological information to identify effective structural perturbations.
- However, EvA and GOttack mainly optimize adversarial misclassification rather than clean-label backdoor logic poisoning. Thus, they do not explicitly encourage the victim GNN to consistently treat the injected trigger as dominant evidence for the target class.
- In contrast, BA-LOGIC consistently maintains outstanding ASR across various datasets while preserving comparable clean accuracy. This demonstrates the superiority of BA-LOGIC in poisoning the inner logic of GNNs for clean-label backdooring.

Table 16: Threat model level comparison.

| Method | Manipulates Existing Nodes/Edges | Alters Training Labels | Knows/Controls Defense for Data Cleaning | Trigger Injection | Comparison to Ours |
|---|---|---|---|---|---|
| GTA | ✗ | ✓ | ✗ | ✓ | Stronger assumption than ours |
| UGBA | ✗ | ✓ | ✗ | ✓ | Stronger assumption than ours |
| DPGBA | ✗ | ✓ | ✗ | ✓ | Stronger assumption than ours |
| TRAP | ✓ | ✓ | ✗ | ✗ | Stronger assumption than ours |
| EBA | ✓ | ✗ | ✗ | ✗ | Stronger assumption than ours |
| CGBA | ✓ | ✗ | ✗ | ✗ | Stronger assumption than ours |
| SCLBA | ✓ | ✗ | ✗ | ✗ | Stronger assumption than ours |
| GCLBA | ✓ | ✗ | ✗ | ✗ | Stronger assumption than ours |
| SNTBA | ✓ | ✓ | ✗ | ✓ | Stronger assumption than ours |
| LB | ✓ | ✓ | ✗ | ✓ | Stronger assumption than ours |
| ERBA | ✗ | ✗ | ✗ | ✓ | Same |
| ECGBA | ✗ | ✗ | ✗ | ✓ | Same |
| Our BA-Logic | ✗ | ✗ | ✗ | ✓ | – |

## B   Threat Model Level Comparison

In the main text of our work, we introduced the threat model of BA-Logic in Sec. 2.2. To highlight the fairness of the comparison between our method and the included baselines, we further present a more comprehensive threat model level comparison.

In general, our method adopts a threat model that has stricter limitations than most existing graph backdoor attacks. Specifically, we assume the attacker's capability is limited to:

- The attacker **can NOT** remove or manipulate the existing nodes and edges in the training graph.
- The attacker **can NOT** alter the labels of the training data.
- The attacker **can NOT** know the information of the target model, such as parameters or gradients.
- The attacker **can NOT** either know or control the defending by data cleaning. Moreover, we demonstrate the effectiveness of BA-Logic against defenders that prune malicious nodes and edges in Sec. 5.4.
- The attacker **can ONLY** inject a small number of triggers to the training nodes. Each trigger is subjected to strict limits on size.

For the attacker's knowledge, our threat model is in line with the commonly adopted black-box graph backdoor attacks. Specifically, we assume that:

- The attacker **can NOT** know target GNN's architecture and hyperparameters. As stated in Sec. 2.2, we adopt a strict black-box setting where the attacker can only employ a surrogate model and transfer the attack to the unseen target model for evaluation.
- The attacker **can ONLY** know partial training data of the target GNN. This widely adopted assumption is reasonable. For example, when a GNN is trained on data from Twitter, much of that data is publicly accessible and can be readily crawled by an attacker.

We further summarize a threat model comparison of our method with existing graph backdoor attacks in Table 16 to highlight the fairness of our comparison, from which we highlight that:

- Our BA-Logic does not enable training label altering, which imposes a stricter limitation and weaker assumption compared to general backdoor attacks.
- Some clean backdoor attacks obtain triggers by altering existing nodes or edges, which is less practical than trigger injection. For instance, in social or transaction networks, attackers could easily register

malicious users and build connections with other users. However, modifying the attributes or connections of existing users, which are well-preserved, is much harder to achieve in practice.

- Our BA-Logic adopts the same threat model as two recently proposed clean-label backdoor methods, i.e., ERBA and ECGBA. This alignment ensures our method is evaluated under a contemporary and practical threat model, facilitating a fair and relevant comparison.

## C  Theoretical Analysis

In the main text of this work, we conclude that the failure of existing clean-label graph backdoors stems from the target model's inability to treat the trigger as a critical factor influencing classification outcomes. To rigorously analyze the failure mechanism of existing clean-label graph backdoor methods, we conduct a theoretical analysis in Sec. 2.3. Here we provide the proof.

**Assumptions on Graphs** Following Dai et al. (2023); Zhang et al. (2025), we consider a graph $\mathcal{G}$ where (i) The node feature $\mathbf{x}_i \in \mathbb{R}^d$ is sampled from a specific feature distribution $F_{y_i}$ that depends on the node label $y_i$. (ii) Dimensional features of $\mathbf{x}_i$ are independent to each other. (iii) The magnitude of node features is bounded by a positive scalar vector $S$, i.e., $\max_{i,j} |\mathbf{x}_i(j)| \leq S$.

These assumptions are reasonable in the context of graph representation learning for the following reasons:

- **Label-correlated feature distributions (Assumption i)**: In graph-structured data, node features often exhibit strong correlations with their labels. For instance, in an academic collaboration network, researchers' publication keywords (features) naturally reflect their disciplinary domains (labels) through semantic correspondence;
- **Independence of feature dimensions (Assumption ii)**: In many real-world graph datasets, especially the high-dimensional ones, the correlations between features are typically weak or statistically insignificant.
- **Boundaries of features (Assumption iii)**: Feature magnitudes are often bounded due to practical constraints in real-world data. Moreover, common techniques such as normalization or standardization during pre-processing can effectively bound them within a certain range.

**Theorem 1.** *We consider a graph $\mathcal{G} = (\mathcal{V}, \mathcal{E}, \mathbf{X})$ follows Assumptions. Given a node $v_i$ with label $y_i$, let $deg_i$ be the degree of $v_i$, and $\gamma$ be the value of the important rate of trigger. For a node $v_i$ attached with trigger $g_i$, the probability for GNN model $f$ predict $v_i$ as target class $y_t$ is bounded by:*

$$\mathbb{P}(f(v_i) = y_t) \leq 2d \cdot \exp\left(-\frac{deg_i \cdot (1-\gamma)^2 \cdot \|\mu_{y_t} - \mu_{y_i}\|_2^2}{2d \cdot S^2}\right), \tag{20}$$

*where $d$ is the node feature dimension, $\mu_{y_t}$ and $\mu_{y_i}$ are the class centroid vectors in the feature space for $y_i$ and $y_t$, respectively.*

*Proof.* We present the pre-activated node representation as $\mathbb{E}[\mathbf{h}_i] = \mathbb{E}[\sum_{j \in \mathcal{N}(i)} \frac{1}{\sqrt{\deg_i}\sqrt{\deg_j}} \mathbf{W}\mathbf{x_j}]$. Following the Assumption (ii), $\mathbb{E}[\mathbf{h}_i]$ can be written as:

$$\begin{aligned}
\mathbb{E}[\mathbf{h}_i] &= \mathbb{E}\left[\sum_{j \in \mathcal{N}(i)} \frac{1}{\sqrt{\deg_i}\sqrt{\deg_j}} \mathbf{W}\mathbf{x}_j\right] \\
&= \sum_{v_j \in \mathcal{N}_c(i)} \frac{1}{\sqrt{\deg_i}\sqrt{\deg_j}} \mathbf{W}\mathbb{E}[\mathbf{x}_j] + \sum_{v_k \in \mathcal{N}_g(i)} \frac{1}{\sqrt{\deg_i}\sqrt{\deg_k}} \mathbf{W}\mathbb{E}[\mathbf{x}_k],
\end{aligned} \tag{21}$$

Let $\alpha_{ij} = \frac{1}{\sqrt{\deg_i}\sqrt{\deg_j}}$ denotes the normalized aggregation weight. To get a tighter bound, we assume that the clean neighbor nodes of $v_i$ are labeled as $y_i$, and the neighbor node within the trigger is already considered important by an arbitrary graph explainer for the prediction of $v_i$. Moreover, following Dai et al. (2022b); Zhang et al. (2025), we consider a regular graph $\mathcal{G}$, i.e., each node has the same number of neighbors. For a node $v_i \in \mathcal{V}_P$ and its neighbors $\mathcal{N}(i)$, after attaching trigger $g_i$ to node $v_i$, its neighbor can be divided

into clean nodes $\mathcal{N}_c(i)$ and trigger nodes $\mathcal{N}_g(i)$. Then we can present the mathematical definition of IRT as follows:

$$
\begin{aligned}
\text{IRT} &= \frac{\#\text{Trigger Nodes in Top-k Important Nodes}}{\#\text{Poisoned Nodes } |\mathcal{V}_P|} \\
&= \frac{1}{|\mathcal{V}_P|} \cdot \sum_{v_i \in \mathcal{V}_\mathcal{P}} \frac{|\mathcal{N}_g(i)|}{|\mathcal{N}_g(i)| + |\mathcal{N}_c(i)|}
\end{aligned}
\tag{22}
$$

As a fixed $\mathcal{V}_P$ makes $\frac{1}{|\mathcal{V}_P|}$ remains constant, the IRT value in our proof can be simplified as $\gamma = \sum_{v_k \in \mathcal{N}(g_i)} \alpha_{ik} = \frac{|\mathcal{N}_g(i)|}{|\mathcal{N}_g(i)| + |\mathcal{N}_c(i)|}$. Substitute $\gamma$ with Eq.(21), we have:

$$
\begin{aligned}
\mathbb{E}[\mathbf{h}_i] &= (1-\gamma)\mathbf{W}\mathbb{E}_{\mathbf{x} \sim F_{y_i}}[\mathbf{x}] + \gamma\mathbf{W}\mathbb{E}_{\mathbf{x} \sim F_{y_t}}[\mathbf{x}] \\
&= (1-\gamma)\mathbf{W}\mu_{y_i} + \gamma\mathbf{W}\mu_{y_t}
\end{aligned}
\tag{23}
$$

Let $\tilde{\mu}_y = \mathbf{W}\mu_y$ denote the class centroid feature vector in the embedding space via the linear mapping by a GNN model's weight matrix $\mathbf{W}$. To get a bound for the distance between $\mathbb{E}[\mathbf{h}_i]$ and $\tilde{\mu}_{y_t}$ in the embedding space, we substitute Eq.(23) with the triangle inequality and have:

$$
\begin{aligned}
\|\mathbb{E}[\mathbf{h}_i] - \tilde{\mu}_{y_t}\|_2 &= \|\mathbb{E}[\mathbf{h}_i] - \tilde{\mu}_{y_i} + \tilde{\mu}_{y_i} - \tilde{\mu}_{y_t}\|_2 \\
&\geq \|\tilde{\mu}_{y_i} - \tilde{\mu}_{y_t}\|_2 - \|\mathbb{E}[\mathbf{h}_i] - \tilde{\mu}_{y_i}\|_2 \\
&= \|\tilde{\mu}_{y_i} - \tilde{\mu}_{y_t}\|_2 - \gamma\|\tilde{\mu}_{y_t} - \tilde{\mu}_{y_i}\|_2 \\
&\geq (1-\gamma) \cdot \|\tilde{\mu}_{y_t} - \tilde{\mu}_{y_i}\|_2
\end{aligned}
\tag{24}
$$

Following Wang & Shen (2024), we consider that the decision boundary in the embedding space for an arbitrary GNN model to predict node $v_i$ as $y_i$ is $\|\mathbf{h}_i - \tilde{\mu}_{y_i}\|_2 < \|\mathbf{h}_i - \tilde{\mu}_{y_j}\|_2, \forall y_i \neq y_j$. For a successful backdoor attack, there must have a small $\epsilon > 0$ such that $\|\mathbf{h}_i - \tilde{\mu}_{y_t}\|_2 < \epsilon < \|\mathbf{h}_i - \tilde{\mu}_{y_i}\|_2$. Substitute the equation with triangle inequality, we have:

$$
\begin{aligned}
\|\mathbf{h}_i - \mathbb{E}[\mathbf{h}_i]\|_2 &\geq \|\mathbb{E}[\mathbf{h}_i] - \tilde{\mu}_{y_t}\|_2 - \|\mathbf{h}_i - \tilde{\mu}_{y_t}\|_2 \\
&\geq \|\mathbb{E}[\mathbf{h}_i] - \tilde{\mu}_{y_t}\|_2 - \epsilon
\end{aligned}
\tag{25}
$$

which indicates the successful backdoor attack is included in the bounds for $\mathbf{h}_i$ deviates from its expectation $\mathbb{E}[\mathbf{h}_i]$.

To continue the proof, we then introduce the celebrated Hoeffding's Inequality:

**Lemma 1. (Hoeffding's Inequality).** *Let $X_1, \ldots, X_n$ be independent bounded random variables with $X_i \in [a, b]$ for all $i$, where $-\infty < a \leq b < \infty$. Then*

$$
\mathbb{P}\left(\frac{1}{n}\sum_{i=1}^n (X_i - \mathbb{E}[X_i]) \geq t\right) \leq \exp\left(-\frac{2nt^2}{(b-a)^2}\right)
\tag{26}
$$

*and*

$$
\mathbb{P}\left(\frac{1}{n}\sum_{i=1}^n (X_i - \mathbb{E}[X_i]) \leq -t\right) \leq \exp\left(-\frac{2nt^2}{(b-a)^2}\right)
\tag{27}
$$

*holds for all $t \geq 0$.*

For each feature dimension $j \in \{1, ..., d\}$, the node embedding $\mathbf{h}_i$ can be decomposed as $\mathbf{h}_i[j] = \sum_{k \in \mathcal{N}(i)} \alpha_{ik}\mathbf{W}\mathbf{x}_k[j]$. For any dimension $j$, $\mathbf{x}_k[j]$ is independent and bounded by $[-S, S]$. Hence, directly use Hoeffding's inequality, for any $t_1 > 0$ and a fixed dimension $j$, we have:

$$
\begin{aligned}
\mathbb{P}\left(|\mathbf{h}_i(j) - \mathbb{E}[\mathbf{h}_i(j)]| \geq t_1\right) &\leq 2\exp\left(-\frac{2t_1^2}{\sum_k (2\alpha_{ik}\mathbf{W}S)^2}\right) \\
&\leq 2\exp\left(-\frac{\deg_i \cdot t_1^2}{2\rho(\mathbf{W})^2 S^2}\right),
\end{aligned}
\tag{28}
$$

where $\rho^2(\mathbf{W})$ denotes the largest singular value of $\mathbf{W}$. By applying union bound over all $d$ dimension, we extend Eq.(28) as:

$$
\begin{aligned}
\mathbb{P}\left(\|\mathbf{h}_i - \mathbb{E}[\mathbf{h}_i]\|_2 \geq t_1\sqrt{d}\right) &\leq \mathbb{P}\left(\bigcup_{j=1}^d \{|\mathbf{h}_i(j) - \mathbb{E}[\mathbf{h}_i(j)]| \geq t_1\}\right) \\
&\leq \sum_{j=1}^d \mathbb{P}\left(|\mathbf{h}_i(j) - \mathbb{E}[\mathbf{h}_i(j)]| \geq t_1\right) \\
&\leq 2d\exp\left(-\frac{t_1^2}{2\rho(\mathbf{W})^2 S^2 \cdot \frac{1}{\deg_i}}\right) \\
&= 2d\exp\left(-\frac{\deg_i t_1^2}{2\rho(\mathbf{W})^2 S^2}\right)
\end{aligned}
\tag{29}
$$

Let $t_2 = t_1 \cdot \sqrt{d} = (1-\gamma) \cdot \|\tilde{\mu}_{y_t} - \tilde{\mu}_{y_i}\|_2$, then we have:

$$
\begin{aligned}
\mathbb{P}\left(\|\mathbf{h}_i - \mathbb{E}[\mathbf{h}_i]\|_2 \geq t_2\right) &\leq 2d\exp\left(-\frac{\deg_i t_1^2}{2\rho(\mathbf{W})^2 S^2 d}\right) \\
&= 2d\exp\left(-\frac{\deg_i \cdot (1-\gamma)^2 \cdot \|\tilde{\mu}_{y_t} - \tilde{\mu}_{y_i}\|_2^2}{2\rho(\mathbf{W})^2 S^2}\right)
\end{aligned}
\tag{30}
$$

Substitute Eq.(30) with the upper bound of probability derived from Eq.(25), we denote $\rho(\mathbf{W}) = \|\mathbf{W}\|_2$ to present the matrix 2-norm of $\mathbf{W}$, then we have:

$$
\begin{aligned}
\mathbb{P}(f(v_i) = y_t) &\leq \mathbb{P}\left(\|\mathbf{h}_i - \mathbb{E}[\mathbf{h}_i]\|_2 \geq t\right) \\
&\leq 2d\exp\left(-\frac{\deg_i\|\mathbf{W}\mu_{y_t} - \mathbf{W}\mu_{y_i}\|_2^2}{2\rho(\mathbf{W})^2 S^2 d}\right) \\
&\leq 2d\exp\left(-\frac{\deg_i\|\mathbf{W}\|_2\|\mu_{y_t} - \mu_{y_i}\|_2^2}{2\rho(\mathbf{W})^2 S^2 d}\right) \\
&= 2d\exp\left(-\frac{\deg_i\|\mu_{y_t} - \mu_{y_i}\|_2^2}{2S^2 d}\right),
\end{aligned}
\tag{31}
$$

which completes the proof. $\qquad\qquad\qquad\qquad\qquad\qquad\qquad\qquad\qquad\qquad\qquad\qquad\qquad\qquad\quad$ $\square$

## D  Time Complexity Analysis

In BA-LOGIC, the time complexity mainly comes from the logic poisoning sample selection and the bi-level optimization of the logic poisoning trigger generator. Let $h$ denote the embedding dimension. The cost of the logic poisoning node selection can be represented approximately as $O(Mdh|\mathcal{V}|)$, where $d$ is the average degree of nodes and $M$ is the number of training iterations for the pre-trained GCN model, which is small. The cost of bi-level optimization consists of updating the weight of the surrogate GNN model in inner iterations and updating the logic poisoning trigger generator in outer iterations. The cost for updating the surrogate model is approximately $O(Ndh|\mathcal{V}_P|)$, where $d$ is the average degree of nodes and $N$ is the number of inner training iterations for the surrogate GNN model. For the trigger generator, the classification loss and prediction logic poisoning loss are computed with a cost of $O(2dh|\mathcal{V}|)$. For the unnoticeable loss $\mathcal{L}_U$, its time complexity is $O(hd|\mathcal{V}_p|\Delta_g)$. Hence, the overall time complexity of each iteration of bi-level optimization is $O(dh(2|\mathcal{V}| + (\Delta_g + N)|\mathcal{V}_P|))$, which is linear to the size of the graph. Hence, BA-LOGIC can efficiently poison the inner prediction logic of target models for clean-label graph backdoor attacks.

## E  Training Algorithm

---

**Algorithm 1** Algorithm of BA-LOGIC

---

**Require:** $\mathcal{G} = (\mathcal{V}, \mathcal{E}, \mathbf{X})$, $\mathcal{Y}_L$, $\beta$, $T$.
**Ensure:** Backdoored graph $\mathcal{G}_B$, trained trigger generator $f_g$
 1: Initialize $\theta$ and $\theta_g$ for surrogate model $f$ and logic poisoning trigger generator $f_g$, respectively;
 2: Select logic poisoning nodes set $\mathcal{V}_P$ based on Eq.(4) ;
 3: **while** not converged yet **do**
 4:    **for** $t = 1, 2, \ldots, N$ **do**
 5:       Update $\theta$ by descent on $\nabla_\theta \mathcal{L}_f$ based on Eq.(11) ;
 6:    **end for**
 7:    Update $\theta_g$ by descent on $\nabla_{\theta_g}(\sum l(\cdot) + \mathcal{L}_U + \beta \mathcal{L}_A)$ based on Eq.(12);
 8: **end while**
 9: **for** $v_i \in \mathcal{V}_P$ **do**
10:    Generate the trigger $g_i$ for $v_i$ by using $f_g$;
11:    Update $\mathcal{G}_B$ based on $a(\mathcal{G}_B^i, g_i)$;
12: **end for**
13: **return** $\mathcal{G}_B$, and $f_g$;

---

We formalize the training algorithm of BA-LOGIC in Algorithm 1. In line 2, we first select the poisoned nodes $\mathcal{V}_P$ with the top-$\Delta_P$ highest scores calculated by Eq.(4). From line 3 to line 9, we train the trigger generator $f_g$ by solving a bi-level optimization problem based on Eq.(10). In detail, we update the lower-level optimization (line 5) to poison the target model's inner logic and the outer-level optimization (line 7) to update trigger generator $f_g$, respectively. These goals are achieved by doing gradient descent on $\theta$ and $\theta_g$ based on Eq.(11) and Eq.(12). From line 10 to line 13, we use the well-trained $f_g$ to generate triggers for each poisoned node $v_i \in \mathcal{V}_P$ and update $\mathcal{G}$ to obtain the backdoored graph $\mathcal{G}_B$.

After presenting the training algorithm of BA-LOGIC, we analyze the time complexity of BA-LOGIC in Appendix D.

# F   Implementation Details

Table 17: The statistics of datasets in our work.

| Dataset | #Nodes | #Edges | #Graphs | #Features | #Classes |
|---|---|---|---|---|---|
| Cora | 2,708 | 5,429 | 1 | 1,433 | 7 |
| Pubmed | 19,717 | 44,338 | 1 | 500 | 3 |
| Flickr | 89,250 | 899,756 | 1 | 500 | 7 |
| Arxiv | 169,343 | 1,166,243 | 1 | 128 | 40 |
| CS | 18,333 | 163,788 | 1 | 6,805 | 15 |
| Physics | 34,493 | 495,924 | 1 | 8,415 | 5 |
| Squirrel | 5,201 | 217,073 | 1 | 2,089 | 5 |
| Chameleon | 2,277 | 36,101 | 1 | 2,325 | 5 |
| Penn | 41,554 | 1,362,229 | 1 | 5 | 2 |
| Genius | 421,961 | 984,979 | 1 | 12 | 2 |
| Products | 2,449,029 | 61,859,140 | 1 | 100 | 47 |
| MUTAG | ~17.9 | ~39.6 | 188 | 7 | 2 |
| NCI1 | ~29.8 | ~32.3 | 4110 | - | 2 |
| PROTEINS | ~39.1 | ~145.6 | 600 | 3 | 6 |

### F.1 Datasets Statistics

In the main text of our work, we evaluate our methods on extensive public real-world graph datasets. These graph datasets are diverse in their sources, scales, heterophily, and tasks. The detailed statistics of these graph datasets are presented in Tab. 17.

### F.2 Details of Compared Methods

In the main text of our work, we compare BA-LOGIC with representative and state-of-the-art graph backdoor attack methods, such as **DPGBA-C** Zhang et al. (2024b), and **UGBA-C** Dai et al. (2023). These methods originally required altering the labels of poisoned nodes. In our experiments, we extend them to the clean-label setting by selecting only poisoned nodes of the target class. We also compare with **ERBA** Xu & Picek (2022), and **ECGBA** Fan & Dai (2024), which are the latest graph backdoor attacks for clean-label settings. We further extend the comparison with our method to **SCLBA** Dai & Sun (2025), **GCLBA** Meguro et al. (2024), **TRAP** Yang et al. (2022) for graph classification; and include **SNTBA** Dai & Sun (2024), **PSO-LB** and **LB** Zheng et al. (2023) for edge prediction. For a fair comparison, hyperparameters of the methods are fine-tuned based on the performance of the validation set. The details of the compared methods are described as follows:

- **ERBA** Xu & Picek (2022): It is the early work among the initial clean-label backdoor attacks on GNNs. ERBA tailors graph classification tasks, samples training graphs randomly as targets, and generates Erdös–Rényi random graphs as triggers. ERBA can also be considered a straightforward variant of SBA Zhang et al. (2021). To adapt ERBA to the settings of our work, we maintain a fixed node size of three within the random graph and select poisoned nodes from training nodes belonging to the target class. All other settings remain consistent with those described in the original work.

- **EBA** Xu et al. (2021): It is the first explainability-based graph backdoor attack. EBA aims to conduct a graph backdoor attack on both node classification and graph classification tasks. For the graph classification task, EBA selects the least important nodes as trigger injecting positions based on the node importance matrix generated by GNNExplainer. Thus, EBA can remain unnoticeable to some extent. For the node classification task, EBA selects the most important node features based on the node importance matrix generated by GraphLIME Huang et al. (2022) and manipulates them as trigger features. To adapt EBA to the settings of our work, we employ GNNExplainer to select the most representative nodes from the target class without altering their labels. The trigger is an Erdös-Rényi random graph with a density of $\rho = 0.8$, as in the original work. All other settings remain consistent with those described in the original work.

- **ECGBA** Fan & Dai (2024): This is one of the latest clean-label graph backdoor attacks focused on node classification. ECGBA completes the graph backdoor attack by coordinating a poison node selector and a trigger generator. It selects nodes that are misclassified as target classes by surrogate GCN as poison nodes, thereby improving performance to a certain extent. However, it should be noted that ECGBA does not consider the inner prediction logic of the target model, and for efficiency, ECGBA's trigger only contains one node, which limits its effect. To adapt ECGBA to the settings of our work, we select poisoned nodes from training nodes in the target class. All other settings remain consistent with those described in the original work.

- **GTA** Xi et al. (2021): GTA selects poisoned nodes randomly but adopts a trigger generator to inject subgraphs as node-specific triggers. The trigger generator is purely optimized by the backdoor attack loss with no constraints. To adapt GTA to the settings of our work, we prohibit GTA from modifying the labels of poisoned nodes and select poisoned nodes from training nodes in the target class. All other settings remain consistent with those described in the original work.

- **UGBA** Dai et al. (2023): It is the state-of-the-art backdoor attack on GNNs. UGBA adopts a representative node selector to utilize the attack budget fully. An adaptive trigger generator is optimized with constraint loss to ensure the generated triggers are unnoticeable. To adapt UGBA to the settings of our work, we prohibit UGBA from modifying the labels of poisoned nodes and select poisoned nodes from training nodes in the target class. All other settings remain with those described in the original work.

- **DPGBA** Zhang et al. (2024b): Except for the node selector and trigger generator, DPGBA adopts an out-of-distribution detector to ensure the attributes of triggers within the distribution and thus achieve unnoticeable attacks. To adapt DPGBA to the settings of our work, we prohibit DPGBA from modifying the labels of poisoned nodes and select poisoned nodes from the target class. All other settings remain consistent with those described in the original work.

- **SCLBA** Dai & Sun (2025): SCLBA is one of the latest clean-label graph backdoor attacks on GNNs for graph classification. SCLBA leverages node semantics by using a specific, naturally occurring type of node as a trigger. Its core design involves selecting semantic trigger nodes based on a node-importance analysis using degree centrality, then injecting these triggers into a subset of samples from the target class graph. To adapt SCLBA to the settings of our work, we select poisoned graphs from training graphs from the target class. All other settings remain consistent with those described in the original work.

- **GCLBA** Meguro et al. (2024): GCLBA is a gradient-based clean-label graph backdoor attack for graph classification. GCLBA comprises two main phases: graph embedding-based pairing and gradient-based trigger injection. The pairing phase establishes relationships between graphs from the target class and those from other classes based on distance in the embedding space, thereby selecting targets far from the decision boundary. The trigger injection phase embeds tailored edges as triggers into paired graphs. To adapt GCLBA to the settings of our work, we select poisoned graphs from training graphs from the target class. All other settings remain consistent with those described in the original work.

- **TRAP** Yang et al. (2022): TRAP is a clean-label graph backdoor attack for graph classification. TRAP generates structure perturbation as triggers without a fixed pattern. TRAP adopts the same black-box setting as SCLBA, achieved by exploiting a surrogate GCN model to generate perturbation triggers via a gradient-based score matrix. To adapt TRAP to the settings of our work, we select poisoned graphs from training graphs from the target class. All other settings remain consistent with those described in the original work.

- **SNTBA** Dai & Sun (2024): SNTBA proposes a backdoor attack targeting GNN models in edge prediction tasks. SNTBA uses a single node as the backdoor trigger, and the backdoor is injected by poisoning selected unlinked node pairs in the training graph. SNTBA injects the trigger to both nodes in the pairs and links them, showing a more relaxed threat model. During inference, the backdoor is activated by linking the trigger node to the two end nodes of unlinked target node pairs in the test graph. The attacked GNN model would incorrectly predict that a link exists between the unlinked target node pairs. To adapt SNTBA to the settings of our work, we select poisoned nodes from training nodes of the target class and prohibit SNTBA from modifying the link state. All other settings remain consistent with those described in the original work.

- **LB** Zheng et al. (2023): LB is a backdoor attack method for edge prediction. LB utilizes a subgraph as a trigger, combining fake/injection nodes with the nodes of the target link. The initial trigger is a random graph comprising two injection nodes and the two target link nodes. LB optimizes triggers using a gradient from an edge-prediction GNN model to minimize the attack objective loss, i.e., the L2 distance between the prediction and the attacker-chosen target link state $T$. The trigger is iteratively updated based on the gradient direction. LB requires modifying the target link state embedded with the trigger to $T$. LB supports white-box and black-box attack scenarios, where the latter utilizes a surrogate model. To adapt LB to the settings of our work, we select poisoned nodes from training nodes of the target class and prohibit LB from modifying the link state. All other settings remain consistent with those described in the original work.

- **PSO-LB** Zheng et al. (2023): PSO-LB is a variant proposed for comparison in the original work of LB. It utilizes particle swarm optimization Kennedy & Eberhart (1995) to modify the injection node features and structure of the trigger. To adapt PSO-LB to the settings of our work, we select poisoned nodes from training nodes of the target class and prohibit PSO-LB from modifying the link state. All other settings remain consistent with those described in the original work.

### F.3   Other Implementation Details

Our implementation is based on PyTorch 2.1.0 and PyTorch Geometric 2.4.0. All the experiments are evaluated on an NVIDIA A100 GPU with 80 GB of memory. The detailed architecture of our method is described as follows. *Firstly*, the framework of Ba-Logic consists of the following modules:

- A 2-layer GCN as the surrogate model.
- A 2-layer GCN as the pre-trained poisoned node selector.
- A 2-layer MLP as the logic-poisoning trigger generator.

*Secondly*, for each architecture of GNN models, we fix the hyperparameters of Ba-Logic as follows:

- Target class: 0
- Trigger size: 3
- Number of GNN layers $L$: 2
- Hidden dimension $H$: 32
- Weight decay: $5e - 3$
- Learning rate: $1e - 2$
- Seeds of NumPy, Torch, and CUDA: 3407
- Activation function: ReLU for GCN and GIN, ELU for GAT

*Thirdly*, the two hyperparameters $\beta$ and $T$ are selected based on the grid search on the validation set. Specifically, $T$ is set as $\{32, 32, 64, 72\}$ and $\beta$ is set as $\{0.8, 0.8, 1.0, 1.2\}$ for **Cora**, **Pubmed**, **Flickr** and **Arxiv**, respectively.

In practice, we set the training epoch for the logic-poisoning trigger generator as 200 for all datasets, and we vary the attack budget $\Delta_P$ on the number of $\mathcal{V}_P$ as $\{100, 100, 200, 200\}$ for **Cora**, **Pubmed**, **Flickr**, and **Arxiv**, respectively.

Table 18: Comparisons of method training time on two datasets.

| Dataset | #Nodes | UGBA | DPGBA | Ba-Logic |
|---------|--------|------|-------|----------|
| Flickr | 89,250 | 32.0s | 57.7s | 123.4s |
| Arxiv | 169,343 | 51.3s | 68.9s | 155.7s |

In our empirical experiments on large-scale graph datasets such as **Flickr** and **Arxiv**, which comprise 89,250 and 169,343 nodes, respectively, we set the size of $\mathcal{V}_P$ to 200 on average and still achieve a much higher ASR than competitors. We also report the overall training time cost of Ba-Logic compared with UGBA and DPGBA on the **Flickr** and **Arxiv** datasets in Tab. 18. The results are consistent with the time complexity analysis in Appendix D, indicating that the Ba-Logic requires only approximately 60 seconds more training time than the two most powerful competitors on a larger graph. The additional time is acceptable given that our Ba-Logic achieves an ASR over 90%, while these competitors achieve an ASR over 60%. This demonstrates that Ba-Logic effectively generates triggers that the target model quickly memorizes by poisoning its inner logic, highlighting its potential for scalability.

## G   Additional Analysis of Attacking Against Adaptive Defenses

In Sec.5.4 and Appendix A.6, we evaluated our method against existing defending strategies. While these widely adopted defense strategies highlight the effectiveness of Ba-Logic, there are gaps between their defending goals and their defenses against logic poisoning.

Specifically, logic poisoning is a novel approach proposed by Ba-Logic that makes the trigger crucial for prediction by forcing a gradient-based importance score to exceed that of clean nodes, thereby directly increasing the probability that the victim model predicts the trigger-attached node as the target class without

altering the label. We find that exploring the performance of Ba-Logic against adaptive defenses, especially those designed to alleviate logic poisoning, could be informative.

In this subsection, we propose four adaptive defenses against logic poisoning attacks to further strengthen our contributions. Here, we present the brief introductions of these adaptive defenses. Specifically:

- **Explainability Regularization (ER)**: We leverage class activation mapping (CAM) Zhou et al. (2016) to measure the neighbors' contribution in predicting the target node. By incorporating an entropy-based regularization term during model training, we penalize low-entropy CAM distributions among neighbors. This in-processing defense aims to prevent any single node from becoming dominant in the prediction.
- **Gradient Masking (GM)**: We first train a victim model and record the neighbors' gradient contribution on the prediction of labeled nodes. Lower entropy implies greater dependence on a specific neighbor, which might be the trigger node. Unlike ER, GM is a pre-processing defense method. We mask out the edges between these nodes and obtain the cleaned graph.
- **Collaborative Defense (CD)**: We train a batch of independent GNNs with diverse initialization, data splits, and hyperparameters, then we adopt an ensemble aggregation to make a final prediction on nodes. Because these independent models have different local prediction logic Deng & Mu (2023), the diverse prediction logic of collaborators can alleviate logic poisoning.
- **Sampling And Masking (SAM)**: We repeatedly sample and mask edges during the training of the victim model. The edges are sampled from a probability distribution indicating the CAM-based importance for node prediction. Note that masking edges enables the model to perform masked forward propagation and update node representations, rather than manipulating the present graph structure. We use the classifier's before-and-after difference in the final prediction as a regularization term to penalize predictions that rely heavily on certain nodes.

Table 19: Results (ASR | clean accuracy(%)) of backdoor methods against adaptive defenses.

| Dataset | Defenses | Acc. | ERBA | ECGBA | EBA-C | GTA-C | UGBA-C | DPGBA-C | Ba-Logic |
|---------|----------|------|------|-------|-------|-------|--------|---------|----------|
| Cora | ER | 84.11 | 7.83\|82.36 | 42.59\|82.07 | 20.74\|75.75 | 30.56\|82.38 | 45.82\|80.37 | 49.31\|82.91 | **89.05\|71.85** |
| | GM | 84.12 | 6.91\|73.71 | 52.31\|75.58 | 19.38\|67.10 | 31.42\|73.73 | 50.41\|71.16 | 51.16\|74.94 | **95.32\|72.31** |
| | CD | 84.10 | 0.05\|74.72 | 44.62\|78.14 | 21.78\|68.11 | 45.34\|74.74 | 62.71\|70.14 | 48.10\|76.45 | **90.47\|70.02** |
| | SAM | 84.09 | 0.03\|73.21 | 45.01\|81.66 | 25.19\|66.60 | 51.08\|73.23 | 39.95\|70.65 | 33.08\|67.87 | **79.56\|75.69** |
| Pubmed | ER | 86.51 | 5.12\|80.57 | 52.17\|86.18 | 17.53\|80.38 | 27.68\|80.98 | 46.11\|74.75 | 57.09\|80.16 | **89.52\|67.22** |
| | GM | 86.51 | 3.21\|75.65 | 56.22\|80.01 | 16.29\|75.46 | 28.45\|76.06 | 52.30\|71.16 | 52.61\|75.16 | **88.07\|66.48** |
| | CD | 86.49 | 0.13\|77.08 | 54.41\|81.55 | 14.68\|76.89 | 21.37\|77.49 | 64.51\|72.24 | 51.11\|76.85 | **83.28\|69.01** |
| | SAM | 86.33 | 0.12\|78.78 | 55.71\|82.13 | 13.24\|78.59 | 20.67\|79.19 | 41.47\|71.65 | 37.01\|81.96 | **86.42\|66.23** |
| Flickr | ER | 46.17 | 7.62\|44.87 | 31.62\|44.28 | 22.34\|44.24 | 31.76\|44.46 | 52.11\|43.92 | 59.74\|43.65 | **79.40\|41.27** |
| | GM | 46.15 | 5.49\|43.49 | 24.85\|43.41 | 20.12\|42.86 | 28.68\|43.08 | 40.26\|42.23 | 47.89\|42.06 | **84.78\|39.84** |
| | CD | 46.15 | 5.12\|44.44 | 27.93\|44.01 | 21.28\|43.81 | 29.53\|44.03 | 46.37\|43.36 | 43.96\|43.18 | **92.29\|40.52** |
| | SAM | 46.04 | 4.38\|43.67 | 26.71\|43.12 | 19.85\|43.04 | 27.83\|43.26 | 43.05\|42.66 | 40.18\|42.47 | **85.91\|39.26** |
| Arxiv | ER | 66.51 | 6.95\|64.81 | 19.84\|65.41 | 20.23\|64.95 | 28.76\|65.17 | 44.39\|64.37 | 51.73\|64.92 | **80.06\|61.83** |
| | GM | 66.52 | 5.82\|64.32 | 17.92\|64.98 | 18.45\|64.46 | 25.88\|64.68 | 49.87\|63.86 | 46.25\|64.39 | **92.36\|60.97** |
| | CD | 66.39 | 5.67\|64.48 | 18.63\|65.12 | 19.56\|64.62 | 26.72\|64.84 | 42.51\|64.01 | 48.37\|64.57 | **85.38\|62.34** |
| | SAM | 66.21 | 5.21\|64.02 | 18.21\|64.73 | 19.12\|64.16 | 27.15\|64.38 | 40.94\|63.52 | 47.06\|64.08 | **81.09\|60.46** |

We evaluate our Ba-Logic and competitors against the proposed adaptive defense methods. We first fine-tuned these adaptive defenses based on their performance against Ba-Logic. We also report the clean accuracy of vanilla GNN models after applying the defenses, with no attacks, as **Accuracy**. Then we present the ASR and clean accuracy(%) of these methods in Tab. 19. The gray cell indicates the competitor with the highest ASR. From the table, we obtain the following key observations:

- The adaptive defense can partially weaken the backdoor, indicating promising directions against logic poisoning. However, under our Ba-Logic, the ASR remains generally high, while clean accuracy significantly drops after applying adaptive defenses. This highlights the need for further in-depth investigation into adaptive defenses.

- Our method consistently maintains the highest ASR (generally over 80%) across adaptive defenses and datasets. This indicates the superiority of our Ba-Logic in poisoning inner logic for clean-label backdoor.
- Our Ba-Logic maintains the effective ASR-clean accuracy trade-off across datasets and various types of defenses. This highlights the challenge of fully cleansing the victim model, which already learns the poisoned prediction logic and relies on the injected triggers when predicting the poisoned nodes

## H  Additional Empirical Validations on Theoretical Analysis

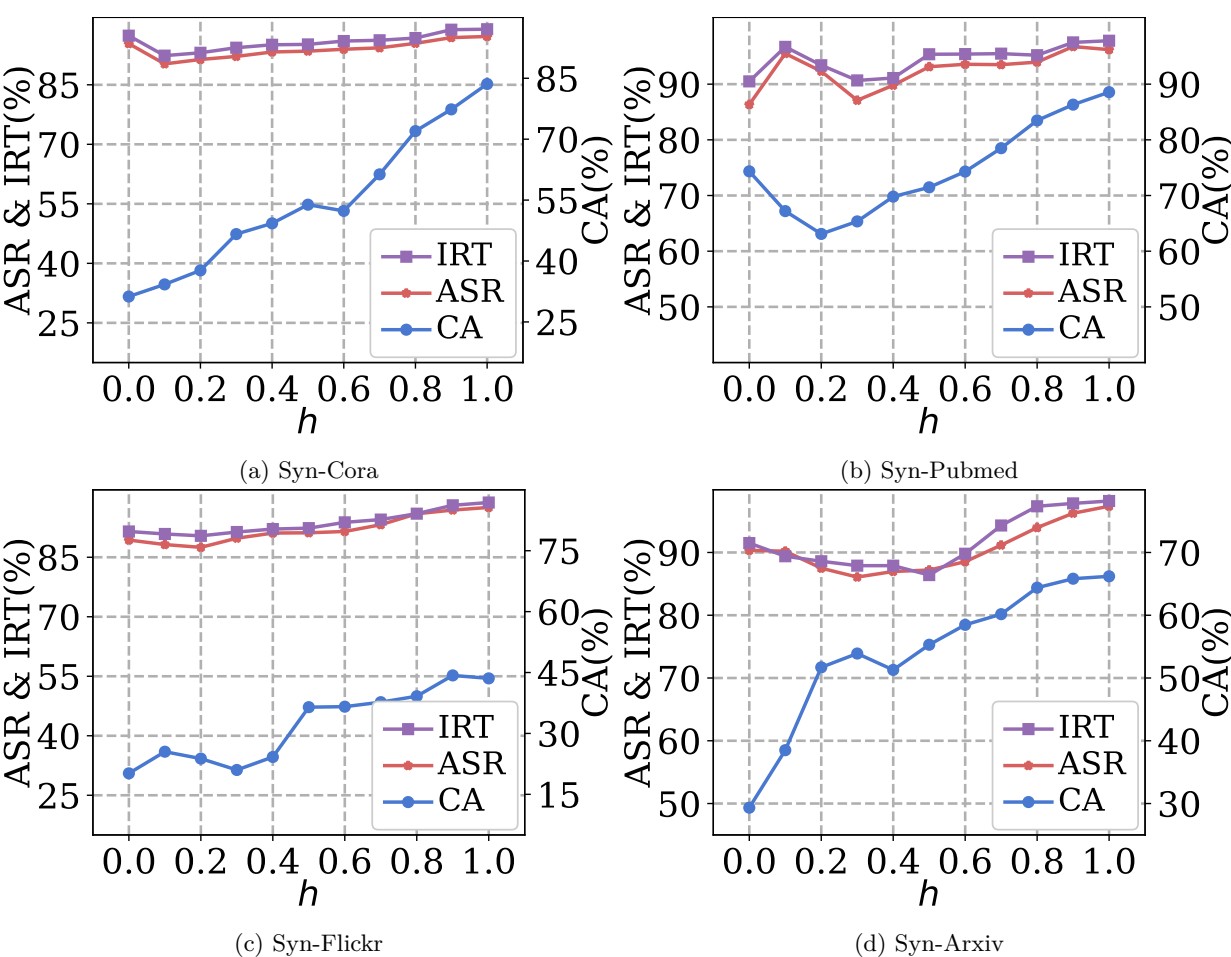

Figure 11: Performance of Ba-Logic on synthetic graphs with different feature-label correlations.

In the main text of our work, we conducted a theoretical analysis to show that the core failure of existing methods under clean-label settings lies in that their triggers are deemed unimportant for prediction by GNN models. Specifically, we propose a novel metric, IRT, to measure the importance rate of triggers and establish a theoretical connection between IRT and attack success.

The theoretical analysis in Sec. 2.3 established the correlation between the IRT value and the probability of attack success. In this subsection, we aim to empirically investigate whether the result of our theoretical analysis, i.e., the probability of attack success is bounded by the IRT value, still holds under various feature-label correlations from real-world graphs.

Inspired by Zhu et al. (2020), we use the **edge homophily** $h$, the fraction of edges in a graph that connect nodes that have the same labels, as a measure for the homophilous or heterophilous level of the feature-label correlation. To investigate how the homophilous and heterophilous correlations affect our method, we generate synthetic graphs based on **Cora**, **Pubmed**, **Flickr**, and **Arxiv** with various $h$. We illustrate the ASR | clean accuracy(%) and IRT(%) of Ba-Logic on the synthetic graphs in Fig. 11.

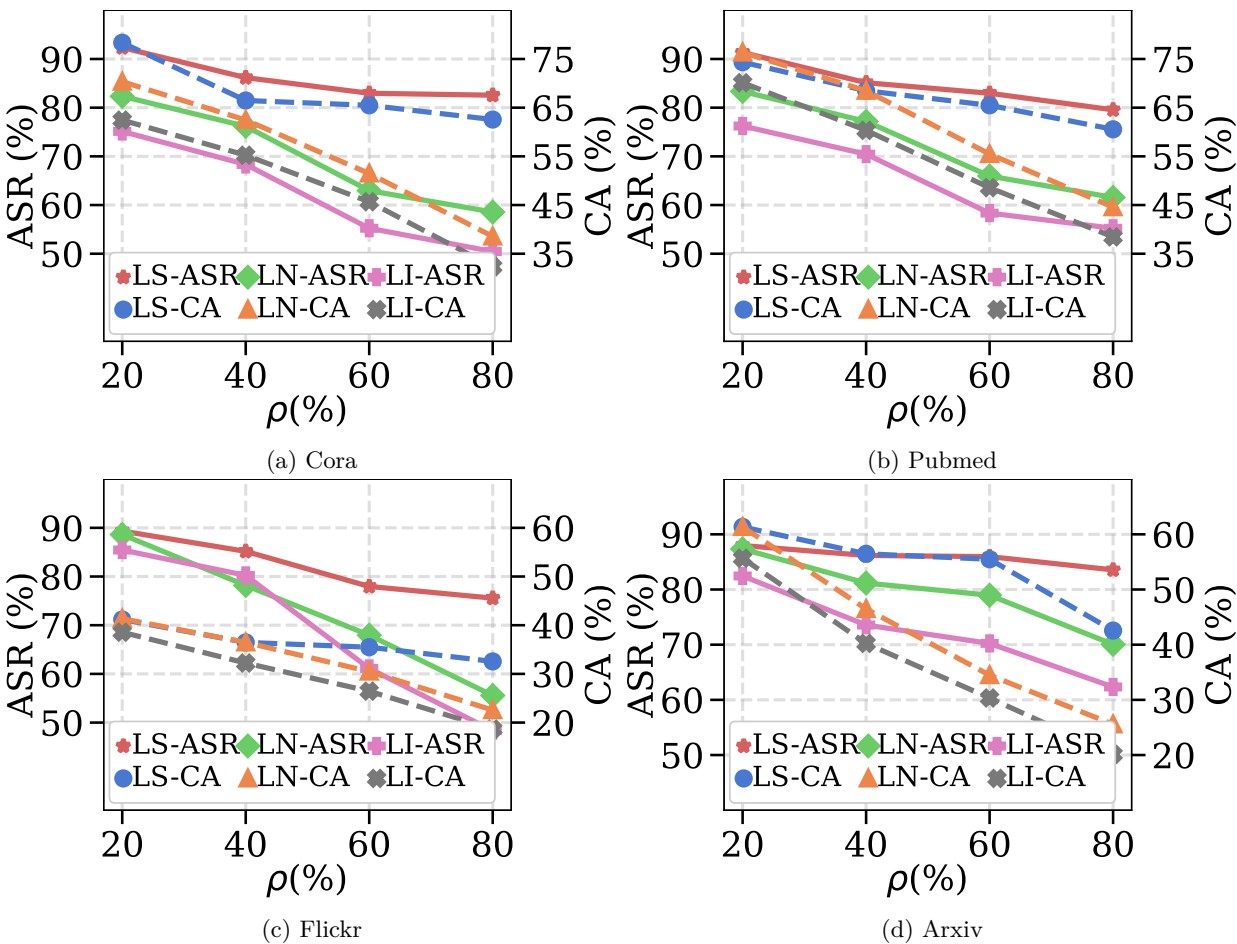

Figure 12: Performance of BA-LOGIC under sparse, noisy, and imbalanced label settings.

From the figures, we obtain the following key insights:

- ASR and IRT values are closely aligned for different $h$. It indicates that the theoretical analysis of triggers with a higher importance rate can achieve better attack performance remains valid for complex feature-structure correlations
- Our BA-LOGIC consistently achieves high ASR across datasets with various $h$. This indicates logic poisoning remains effective, facing complex feature-label correlations, as the GNN with poisoned prediction logic still identifies our trigger as important for prediction
- Clean accuracy changes significantly with varying $h$. It is consistent with observations from prior works that homophily mainly affects the generalization of GNNs on clean nodes.

## I  Additional Analysis on Generalizability under Challenge Settings

In the main text of our work, we evaluated our method across various graphs and target models. To further assess the generalizability of our method, here we systematically evaluate BA-LOGIC under five challenging yet realistic settings.

### I.1  Generalizability to Sparse, Noisy, and Imbalanced Labels

In real-world graphs, supervision can be incomplete or corrupted, which may challenge the adaptive poisoned node selection strategy. To investigate the selection strategy and performance of BA-LOGIC with low-quality supervision, we propose three challenging settings of labels, specifically:

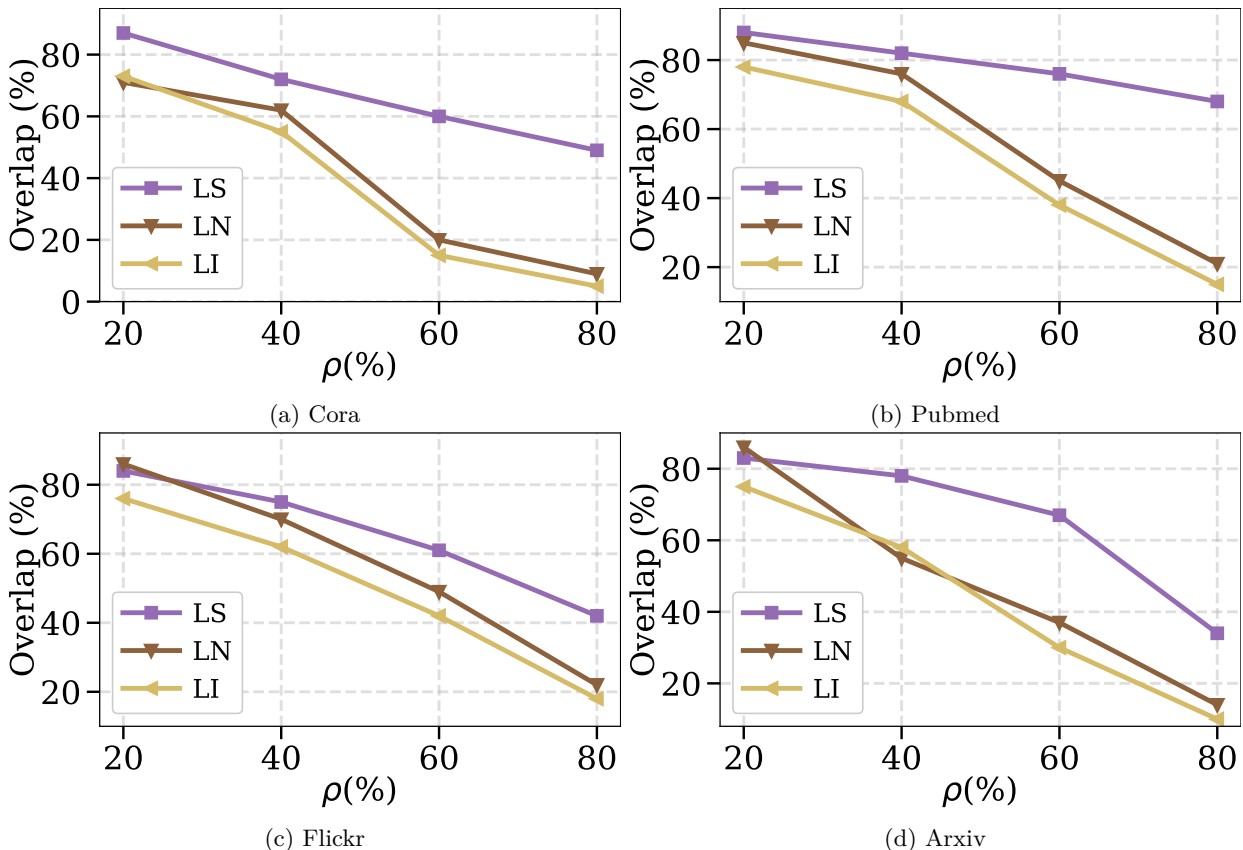

Figure 13: The Jaccard overlap of poisoned node selection under different label settings.

- **Label Sparsity (LS)**: We randomly mask a ratio of labels across all training nodes, and then retrain the poisoned node selector to obtain a new set of poisoned nodes.

- **Label Noise (LN)**: We randomly flip a ratio of labels of training nodes to other classes to simulate bad annotations, and then retrain the poisoned node selector to obtain a new set of poisoned nodes.

- **Label Imbalance (LI)**: We randomly mask a ratio of labeled nodes from the target class while keeping training nodes of other classes unchanged.

For each setting, we report (i) ASR and clean accuracy(%) of our method and (ii) the Jaccard overlap between the original set of poisoned nodes and the set selected under various challenging settings. We present the results in four graphs: Fig. 12 shows performance, and Fig. 13 shows overlap. From the figures, we have the following key findings:

- These supervision perturbations consistently degrade ASR and clean accuracy across all four datasets. But it mainly affects the clean accuracy of GNN models, not our method, as it retains high ASR while clean accuracy drops significantly.

- The Jaccard overlap between the original set of poisoned nodes and the manipulated set decreases smoothly as labels become sparser or noisier. It indicates that the perturbations challenge the poisoned node selector, since it is a standard 2-layer GCN whose generalizability may be affected by the settings.

- Label sparsity is consistently less harmful than label noise and label imbalance, as both ASR and overlap stay higher under LS than LN or LI. It suggests that Ba-Logic prefers fewer but more reliable labels over many corrupted ones.

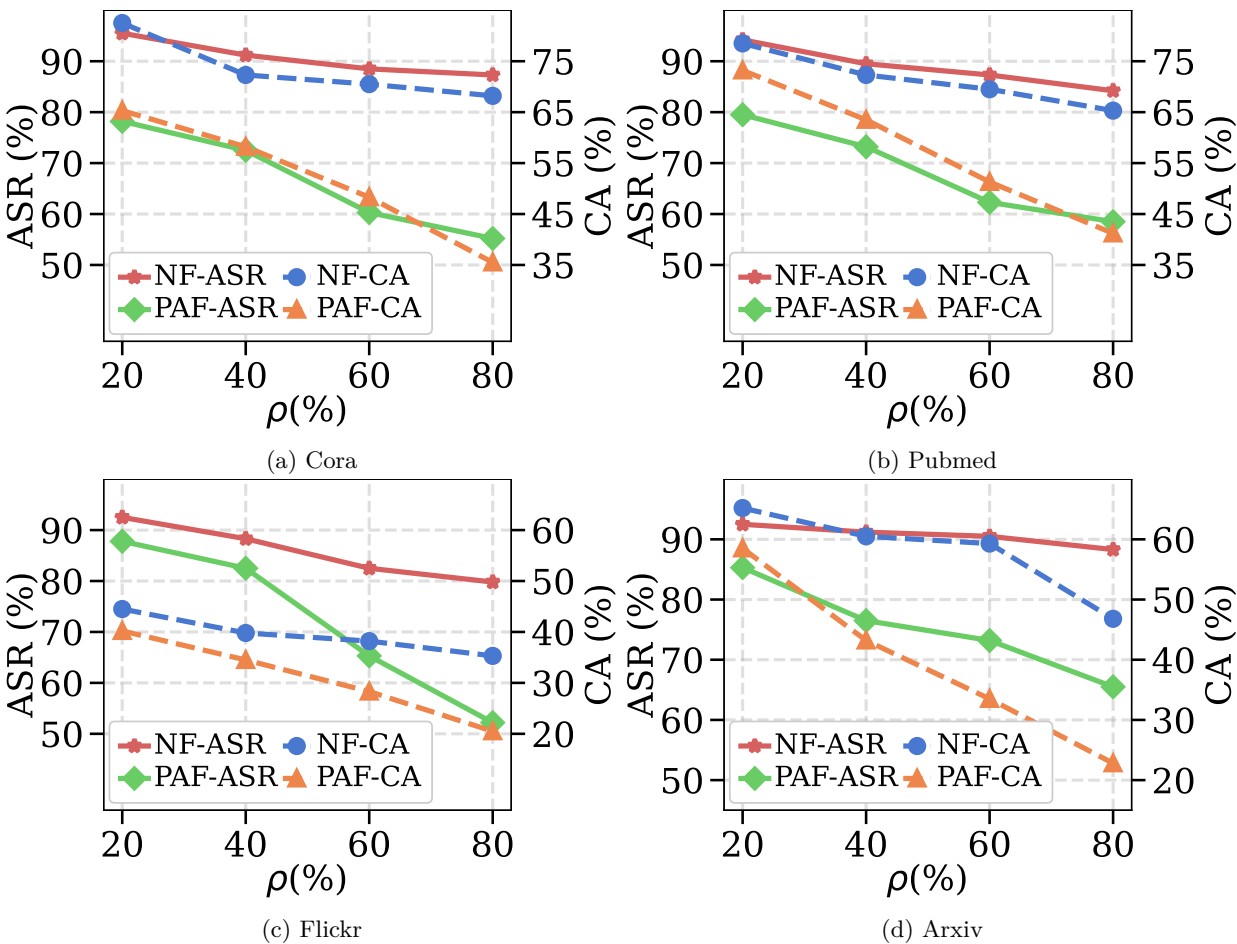

Figure 14: Performance of Ba-Logic under noisy and partially accessible feature settings.

## I.2 Generalizability to Noisy and Partially Accessible Features

In addition to the challenges posed by labels, node features in real-world graphs can be noisy or only partially accessible. To investigate the performance of Ba-Logic under degraded feature quality and accessibility, we further introduce two challenging settings of features, specifically:

- **Noisy Features (NF)**: We add dimension-wise Gaussian noise to node features before training Ba-Logic on the same graph, and vary the noise level to simulate different degrees of feature corruption.

- **Partially Accessible Features (PAF)**: We restrict the access to node features to only a part of the dimensions during the training of Ba-Logic, while the target GNN is still trained on the full feature space.

We first unify the perturbation ratios into normalized levels for the two settings, defined as:

$$\rho_{NF} = \frac{\sigma - \sigma_{\min}}{\sigma_{\max} - \sigma_{\min}}, \qquad \rho_{PAF} = 1 - \frac{d'}{d}, \tag{32}$$

where $\sigma$ is the standard deviation of the injected Gaussian noise, $\sigma_{\min}$ and $\sigma_{\max}$ are the minimum and maximum noise levels used in our experiments, and $\frac{d'}{d}$ denotes the ratio of visible feature dimensions under the PAF setting.

We illustrate the results across four graphs with respect to ASR and clean accuracy(%) of our method in Fig. 14 with different noise levels and feature partials. From the figures, we have the following key findings:

- Increasing the feature perturbation level consistently degrades both ASR and clean accuracy across all four datasets. Our method retains high ASR, showing that logic poisoning remains effective even when feature quality deteriorates.

- NF mainly affects the GNN's clean accuracy, while PAF has a more significant impact on the ASR. It is consistent with the label noisy analysis, indicating that noise primarily affects GNN performance rather than logic poisoning.

