# OpenReview forum: "Poisoning the Inner Prediction Logic of Graph Neural Networks for Clean-Label Backdoor Attacks"
_TMLR — Under review for TMLR_

### Review · Reviewer_wgVA · 2026-05-05

**Summary Of Contributions:**

This paper proposes a method for backdoor attacks on GNN models. Unlike existing works that focus on manipulating labels of training nodes, this work argues that manipulating training node labels is not always feasible. To this end, this paper studies the 'clean-label' setting, where the labels of training nodes are never changed. The methodology proposed by this paper is to keep the labels unchanged and inject selective triggers to selected nodes. Attacked nodes are selected according to prediction uncertainty, and the triggers are optimized to maximize their importance scores towards the target class.

The proposed method BA-logic achieves good attack performance (~100% attack success rate), good clean accuracy, and good transferability across architectures.

**Audience:**

Yes

**Audience Explanation:**

GNN is the cornerstone of various applications involving search, advertising, and recommendation, and therefore the robustness of GNNs would be of interest to a wide range of practitioners.

**Broader Impact Concerns:**

No concerns.

**Claims And Evidence:**

No

**Claims Explanation:**

From my point of view, this paper is clearly written and easy to follow. The proposed problem of "clean-label" backdoor attack is of practical importance as it does not require changes to the node labels. The experimental results are extensive and cover most aspects that I am interested.

Some of my questions and weaknesses of this paper (that I found) are listed as follows.

1. Potentially unfair comparison against baselines. As this paper tackles a new problem of clean-label attacks, the baselines (which are designed for label-manipulating attacks) will inevitably be unsuitable under this setting. It would be slightly unfair, if the authors did not adequately modify some of the algorithm setups of the baselines under the new setting.

2. Potential train-test mismatch issue (but not seen in the experiments). The authors state that the attacked nodes during training are those that exhibit high uncertainty. It makes sense, but the attacked nodes during inference may not follow such an assumption (e.g. the nodes to be attacked during inference may be quite certain). It makes me wonder, why is the method able to overcome such train-test mismatch and achieve ~100% ASR? It may not be a weakness, but in my opinion an observation that should be better justified.

3. The 'unnoticable‘ loss should be better justified. From my understanding, "unnoticable" should contain more meanings than a plain cosine similarity of features. It may contain some more nuanced features e.g. connectivity, scale of node features, etc. It may not be sufficient to optimize cosine similarity alone. A trivial example would be that, some classes are associated with very high feature scales, and the triggers may simply be some nodes that have features with large magnitudes.

4. I would like to see some more analyses on the triggers themselves. It seems that, through the importance score analysis, the generated triggers can be more or less explained. However, from the authors' methodology of maximizing gradient, it may also look like an "adversarial" attack with no clear semantics but just simply doing gradient ascent. I wonder which understanding is correct for BA-logic, and how are they different from existing backdoor attack patterns.

**Requested Changes:**

The paper is generally solid and some more analyses on the points I mentioned would strengthen this paper a lot.

---

> ### Author Response · Authors · 2026-05-27
> **Response to Reviewer wgVA Part 1**
>
> We are delighted that the reviewer finds our results promising. We also appreciate the reviewer for the constructive feedback. We hope the following additional experiments and clarifications can address the reviewer's remaining concerns.
>
> ## 1. Clarification on Baselines
>
> > "Potentially unfair comparison against baselines." (W1)
>
> We clarify that we have made great efforts in adopting these existing backdoor attacks to the clean-label setting for a fair comparison. We kindly invite the reviewer to check **Appendix F.2** for details of the adaptation. Specifically:
>
> * For attacks designed for clean-label backdoors that do not alter labels, we directly apply them.
> * For dirty-label backdoor attacks that alter labels, we restrict them to injecting triggers into target class nodes, thus not needing to alter labels.
> * Hyperparameters of the baselines are fine-tuned based on the performance of the validation set. Main algorithms remain consistent with those described in the original work.
>
> Besides, we clarify that the premise of label alteration is not decisive for the performance of the backdoor, as:
>
> * Our analysis in **Sec 2.3** indicates that performance depends on whether triggers **poison the inner prediction logic**, not on label alteration
> *  **Table 3** proves that leading dirty-label competitors, such as GTA, UGBA, and DPGBA, can still achieve successful attacks **without altered labels**. The results are consistent with the analysis in **Appendix A.8.1** that reveals that effective triggers present higher IRT values, as the backdoored model identifies them as more important than normal nodes for prediction.
>
> ## 2. Clarification on Novelty and Contribution
>
> > "Potential train-test mismatch issue" (W2)
>
> We appreciate the reviewer raising the question. In our paper, the main purpose of poison node selection is more efficient usage of attack budget, and BA-Logic's performance is primarily contributed by inner logic poisoning, rather than poison node selection.
>
> Specifically, we summarize the pipeline of logic poisoning as:
> * The poisoned node selection serves solely as positions for trigger injection, as these nodes make injected triggers appear as key features for the target class
> * The logic poisoning trigger generator creates adaptive triggers, guided by the prediction logic poisoning loss in **Eq.7**, to mislead the model to treat these triggers as crucial patterns for predicting the target class.
> * After injecting the triggers, the target model trained on the backdoored graph is backdoored, and **any test node could be successfully attacked** by attaching triggers during the inference of the target model, no matter whether the prediction on the target node is certain or not.
>
> For your convenience, we compare the **ASR|CA(%)** of diverse variants of baselines and BA-Logic on **Pubmed** and **Arxiv** with GCN as the target model. Specifically, **-S** denotes baselines updated with our poison node selection; **BA-Logic\S** randomly selects poisoned nodes from the training graph; and **BA-Logic\T** removes the logic-poisoning loss.
>
> ||GTA-C|GTA-S|UGBA-C|UGBA-S|BA-Logic|BA-Logic\S|BA-Logic\T|
> |-|-|-|-|-|-|-|-|
> |Pubmed|38.84\|86.17|41.57\|86.10|71.24\|85.31|68.87\|85.82|96.75\|86.03|89.8\|85.72|18.4\|85.95|
> |Arxiv|37.16\|66.29|37.98\|65.79|69.71\|66.57|70.79\|66.08|98.04\|65.82|85.0\|65.55|15.3\|65.71|
>
>
> From the tables, we have the following key findings:
> * BA-Logic\S exhibits a slightly lower ASR than BA-Logic, while BA-Logic\T suffers an ASR drop by a large margin. It highlights that our logic poisoning triggers are paramount for effective graph backdoors.
> * All baselines show enhanced ASR when using our poison node selection, confirming its effectiveness and generalizability.
> * The primary source of our method’s superior ASR is the logic poisoning mechanism with solid theoretical ground. This is evidenced by that updated baselines still fail to compare with our method.

---

> ### Author Response · Authors · 2026-05-27
> **Response to Reviewer wgVA Part 2**
>
> > "More analyses on the triggers themselves, and how BA-Logic is different from existing methods." (W4-1)
>
> We clarify the inherent differences between BA-Logic and existing graph backdoor attacks as follows:
>
> * For the objective, our method aims to poison the inner prediction logic of GNNs for clean-label graph backdoor attacks. In contrast, existing methods aim to obtain triggers that are difficult to be noticed for unnoticeable backdoor attacks (e.g., UGBA), or that are efficient in trigger size (e.g., ECGBA)
>
> * For trigger generation, our method enforces the importance scores of trigger nodes to exceed those of clean nodes within the computational graph of poisoned nodes by optimizing the logic poisoning loss in **Eq.7**.  In contrast, existing methods, such as UGBA, optimize the GNN's task loss to predict trigger-attached nodes as the target class, without explicit design on poisoning the prediction logic
>
> * For poisoned node selection, our method selects high-uncertainty nodes from the target class. The probability that these nodes are predicted as the target class is low, and they are also uncertain for all other classes, so the trigger can take over the prediction logic on these nodes. In contrast, existing works, such as UGBA, uses K-Means to select representative nodes that are strongly correlated with their class labels. Thus, GNN models can still predict with normal patterns and ignore triggers
>
> ---
>
> > "It seems that it may also look like an "adversarial" attack. (W4-2)"
>
> To highlight the difference between our method and existing adversarial attacks, we clarify that clean-label setting is a more practical and challenging scenario for graph attacks. For instance, account labels of Twitter are annotated and stored within a well-protected backend system, making it nearly impossible for attackers to alter the labels.
>
> For this rebuttal, we summarize the difference between backdoor attacks and adversarial attacks as follows:
>
> * Backdoor attack aims to associate the trigger with the target class in the training data to mislead target models. The backdoored model predicts the target class given an arbitrary sample attached with the trigger, while behaves normally on clean samples. In contrast, adversarial attacks manipulate the training graph of the model to control the predictions on target nodes [1].
>
> * Backdoor attacks focus on an inductive setting, aiming to generalize the backdoored pattern to unseen test nodes with triggers attached. In contrast, the majority of adversarial attacks focus on a transductive setting, in which the target nodes are accessible during the training of adversarial attacks [1].
>
> Besides the inherent differences, our method focuses on poisoning the inner prediction logic of target GNNs for clean-label graph backdoor attacks. To achieve this, our method optimizes the importance scores of trigger nodes to exceed those of clean neighbor nodes for effective logic-poisoning trigger generator. A poisoned node selection process designed for clean-label setting is further adopted.
>
>
> For clarity, we summarize a table of differences between our method and adversarial attacks from three aspects:
>
> |Aspect|Adversarial Attacks|BA-Logic|
> |-|-|-|
> |**Poison Target**|Training data|GNN's inner logic|
> |**How Achieve**|Manipulate feature and structure|Inject trigger|
> |**Victim Behavior**|Misclassify specific targets|Misclassify arbitrary targets with trigger attached|
>
>
> [1] A Comprehensive Survey on Trustworthy Graph Neural Networks: Privacy, Robustness, Fairness, and Explainability

---

> ### Author Response · Authors · 2026-05-27
> **Response to Reviewer wgVA Part 3**
>
> ## 3. Additional Analysis on Unnoticeable Constraint
>
> > "The 'unnoticable‘ loss should be better justified. It may not be sufficient to optimize cosine similarity alone." (W3)
>
> We'd like to clarify that the cosine similarity constraint is commonly used by previous work [1,2] due to its interpretability and effectiveness.
>
> Following the reviewer's suggestion, we further validate two more unnoticeable constraints. Specifically:
>
> **Feature Magnitude**: To avoid triggers relying on abnormal feature scales, we constrain the trigger nodes to match the local feature magnitude of the target node:
>
> $$ \min_{\theta_ g}\mathcal{L}_ {M}=\sum_{v_ i\in V}\sum_{v_ g\in g_ i}\left(\|\mathbf{x}_ g\|_ 2-\bar{m}_ i\right)^2,\quad \bar{m}_ i=\frac{1}{|\mathcal{N}_ i|}\sum_{v_ c\in\mathcal{N}_ i}\|\mathbf{x}_ c\|_ 2 $$
>
> where $\mathcal{N}_ i=\mathcal{N}(v_ i)\cup\{v_ i\}$ denotes the local neighborhood of $v_ i$.
>
> **Edge Connection**: Since edge-based defenses can remove edges connecting dissimilar nodes, we constrain trigger-related edges to follow the local edge-similarity pattern:
>
> $$ \min_{\theta_ g}\mathcal{L}_ {C}=\sum_{v_ i\in V}\sum_{(v_ j,v_ k)\in E_ B^i}\max\left(0, \bar{s}_ i-\mathrm{sim}(v_ j,v_ k)\right)^2,\quad \bar{s}_ i=\frac{1}{|E_ i|}\sum_{(v_ p,v_ q)\in E_ i}\mathrm{sim}(v_ p,v_ q) $$
>
> where $E_ i$ denotes the edge set in the local neighborhood of $v_ i$, and $E_ B^i$ follows Eq.(8), containing edges inside trigger $g_ i$ and edges attaching $g_ i$ to node $v_ i$.
>
> With the introduction, we further evaluate **BA-Logic-M** and **BA-Logic-E** against four existing defenses and four adaptive defenses. We report **ASR|CA(%)** of our methods against various defenses on **Pubmed** and **Arxiv** below.
>
>
> **Pubmed**
>
> ||GCN-Prune|RobustGCN|GNNGuard|RIGBD|ER|GM|CD|SAM|
> |-|-|-|-|-|-|-|-|-|
> |BA-Logic|97.95\|85.06|98.10\|84.83|95.46\|84.54|93.01\|84.32|89.52\|67.22|88.07\|66.48|83.28\|69.01|86.42\|66.23|
> |BA-Logic-M|97.68\|85.10|98.32\|84.91|95.18\|84.60|92.74\|84.35|88.64\|67.65|87.45\|66.92|82.71\|69.44|85.96\|66.71|
> |BA-Logic-E|98.24\|85.04|97.85\|84.79|96.02\|84.61|93.44\|84.26|88.93\|67.31|88.54\|66.57|83.06\|69.10|86.81\|66.36|
>
> **Arxiv**
>
> ||GCN-Prune|RobustGCN|GNNGuard|RIGBD|ER|GM|CD|SAM|
> |-|-|-|-|-|-|-|-|-|
> |BA-Logic|96.75\|63.22|97.03\|64.07|95.37\|63.31|93.23\|61.99|80.06\|61.83|92.36\|60.97|85.38\|62.34|81.09\|60.46|
> |BA-Logic-M|96.42\|63.30|97.21\|64.12|94.96\|63.39|92.87\|62.08|79.34\|62.06|91.72\|61.24|84.86\|62.55|80.52\|60.77|
> |BA-Logic-E|97.18\|63.28|96.78\|64.10|95.91\|63.42|93.66\|62.05|79.71\|61.95|92.74\|61.09|85.12\|62.43|81.46\|60.62|
>
> From the tables, we have the following key findings:
> * **BA-Logic-M** and **BA-Logic-E** preserve comparable ASR and CA to BA-Logic, showing that the extended unnoticeable constraints do not weaken logic poisoning.
>
> We have updated our draft with these new unnoticeable constraints and kindly invite the reviewer to check the additional analysis in **Appendix A.9**
>
>
> [1] Unnoticeable backdoor attacks on graph neural networks. WWW 2023
>
> [2] Rethinking graph backdoor attacks: A distribution-preserving perspective. KDD 2024

---

### Review · Reviewer_A8JU · 2026-05-11

**Summary Of Contributions:**

Summary:
This paper proposes a clean-label graph backdoor attack that poisons GNNs' inner prediction logic via uncertainty-based node selection and gradient-based importance-margin loss to make triggers dominate predictions.

Strengths:
1 - Well-motivated problem; preliminary analysis with IRT metric clearly shows existing clean-label methods fail to make triggers influential.
2 - Combines empirical study, giving theoretical bound linking IRT to attack success and a bi-level optimization design.
3 - Extensive experiments across node/graph/edge tasks, homophilous and heterophilous graphs, multiple defenses including adaptive ones.

Weaknesses:
1 - Theoretical bound relies on strong assumptions (regular graph, dimensional feature independence, clean neighbors all share label), limiting real-world relevance.
2 - Upper bound in Theorem 1 does not actually depend on $y_t$ vs other classes asymmetrically; only shows probability bound grows with $\gamma$, not that attack succeeds. This raises a gap between bound and claimed implication.
3 - Sensitivity analysis shows performance varies sharply with $T$ and $\beta$, contradicting claim of robustness to hyperparameters.
4 - Unnoticeability evaluated mainly via cosine-similarity constraint and degree-pruning; no human or distributional detectability study, and RIGBD is the only recent detection-style defense tested.
5 - Clean accuracy of Ba-Logic still drops noticeably under adaptive defenses, but this trade-off is downplayed in the authors' narrative.

**Audience:**

Yes

**Audience Explanation:**

Yes, the paper studies a realistic and underexplored clean-label graph backdoor setting with a novel logic-poisoning perspective, which can interest researchers in TMLR's audience working on graph learning, trustworthy ML, and adversarial robustness.

**Claims And Evidence:**

No

**Claims Explanation:**

See Weaknesses 2 and 3.

**Requested Changes:**

I would appreciate the revisions addressing weaknesses 1-5, which I believe will be necessary before this paper gets ready for publication. Plus, does the unnoticeability constraint hold when triggers must have high importance? In other words, is there an inherent tension between Eq.(7) and Eq.(8)? To me they seem to show conflict requirements, or there should be a pareto optimal frontier for these triggers?

---

> ### Author Response · Authors · 2026-05-27
> **Response to Reviewer A8JU Part 1**
>
> We are delighted that the reviewer finds our research problem well-motivated. We also thank the reviewer for the constructive feedback. We hope the following additional experiments and clarifications can address the reviewer's remaining concerns.
>
> ## 1. Clarification on Theoretical Analysis
>
> > "Theoretical bound relies on strong assumptions (regular graph, dimensional feature independence, clean neighbors all share label), limiting real-world relevance (W1), and its upper bound does not actually depend on  vs other classes asymmetrically (W2)."
>
>
> We clarify that we do not assume clean neighbors all share the same label. In practice, feature–structure correlations in real graphs arise because labels are annotated based on features, so structure correlates with labels and thus indirectly with features [1]. We use common assumptions for graph machine learning adopted by theoretical analysis of prior works [2,3].
>
> Additionally, we clarify that we investigated whether our theoretical analysis still held on real-world graphs with complex feature-structure correlations in **Appendix H**. Specifically, we use **edge homophily $h$**, the fraction of edges in a graph that connect nodes that have the same labels, as a measure for the strength of the feature-structure correlation.
>
> For your convenience, we record the average **ASR|CA(%) and IRT(%)** of our method on two synthetic graphs with various $h$ as follows:
>
> **syn-cora**
>
> |$h$|ASR\|CA|IRT|
> |-|-|-|
> |0.1|91.27\|34.24|89.35|
> |0.3|93.15\|46.63|91.35|
> |0.5|94.48\|53.87|93.18|
> |0.7|95.31\|61.34|95.22|
> |0.9|97.89\|77.34|96.89|
>
> **syn-pubmed**
>
> |$h$|ASR\|CA|IRT|
> |-|-|-|
> |0.1|86.21\|74.17|90.50|
> |0.3|92.33\|63.10|93.41|
> |0.5|89.80\|69.73|91.10|
> |0.7|93.54\|74.31|95.43|
> |0.9|96.74\|86.22|97.49|
>
> Key findings:
>
> * ASR and IRT values are closely aligned. It indicates that the theoretical analysis of triggers with a higher importance rate can achieve better attack performance remains valid for complex feature-structure correlations
>
> * Our method consistently achieves high ASR across datasets with various $h$. This indicates logic poisoning remains effective facing complex feature-label correlations, as the GNN with poisoned prediction logic still identifies our trigger as important for prediction
>
> * CA changes significantly with varying $h$. It is consistent with observation from prior work [4] that homophily mainly affects the generalization of GNNs on clean nodes.
>
>
> Besides, we would like to clarify the contribution of our theoretical analysis.
>
>
> Intuitively, the importance scores of triggers reflect their influence on the predictions of backdoored GNNs. However, there remains a gap: triggers with small importance scores might still cause changes in predicted labels.
>
> To bridge this gap, our theorem shows that under the backdoor setting, higher trigger importance scores lead to higher attack success rates.  This proves that increasing the importance scores of triggers is an effective way for successful clean-label backdooring. It also directly motivates our logic poisoning loss $\mathcal{L}_A$ in **Eq.7**, which enforces the important score of trigger nodes to exceed those of clean neighbors of trigger-attached nodes.
>
>
> [1] On Node Features for Graph Neural Networks, NeurIPS 2019
>
> [2] Robustness Inspired Graph Backdoor Defense, ICLR 2025
>
> [3] Label-wise Graph Convolutional Network for Heterophilic Graphs, LoG 2022
>
> [4] Beyond Homophily in Graph Neural Networks: Current Limitations and Effective Designs, NeurIPS 2020

---

> ### Author Response · Authors · 2026-05-27
> **Response to Reviewer A8JU Part 2**
>
> ## 2. Clarification and Additional Analysis on Performance
>
> > "Sensitivity analysis shows performance varies, contradicting claim of robustness to hyperparameters." (W3)
>
>
> We clarify that the claim of robustness to hyperparameters remains valid because hyperparameters vary significantly in the corresponding analysis.
>
> * We vary $T$ and $\beta$ over a very wide range, including ablating the logic-poisoning loss and amplifying it by up to $100\times$. The performance variation under such extreme settings is expected.
>
> * Our robustness claim refers to practical hyperparameter choices around the validated setting, where BA-Logic still maintains high ASR.
>
> * The trend further supports that BA-Logic performs well once the logic-poisoning strength is sufficient. While large $T$ or $\beta$ can hinder optimization, it is a normal trade-off rather than a limitation or contradiction.
>
>
> ---
>
> > "Unnoticeability evaluated mainly via cosine-similarity constraint and degree-pruning; no human or distributional detectability study. (W4) Does the unnoticeability constraint hold when triggers must have high importance? In other words, is there an inherent tension between Eq.(7) and Eq.(8)? (RQ) "
>
> We clarify that the cosine similarity constraint is commonly used in prior work [1,2] for its interpretability and effectiveness. And the high importance of triggers does not conflict with maintaining concealment, as the optimization objectives in Eq.7 and Eq.8 differ.
>
> Following the reviewer's suggestion, we further validate one more unnoticeable constraint. Specifically:
>
> **Feature Distribution**: To avoid triggers exhibiting abnormal feature-level distributional patterns, we constrain the trigger nodes to match the local feature distribution of the target node:
>
> $$ \min_{\theta_ g}\mathcal{L}_ {D}=\sum_{v_ i\in V}\Delta_{\mathrm{feat}}\left(\{\mathbf{x}_ g\mid v_ g\in g_ i\},\{\mathbf{x}_ c\mid v_ c\in\mathcal{N}_ i\}\right) $$
>
> where $\mathcal{N}_ i=\mathcal{N}(v_ i)\cup\{v_ i\}$ denotes the local neighborhood of $v_ i$. $\Delta_{\mathrm{feat}}(\cdot,\cdot)$ measures the distribution discrepancy between the generated trigger node features and the local clean node features, which is calculated by the squared Maximum Mean Discrepancy (MMD). Different from cosine similarity, this compares two sets of node features at the distribution level.
>
> With the introduction, we further evaluate **BA-Logic-D** against four existing defenses and four adaptive defenses. We report **ASR|CA(%)** of our methods against various defenses on **Pubmed** and **Arxiv** below.
>
> **Pubmed**
>
> ||GCN-Prune|RobustGCN|GNNGuard|RIGBD|ER|GM|CD|SAM|
> |-|-|-|-|-|-|-|-|-|
> |BA-Logic|97.95\|85.06|98.10\|84.83|95.46\|84.54|93.01\|84.32|89.52\|67.22|88.07\|66.48|83.28\|69.01|86.42\|66.23|
> |BA-Logic-D|96.91\|85.28|97.42\|85.02|94.68\|84.72|91.85\|84.54|87.36\|67.86|86.91\|67.11|81.74\|69.66|84.93\|66.91|
>
> **Arxiv**
>
> ||GCN-Prune|RobustGCN|GNNGuard|RIGBD|ER|GM|CD|SAM|
> |-|-|-|-|-|-|-|-|-|
> |BA-Logic|96.75\|63.22|97.03\|64.07|95.37\|63.31|93.23\|61.99|80.06\|61.83|92.36\|60.97|85.38\|62.34|81.09\|60.46|
> |BA-Logic-D|95.86\|63.46|96.31\|64.30|94.42\|63.55|92.14\|62.21|78.58\|62.34|90.84\|61.45|83.91\|62.79|79.64\|60.98|
>
> From the tables, we have the following key finding:
> * **BA-Logic-D** preserves comparable ASR and CA to BA-Logic, showing that explicitly reducing feature-distribution discrepancy does not weaken logic poisoning.
>
> * These results provide additional evidence that BA-Logic remains unnoticeable not only under cosine-similarity constraints and degree-pruning defenses, but also under distribution-level constraints.
>
> We have updated our draft with three new unnoticeable constraints and kindly invite the reviewer to check the additional analysis in **Appendix A.9**
>
> [1] Unnoticeable backdoor attacks on graph neural networks. WWW 2023
>
> [2] Rethinking graph backdoor attacks: A distribution-preserving perspective. KDD 2024
>
> ---
>
> > "Clean accuracy of Ba-Logic still drops noticeably under adaptive defenses, but this trade-off is downplayed in the authors' narrative." (W5)
>
> We would like to clarify the clean accuracy drops of our method against adaptive defenses.
>
> * A successful graph backdoor defense should ideally satisfy two goals at the same time: maintaining the clean accuracy of the defended GNN model, and reducing the effectiveness of the backdoor attack [3].
>
> * BA-Logic consistently maintains a high attack success rate, indicating that defenses do not effectively suppress the backdoor behavior. Meanwhile, the observed clean accuracy drop should be considered as a limitation of current defenses, rather than as a drawback of BA-Logic.
>
> Furthermore, we kindly invite the reviewer to check **Appendix A.4** for more analysis on clean accuracy drop.
>
> [3] Robustness Inspired Graph Backdoor Defense, ICLR 2025

---

> ### Comment · Reviewer_A8JU · 2026-06-10
>
> I thank the authors for the responses and they did address some of my concerns. However, it seems that the Theorem 1 gap and clean-accuracy trade-off remain unaddressed. Specifically, the result is only an upper bound that grows with $\gamma$ (becoming vacuous for large $\gamma$ given the $2 d$ prefactor) and compares the target only against the node's original class, so it still does not establish that higher trigger importance makes the target class win the multi-class prediction. I will thus maintain my current score for now.

---

> > ### Author Response · Authors · 2026-06-30
> >
> > We appreciate the reviewer's feedback, and we hope the following discussion can address your remaining concerns.
> >
> > ## 1. Clarification on theoretical analysis
> >
> > We respectfully clarify the reviewer's misunderstanding regarding our theoretical analysis:
> >
> > ---
> > > 'The result is vacuous for large $\gamma$ given the 2$d$ prefactor'
> >
> > We clarify that the $2d$ prefactor comes from applying a union bound over all $d$ dimensions, as we had emphasized in **Appendix C**. This does not change the conclusion of **Theorem 1** that low important rate of triggers would lead to poor attack performance under the clean-label setting.
> >
> > ---
> > > 'The result does not indicate that higher trigger importance wins a multi-class prediction'
> >
> > We clarify that $y_i$ is the ground-truth class of node $v_i$. In the clean-label setting, poisoned samples are correctly labeled with ground-truth labels, clean neighbors dominate the prediction of the target GNN model, and the GNN model naturally learns correct patterns associated with the labeled class. Thus, predicting the trigger-attached node $v_i$ as target class $y_t$ already addresses the strongest clean prediction logic.
> >
> > ---
> > > 'The result is only an upper bound that grows with $\gamma$'
> >
> > We clarify that establishing a quantitative relationship between IRT and ASR is non-trivial. It demands much stronger assumptions and complex derivations regarding both the graph dataset and the GNN model, which are outside the research scope of our work. Instead, we provide comprehensive empirical validations in **Appendix H**. The results across various datasets and models indicate that ASR and IRT values are closely aligned.
> >
> > ---
> >
> > Moreover, we'd like to emphasize that we have conducted theoretical analysis to highlight why **poisoning inner logic** is essential to effective clean-label graph backdoors. For your convenience, we summarize the contribution of our theoretical analysis as follows:
> >
> > - It bridges the gap that triggers with small importance scores might still cause changes in predicted labels by showing that under the backdoor setting, higher trigger importance scores lead to higher attack success rates.
> >
> > - It proves that increasing the importance scores of triggers is an effective way for successful clean-label backdooring.
> >
> > - It also directly motivates our logic poisoning loss $\mathcal{L}_A$ in **Eq.7**, which enforces the important score of trigger nodes to exceed those of clean neighbors of trigger-attached nodes.
> >
> >
> > ## 2. Clarification on clean-accuracy trade-off
> >
> > We clarify that we have conducted a thorough analysis to demonstrate the clean-accuracy trade-off in our paper. For this rebuttal, we first summarize the average clean-accuracy drop (**CAD**) along with the main results of our BA-Logic in **Tab 3** as follows:
> >
> > |Dataset|Vanilla Acc.|**ASR**|**Clean accuracy**|**CAD**|
> > |-|-|-|-|-|
> > |Cora|84.11|**98.20**|**83.72**|**0.39**|
> > |Pubmed|86.47|**96.89**|**85.79**|**0.68**|
> > |Flickr|46.17|**99.90**|**45.70**|**0.47**|
> > |Arxiv|66.44|**98.03**|**66.02**|**0.43**|
> >
> > From the table, we observe that our BA-Logic achieves ASR higher than **96.89%** while keeping CAD below **0.68%** across all four datasets. The results highlight that our method already preserves the best trade-off between ASR and CAD when facing no defense and existing defense methods.
> >
> > Furthermore, we propose four adaptive defenses tailored for logic poisoning to further strengthen our contribution in **Sec 5.4**. For your convenience, we summarize **ASR|CAD(%)** of our BA-Logic and two leading baselines on **Cora and Arxiv** when facing two adaptive defenses as follows:
> >
> > **Cora:**
> >
> > |Defense|EBA-C|UGBA-C|**BA-Logic**|
> > |-|-|-|-|
> > |GM|19.38\|17.02|50.41\|12.96|**95.32\|11.81**|
> > |CD|21.78\|15.99|62.71\|13.96|**90.47\|14.08**|
> >
> > **Arxiv:**
> >
> > |Defense|EBA-C|UGBA-C|**BA-Logic**|
> > |-|-|-|-|
> > |GM|18.45\|2.06|49.87\|2.66|**92.36\|5.55**|
> > |CD|19.56\|1.77|42.51\|2.38|**85.38\|4.05**|
> >
> > From the results, we conclude:
> >
> > - The adaptive defense can partially weaken the backdoor, indicating promising directions against logic poisoning. However, our BA-Logic consistently maintains the highest ASR (generally over 85%). This indicates the superiority of our BA-Logic in poisoning inner logic for clean-label backdooring.
> >
> > - Clean-accuracy drop highlights the need for further in-depth investigation into adaptive defenses, instead of our limitation. This is because successful graph backdoor defenses should satisfy two goals at the same time: maintaining the clean accuracy of the defended GNN model, and reducing the effectiveness of the backdoor attack[1].
> >
> > Besides, we kindly invite the reviewer to check **Appendix A.4** for more analysis on clean accuracy drop, and to check **Appendix G** for more analysis on adaptive defenses.
> >
> > [1] Robustness Inspired Graph Backdoor Defense, ICLR 2025

---

### Review · Reviewer_QQmo · 2026-05-15

**Summary Of Contributions:**

The paper studies clean-label graph backdoor attacks on Graph Neural Networks (GNNs), where attackers cannot modify training labels. The authors argue that existing graph backdoor attacks fail under clean-label settings because the injected triggers are not sufficiently important to alter the prediction logic of GNNs. To address this issue, the paper proposes Ba-Logic, a framework that jointly optimizes poisoned node selection and logic-poisoning trigger generation. Extensive experiments across multiple datasets and GNN architectures show improved attack success rates over prior methods.

**Strengths:**

1. Studies an important and realistic clean-label attack setting.
2. Introduces a novel logic-poisoning perspective for graph backdoor attacks.
3. Provides extensive empirical evaluation on multiple datasets and GNN models.
4. Demonstrates generally strong attack performance compared to existing baselines.

**Weaknesses:**

1. Theoretical analysis is somewhat intuitive and lacks stronger practical justification.
2. Optimization approximations and convergence behavior are not sufficiently analyzed.
3. Missing comparisons with several recent graph attack baselines.
4. Scalability and computational cost analysis are limited.
5. Some implementation and hyper-parameter details are missing for reproducibility.

**Audience:**

Yes

**Audience Explanation:**

Graph backdoor attacks and adversarial robustness for GNNs are active research areas in trustworthy machine learning and graph learning. The problem of clean-label graph backdoor attacks is particularly relevant because modifying labels is often unrealistic in real-world applications. The proposed idea of poisoning the prediction logic of GNNs introduces a different attack perspective that could motivate future research on both attacks and defenses. Researchers working on graph security, adversarial machine learning, trustworthy AI, and GNN robustness would likely find the paper interesting.

**Broader Impact Concerns:**

Research on attack methodologies is also important for understanding vulnerabilities and developing stronger defenses. The paper would benefit from a clearer discussion regarding the responsible use of adversarial research and the defensive implications of the proposed framework.

**Claims And Evidence:**

Yes

**Claims Explanation:**

The empirical results generally support the paper’s claims that Ba-Logic improves clean-label graph backdoor attack effectiveness across multiple datasets and GNN architectures. The experimental evaluation is relatively comprehensive and includes ablation studies that help support the proposed intuition regarding logic poisoning and trigger importance.

However, some concerns remain regarding the theoretical analysis and practical validation. Theorem 1 mainly provides an intuitive upper-bound analysis showing that stronger trigger importance increases the probability of target prediction. While reasonable, the novelty and practical significance of the bound are not fully clear. Additionally, the optimization framework uses first-order approximations and limited inner optimization steps without sufficient analysis of convergence or optimization stability.

Some experimental observations also require further clarification. For example, Ba-Logic performs relatively weaker on Arxiv and Flickr datasets, and the ERBA baseline reports nearly zero ASR on certain datasets, raising questions about implementation settings and fairness of comparison. Furthermore, the paper lacks runtime and memory consumption analysis, and scalability experiments on larger graph datasets are missing.

Overall, the experimental evidence is reasonably convincing, but stronger theoretical clarification, reproducibility details, and broader empirical evaluation would further strengthen the claims.

**Requested Changes:**

1. Clarify the novel theoretical insight provided by Theorem 1, as the result currently appears somewhat intuitive since stronger trigger importance would naturally increase target prediction probability.

2. Provide additional empirical validation demonstrating how tightly the theoretical upper bound correlates with actual attack success rates across datasets and GNN architectures.

3. Discuss the sensitivity of the theoretical assumptions regarding class centroid separability, feature distributions, and graph structure under noisy or heterogeneous graph settings.

4. Clarify how the first-order approximation and limited inner-loop optimization iterations affect optimization accuracy, convergence, and attack performance.

5. Include runtime, training time, and GPU memory usage comparisons in the experimental section.

6. Provide additional discussion regarding the relatively weaker performance of Ba-Logic on Arxiv and Flickr datasets.

7. Clarify whether the official implementation and recommended hyper-parameters of ERBA were strictly followed, as the reported ASR values on Flickr and Arxiv appear unusually low.

8. Include scalability experiments on larger-scale graph datasets such as OGB Products.

9. Compare against more recent graph attack baselines such as GOttack [1] and EvA [2], or clarify why these methods are not applicable.

10. Add more implementation details and hyper-parameter settings for all baselines and models in the appendix to improve reproducibility.

[1] GOttack: Universal Adversarial Attacks on Graph Neural Networks via Graph Orbits Learning, ICLR 2025

[2] EvA: Evolutionary Attacks on Graphs, ICLR 2026

---

> ### Author Response · Authors · 2026-05-27
> **Response to Reviewer QQmo Part 1**
>
> We appreciate the reviewer for the constructive feedback. We hope the following discussions can address the reviewer's remaining concerns.
>
> ## 1. Clarification on Theoretical Analysis
>
> ---
>
> > "Clarify the novel theoretical insight provided by Theorem 1."(RC. 1)
>
>
> Intuitively, the importance scores of triggers reflect their influence on the predictions of backdoored GNNs. However, there remains a gap: triggers with small importance scores might still cause changes in predicted labels.
>
> To bridge this gap, our theorem shows that under the backdoor setting, higher trigger importance scores lead to higher attack success rates.  This proves that increasing the importance scores of triggers is an effective way for successful clean-label backdooring. It also directly motivates our logic poisoning loss $\mathcal{L}_A$ in **Eq.7**, which enforces the important score of trigger nodes to exceed those of clean neighbors of trigger-attached nodes.
>
> ---
>
> > "Provide empirical validation on how tightly the theoretical upper bound correlates with ASR across datasets and models." (RC. 2)
>
> For this rebuttal, we record the average **ASR|CA(%) and IRT(%)** of our method on four graphs with GCN, GIN, and GAT as target models as follows:
>
> **Cora**
>
> |Target Model|ASR\|CA|IRT|
> |-|-|-|
> |GCN|98.52\|83.59|99.13|
> |GIN|98.97\|83.81|99.42|
> |GAT|97.12\|83.76|97.86|
>
> **Pubmed**
>
> |Target Model|ASR\|CA|IRT|
> |-|-|-|
> |GCN|96.75\|86.03|97.41|
> |GIN|99.04\|86.21|99.53|
> |GAT|94.88\|85.13|95.62|
>
> **Flickr**
>
> |Target Model|ASR\|CA|IRT|
> |-|-|-|
> |GCN|99.98\|46.05|99.99|
> |GIN|100.00\|46.14|100.00|
> |GAT|99.72\|44.91|99.86|
>
> **Arxiv**
>
> |Target Model|ASR\|CA|IRT|
> |-|-|-|
> |GCN|98.04\|65.82|98.61|
> |GIN|97.62\|66.83|98.24|
> |GAT|98.43\|65.40|98.89|
>
>
> We have the following key findings from the tables:
>
> * ASR and IRT values are closely aligned across diverse graphs and models. It indicates that the theoretical analysis of triggers with a higher importance rate can achieve better attack performance remains valid for various graphs
>
> * Our method consistently achieves high ASR across datasets and models. This indicates logic poisoning remains effective facing various graph correlations, as the target model with poisoned prediction logic still identifies our trigger as important for prediction
>
> ---
>
> > "Discuss the sensitivity of the theoretical assumptions under noisy or heterogeneous graph settings." (RC. 3)
>
> We clarify that we investigated how the noise and heterophily settings affect our method in **Appendix H** and **Appendix I**. Specifically, we generate synthetic graphs based on real-world graphs by varying **edge homophily $h$**, which is the fraction of edges in a graph that connect nodes with the same labels. For your convenience, we present the average **ASR|CA(%) and IRT(%)** of our method on two synthetic graphs with various $h$ as follows:
>
> **syn-flickr**
>
> |$h$|ASR\|CA|IRT|
> |-|-|-|
> |0.1|88.20\|25.59|90.87|
> |0.3|89.83\|21.04|91.18|
> |0.5|91.17\|36.48|92.35|
> |0.7|93.21\|37.74|94.52|
> |0.9|96.60\|44.39|98.09|
>
> **syn-arxiv**
>
> |$h$|ASR\|CA|IRT|
> |-|-|-|
> |0.1|90.12\|38.53|89.47|
> |0.3|86.08\|53.89|87.89|
> |0.5|87.41\|55.37|86.49|
> |0.7|91.34\|65.72|94.25|
> |0.9|96.25\|86.22|97.28|
>
> We have the following observations from the tables:
>
> * Our method consistently achieves high ASR across different synthetic graphs. This indicates that our triggers with a high importance rate can still be identified as crucial for prediction on graphs with complex properties
>
> * The graph properties, such as homophily, mainly affect CA of the target GNNs rather than ASR of our method. This indicates that graph property, such as homophily, mainly affects the generalization of GNNs on clean nodes rather than our logic poisoning
>
> ---
>
> > "Clarify how the first-order approximation and limited inner-loop optimization iterations affect optimization accuracy, convergence, and attack performance." (RC. 4)
>
> We clarify that the first-order approximation is a commonly used method to avoid costly second-order computation while retaining strong empirical performance [1,2]. For this rebuttal, we report the **ASR, CA, IRT (%)** and **Training Time (s)** by varying the inner training epoch $N$ in our training algorithm in **Appendix E** as follows:
>
> **Pubmed**
>
> |Metric↓ & $N$→|5|10|15|20|
> |-|-|-|-|-|
> |ASR|95.84|96.75|96.88|96.91|
> |CA|86.08|86.03|86.01|85.98|
> |IRT|90.91|92.36|92.58|92.63|
> |Time|34.5|46.8|58.9|71.1|
>
>
> **Arxiv**
>
> |Metric↓ & $N$→|5|10|15|20|
> |-|-|-|-|-|
> |ASR|96.94|98.04|98.11|98.08|
> |CA|65.91|65.82|65.79|65.75|
> |IRT|91.27|93.08|93.25|93.31|
> |Time|111.5|155.7|199.9|244.2|
>
>
> Key insights:
>
> * ASR and IRT saturate quickly, showing that limited inner-loop updates are sufficient for effective logic poisoning.
> * CA remains stable across different $N$, indicating the approximation does not harm benign performance.
>
>
> [1] Model-Agnostic Meta-Learning for Fast Adaptation of Deep Networks, ICML 2017
>
> [2] DARTS: Differentiable Architecture Search, ICLR 2019

---

> ### Author Response · Authors · 2026-05-27
> **Response to Reviewer QQmo Part 2**
>
> ## 2. Clarification on Scalability
>
>
> > "Include runtime, training time, and GPU memory usage comparisons in the experimental section, and scalability experiments on larger-scale graph datasets such as OGB Products." (RC. 5&8)
>
>
> We clarify that we had evaluated our method on `ogbn-products` with 2.4M nodes, and we have kindly invited the reviewers to the results in **Appendix A.2**.  For your convenience, we present the results as follows:
>
> |Surrogate Model|Training Time(s)|GPU Memory Peak(GB)|ASR\|CA(\%)|
> |-|-|-|-|
> |GCN|1678.05|23.52|87.07\|78.51|
> |GraphSAGE|1531.39|22.36|83.69\|80.27|
>
> From the table, we have the following key observations:
>
> * The scalability of our method is demonstrated with feasible resource usage on the industry-scale graph with 2.4M nodes, indicating that our method remains practical for large-scale attacks
>
> * Training time for large-scale attacks is acceptable given the performance. It is also consistent with the approximately linear complexity with respect to graph size, as we analyzed in **Appendix B**
>
>
> ## 3. Additional Analysis on Performance
>
>
> > "Provide additional discussion regarding the relatively weaker performance of Ba-Logic on Arxiv and Flickr datasets."(RC. 6)
>
> We clarify that BA-Logic consistently exhibits leading performance compared with baselines on Arxiv and Flickr.  And we summarize the reasons as follows:
>
> * Compared with Cora and Pubmed, Arxiv and Flickr are slightly larger and contain broader local neighborhoods, which can mildly dilute the influence of a fixed-size trigger.
>
> * Arxiv and Flickr nodes might involve more diverse semantics, such as paper topics in Arxiv and visual or social contexts in Flickr, making the clean prediction logic less concentrated.
>
> * A higher node degree indicates richer clean-neighbor evidence, making it slightly harder for the trigger to dominate the model’s prediction.
>
>
> ---
>
> > "Clarify whether the official implementation and recommended hyper-parameters of ERBA were strictly followed, as the reported ASR values on Flickr and Arxiv appear unusually low."(RC. 7)
>
> We clarify that we followed the official implementation of baselines, and we provided the details of adaptation in **Appendix F.2**. We faithfully reproduced baselines based on their original paper and fine-tuned all methods on the validation set before evaluation.
>
> For the performance of ERBA, we provide the following insights:
>
> * ERBA tailors for graph classification tasks, utilizing Erdős–Rényi random subgraphs as triggers, which are generally irrelevant for prediction in practice.
>
> * ERBA selects poisoned nodes at random, leading to inefficient use of the limited attack budget.
>
> To adapt ERBA for a fair comparison, we made the following efforts:
>
> * We maintain a fixed node size of three within the random graph
> * We select poisoned nodes from training nodes belonging to the target class
> * All other settings remain consistent with those described in the original work.
>
> ---
>
> > "Compare against more recent graph attack baselines such as GOttack and EvA, or clarify why these methods are not applicable." (RC. 9)
>
> For this rebuttal, we extend our method and compare it with the additional baselines suggested by the reviewer. We report the best **ASR|CA(%)** of these methods on four graph datasets as follows:
>
> |Dataset|EvA|GOttack|BA-Logic|
> |-|-|-|-|
> |Cora|91.84\|82.76|89.57\|82.91|**98.20\|83.72**|
> |Pubmed|94.38\|85.42|92.16\|85.31|**96.89\|85.79**|
> |Flickr|86.73\|45.38|84.92\|45.61|**99.90\|45.70**|
> |Arxiv|93.54\|65.74|91.89\|65.88|**98.03\|66.02**|
>
> From the table, we'd like to highlight that:
>
> - EvA and GOttack are strong because EvA directly searches for discrete edge perturbations with evolutionary optimization, while GOttack exploits topology-aware graph orbits to find effective structural perturbations.
>
> - EvA and GOttack optimize adversarial misclassification rather than clean-label backdoor logic poisoning, so they do not explicitly make the victim GNN consistently treat the injected trigger as the dominant evidence for the target class.
>
> * Our BA-Logic consistently maintains the outstanding ASR across various datasets. This indicates the superiority of our BA-Logic in poisoning inner logic for clean-label backdooring.
>
>
> We have updated our draft with the comparison in **Appendix A.10**, and we kindly invite the reviewer to check the details.
>
>
> ---
>
> > "Add more implementation details and hyper-parameter settings for all baselines and models to improve reproducibility." (RC. 10)
>
> For the reproducibility of our BA-Logic and comparing baselines, we made the following efforts:
>
> * We clarify that we reported all the details of our method in **Appendix F.3**, including training parameters, hyperparameters, attack budgets across diverse datasets, and other details.
>
> * We also reported the implementation details of baselines in **Appendix F.2**. Additionally, we open-sourced our code repository.
>
> We believe these efforts ensure the reproducibility of our work.

---

> > ### Comment · Reviewer_QQmo · 2026-06-30
> >
> > I thank the authors for the detailed rebuttal. They addressed most of my experimental concerns, including additional comparisons, scalability analysis, and implementation details. However, my main theoretical concern remains unresolved. Theorem 1 still provides only an upper bound whose tightness is not theoretically justified and compares the target class only against the original class, rather than establishing that the target class becomes the highest-probability class in the multi-class setting. The theoretical trade-off between increasing trigger importance and maintaining clean accuracy is also not analyzed. I also agree with the concerns raised by the other reviewers. Therefore, I will maintain my current score.

---

> > > ### Author Response · Authors · 2026-06-30
> > >
> > > We are glad to hear that our response has addressed most of the reviewer's concerns. We hope the following discussion can address the remaining concerns.
> > >
> > > ## 1. Clarification on theoretical analysis
> > >
> > > We respectfully clarify the reviewer's misunderstanding regarding our theoretical analysis:
> > >
> > > ---
> > > > 'The result does not support the multi-class setting'
> > >
> > > We clarify that $y_i$ is the ground-truth class of node $v_i$. In the clean-label setting, poisoned samples are correctly labeled with ground-truth labels, clean neighbors dominate the prediction of the target GNN model, and the GNN model naturally learns correct patterns associated with the labeled class. Thus, predicting the trigger-attached node $v_i$ as target class $y_t$ already addresses the strongest clean prediction logic.
> > >
> > > ---
> > > > 'The result is only an upper bound that grows with $\gamma$'
> > >
> > > We clarify that establishing a quantitative relationship between IRT and ASR is non-trivial. It demands much stronger assumptions and complex derivations regarding both the graph dataset and the GNN model, which are outside the research scope of our work. Instead, we provide comprehensive empirical validations in **Appendix H**. The results across various datasets and models indicate that ASR and IRT values are closely aligned.
> > >
> > > ---
> > >
> > > Moreover, we'd like to emphasize that we have conducted theoretical analysis to highlight why **poisoning inner logic** is essential to effective clean-label graph backdoors. For your convenience, we summarize the contribution of our theoretical analysis as follows:
> > >
> > > - It bridges the gap that triggers with small importance scores might still cause changes in predicted labels by showing that under the backdoor setting, higher trigger importance scores lead to higher attack success rates.
> > >
> > > - It proves that increasing the importance scores of triggers is an effective way for successful clean-label backdooring.
> > >
> > > - It also directly motivates our logic poisoning loss $\mathcal{L}_A$ in **Eq.7**, which enforces the important score of trigger nodes to exceed those of clean neighbors of trigger-attached nodes.
> > >
> > >
> > > ## 2. Clarification on clean-accuracy trade-off
> > >
> > > We clarify that we have conducted a thorough analysis to demonstrate the clean-accuracy trade-off in our paper. For this rebuttal, we first summarize the average clean-accuracy drop (**CAD**) along with the main results of our BA-Logic in **Tab 3** as follows:
> > >
> > > |Dataset|Vanilla Acc.|**ASR**|**Clean accuracy**|**CAD**|
> > > |-|-|-|-|-|
> > > |Cora|84.11|**98.20**|**83.72**|**0.39**|
> > > |Pubmed|86.47|**96.89**|**85.79**|**0.68**|
> > > |Flickr|46.17|**99.90**|**45.70**|**0.47**|
> > > |Arxiv|66.44|**98.03**|**66.02**|**0.43**|
> > >
> > > From the table, we observe that our BA-Logic achieves ASR higher than **96.89%** while keeping CAD below **0.68%** across all four datasets. The results highlight that our method already preserves the best trade-off between ASR and CAD when facing no defense and existing defense methods.
> > >
> > > Furthermore, we propose four adaptive defenses tailored for logic poisoning to further strengthen our contribution in **Sec 5.4**. For your convenience, we summarize **ASR|CAD(%)** of our BA-Logic and two leading baselines on **Cora and Arxiv** when facing two adaptive defenses as follows:
> > >
> > > **Cora:**
> > >
> > > |Defense|EBA-C|UGBA-C|**BA-Logic**|
> > > |-|-|-|-|
> > > |GM|19.38\|17.02|50.41\|12.96|**95.32\|11.81**|
> > > |CD|21.78\|15.99|62.71\|13.96|**90.47\|14.08**|
> > >
> > > **Arxiv:**
> > >
> > > |Defense|EBA-C|UGBA-C|**BA-Logic**|
> > > |-|-|-|-|
> > > |GM|18.45\|2.06|49.87\|2.66|**92.36\|5.55**|
> > > |CD|19.56\|1.77|42.51\|2.38|**85.38\|4.05**|
> > >
> > > From the results, we conclude:
> > >
> > > - The adaptive defense can partially weaken the backdoor, indicating promising directions against logic poisoning. However, our BA-Logic consistently maintains the highest ASR (generally over 85%). This indicates the superiority of our BA-Logic in poisoning inner logic for clean-label backdooring.
> > >
> > > - Clean-accuracy drop highlights the need for further in-depth investigation into adaptive defenses, instead of our limitation. This is because successful graph backdoor defenses should satisfy two goals simultaneously: maintaining the clean accuracy of the defended GNN model and reducing the effectiveness of the backdoor attack [1].
> > >
> > > Besides, we kindly invite the reviewer to check **Appendix A.4** for more analysis on clean accuracy drop, and to check **Appendix G** for more analysis on adaptive defenses.
> > >
> > > [1] Robustness Inspired Graph Backdoor Defense, ICLR 2025

---

### Author Response · Authors · 2026-05-27
**General Response from Authors**

Dear Reviewers, Action Editors, and Editors In Chief of Submission 8277:

We sincerely thank you again for your valuable time and effort in assessing our work. We have responded to the questions of reviewers through additional experiments and discussions. We also update our manuscript and highlight the corresponding content in blue. Specifically, we made the following major updates:

* We further propose three unnoticeable constraints tailored to logic poisoning triggers during rebuttal, aiming to show the robustness of our logic poisoning design and further strengthen our technical contributions

* We add comparison with two more latest graph attack baselines to highlight the leading performance of our method.

* Other additional experiments and discussions raised by reviewers.

We also appreciate the reviewers' recognition of the following key contributions:

* We investigate an important and novel topic of poisoning the inner prediction logic for clean-label graph backdoor attacks. We study a realistic threat model, in which the attackers can not alter the present label for attacks.

* We conduct a theoretical analysis that reveals the existing methods fail to poison the inner prediction logic of target models under clean-label settings. This failure occurs because clean neighbors exhibit typical patterns strongly associated with the target class and dominate the prediction of the GNN model, while the injected triggers are treated as irrelevant information in prediction, resulting in poor clean-label backdoor performance.

* We propose a novel framework, BA-Logic, which can poison the inner prediction logic of target models for clean-label graph backdoor attacks. Specifically, BA-Logic optimizes the importance scores of trigger nodes to exceed those of clean neighbor nodes for effective logic-poisoning trigger generator. A poisoned node selection process designed for clean-label setting is further adopted.

* We conduct extensive experiments to demonstrate our method's superiority. The evaluations of our method on 18 graph datasets with 15 baselines across three GNN downstream tasks, 6 backbone GNNs, against 8 defense methods, consistently demonstrate **superior ASR over SOTA methods**.

We appreciate the thoughtful feedback from the reviewers again. And we hope our efforts during rebuttal can adequately address the remaining concerns.

Sincerely,

Authors of Submission 8277